# Conditional Testing based on Localized Conformal $p$-values

**Xiaoyang Wu**[*], **Lin Lu**[*], **Zhaojun Wang**, **Changliang Zou**[†]
NITFID, School of Statistics and Data Science, LPMC and KLMDASR and LEBPS,
Nankai University, Tianjin, China
`xywu@mail.nankai.edu.cn`, `linlu102099@gmail.com`,
`zjwangnk@126.com`, `zoucl@nankai.edu.cn`

## Abstract

In this paper, we address conditional testing problems through the conformal inference framework. We define the localized conformal $p$-values by inverting prediction intervals and prove their theoretical properties. These defined $p$-values are then applied to several conditional testing problems to illustrate their practicality. Firstly, we propose a conditional outlier detection procedure to test for outliers in the conditional distribution with finite-sample false discovery rate (FDR) control. We also introduce a novel conditional label screening problem with the goal of screening multivariate response variables and propose a screening procedure to control the family-wise error rate (FWER). Finally, we consider the two-sample conditional distribution test and define a weighted U-statistic through the aggregation of localized $p$-values. Numerical simulations and real-data examples validate the superior performance of our proposed strategies.

## 1 Introduction

Nowadays, conformal inference has become an increasingly popular framework for quantifying uncertainty of machine learning models. Suppose we have i.i.d. training data $\mathcal{D}_1 = \{(\mathbf{X}_{1i}, Y_{1i})\}_{i=1}^n$ and test data $\mathcal{D}_2 = \{(\mathbf{X}_{2j}, Y_{2j})\}_{j=1}^m$, where test responses $\{Y_{2j}\}_{j=1}^m$ are unobserved. The goal is to construct prediction intervals $\mathrm{PI}(\mathbf{X}_{2j})$ for $\mathbf{X}_{2j} \in \mathcal{D}_2$ with marginal coverage guarantee

$$\Pr(Y_{2j} \in \mathrm{PI}(\mathbf{X}_{2j})) \geq 1 - \alpha. \tag{1.1}$$

By sample splitting or cross-fitting, conformal prediction methods (Vovk et al., 2005) can be coupled with any machine learning algorithm to construct distribution-free prediction intervals with finite-sample coverage guarantee. However, the marginal coverage (1.1) alone is not sufficient for an efficient prediction interval. A marginally valid prediction interval could have a miscoverage rate much higher than $\alpha$ in some subgroups of the data. Therefore, the conditional coverage is also an important aspect:

$$\Pr(Y_{2j} \in \mathrm{PI}(\mathbf{X}_{2j}) \mid \mathbf{X}_{2j} = \mathbf{x}) \geq 1 - \alpha. \tag{1.2}$$

Although appealing, achieving (1.2) in a finite-sample and distribution-free context is impossible (Vovk, 2013). Recent works have proposed many methods to construct prediction intervals with approximate or asymptotic conditional coverage guarantee by either modifying the calibration step (Lei & Wasserman, 2013; Guan, 2023) or using different score functions (Romano et al., 2019; Chernozhukov et al., 2021).

In addition to prediction intervals, the conformal inference framework is also valuable for other inference problems. With a similar calibration procedure, the conformal $p$-value is defined to address various testing problems. This includes classical two-sample tests (Hu & Lei, 2023) and multiple testing problems such as outlier detection (Bates et al., 2023; Zhang et al., 2022) and data selection/sampling (Jin & Candès, 2023b; Wu et al., 2023). Therefore, as a parallel development, our

---

[*]Equal contribution.
[†]Correspondence to: Changliang Zou, Email: zoucl@nankai.edu.cn.

paper moves beyond conditional prediction intervals and defines conformal $p$-values tailored for conditional testing problems.

We define localized conformal $p$-values by leveraging recent works Guan (2023) and Hore & Barber (2024) which constructed prediction intervals to adapt to the conditional distribution of the response. We present some fundamental properties of our defined $p$-values and demonstrate why they can effectively resolve these problems. More importantly, we also consider several non-trivial applications of localized conformal $p$-values, encompassing various testing problems with different error criteria (e.g., FDR, FWER or type I error). We show that the proposed testing rule can ensure valid corresponding error rate control. Here we provide a brief overview of those problems and defer rigorous formulations to later sections.

- **Conditional outlier detection:** Outlyingness in data sets may come from various forms of heterogeneity. One of the important cases is that a few individuals in the test data do not share the same regression functions with majority (Peng et al., 2023). This amounts to detecting outliers in the functional relationship between the response and covariates that can be represented by the conditional distribution of $Y \mid \mathbf{X}$. This is common in many spatial or temporal contexts where the scale or variance of the response variable varies with location or time (Catterson et al., 2010).

- **Conditional label screening:** Consider a scenario where the response $\mathbf{Y}$ is multivariate. Our goal is to determine whether each component of the response satisfies a pre-specified rule for an unlabeled data point. For example, in the LLM factuality problem (Mohri & Hashimoto, 2024; Cherian et al., 2024) we have a vector of claims output by a LLM. It is desirable to screen out false claims to ensure reliability. In this example, the conditional performance of the screening procedure is important, which needs to be addressed properly to avoid extreme or imbalanced results.

- **Two-sample conditional distribution test:** The goal is to test for equality of conditional distributions of $Y \mid \mathbf{X}$ between two samples (Hu & Lei, 2023). This is also known as the problem of comparison of regression curves (Dette & Neumeyer, 2003). Some contemporary applications include assessing casuality by testing whether the conditional distribution of the response given the covariates remains the same between two samples (Bühlmann, 2020) and testing whether the pre-trained model can still perform well on a new test sample (Farahani et al., 2021).

### 1.1 OUR CONTRIBUTIONS

In this article, building on Hore & Barber (2024)'s randomly localized conformal prediction interval, we construct localized conformal $p$-values, study their theoretical properties, and apply them to several conditional testing problems. Our contributions to the applications are as follows:

- For the conditional outlier detection problem, we propose a procedure by utilizing the Benjamini-Hochberg (BH) procedure (Benjamini & Hochberg, 1995) and conditional calibration technique (Fithian & Lei, 2022). Our procedure controls the FDR in finite-sample.

- We elaborate a novel conditional label screening problem with rigorous formulation. We use the localized conformal $p$-value to construct a screening procedure with finite-sample marginal FWER control. We also present a conditional FWER inflation bound of our procedure to demonstrate its robustness against conditionality.

- We propose a U-statistic for the two-sample conditional distribution testing problem by aggregating the localized conformal $p$-values. The asymptotic normality of the test statistic under null and local alternatives is established.

We also validate our methods through simulations and real-data experiments. With commonly used prediction algorithms, our proposed methods exhibit superiority in terms of both validity and power compared to existing approaches.

### 1.2 RELATED WORKS

Conformal inference was originally designed to construct prediction intervals and enjoy valid and distribution-free properties with the only assumption that the data are exchangeable. There have

been several extensions to this framework. For example, Tibshirani et al. (2019) proposed weighted conformal prediction for covariate shift settings, and Romano et al. (2019) introduced conformalized quantile regression to account for heteroscedasticity. Other applications of the conformal inference framework include causal inference (Lei & Candès, 2021), selective inference (Bao et al., 2024), and survival analysis (Candès et al., 2023), among others. Besides conformal inference, we also briefly review recent literature about our application problems in Appendix B.

## 1.3 NOTATIONS

The $\mathbb{I}\{\cdot\}$ is the indicator function and $\|\cdot\|_2, \|\cdot\|_\infty$ are the $L_2$- and $L_\infty$-norm. The $[n]$ denotes the set $\{1, 2, \ldots, n\}$. The $\delta_{\mathbf{z}}$ denotes the point mass at value $\mathbf{z}$. The notation $Q(\alpha; P)$ denotes the $\alpha$-th quantile of distribution $P$. The subscripts of $\mathbb{E}$ and $\mathrm{Pr}$ indicate distributions of random variables in the expectations and probabilities.

## 2 LOCALIZED CONFORMAL $p$-VALUES

In this section, we first revisit the definition of the split conformal prediction and its localized extensions by Guan (2023) and Hore & Barber (2024) in Section 2.1. Then we propose the localized conformal $p$-values by inverting the localized prediction intervals in Section 2.2. At last we discuss basic properties of the defined $p$-values in Section 2.3.

### 2.1 RECAP: CONFORMAL PREDICTION AND LOCALIZATION

We consider data pair $(\mathbf{X}, Y) \in \mathcal{X} \times \mathcal{Y}$. Suppose we have $\mathcal{D}_1 = \{(\mathbf{X}_{1i}, Y_{1i})\}_{i=1}^n \sim P_1 = P_{1,\mathbf{X}} \times P_{1,Y|\mathbf{X}}$ and $\mathcal{D}_2 = \{(\mathbf{X}_{2j}, Y_{2j})\}_{j=1}^m$ with each data point $(\mathbf{X}_{2j}, Y_{2j}) \sim P_{2,j} = P_{2,\mathbf{X}} \times P_{2,j,Y|\mathbf{X}}$. In different problems, the responses $Y_{2j}$ of the second sample can be observed or unobserved.

In the classical split conformal prediction, $\mathcal{D}_1$ is divided into the training and calibration sets $\mathcal{D}_1 = \mathcal{D}_\mathcal{T} \cup \mathcal{D}_\mathcal{C}$ with index sets $\mathcal{T} \cup \mathcal{C} = \{1, 2, \ldots, n\}$. The training set $\mathcal{D}_\mathcal{T}$ is used to train a prediction function $\widehat{\mu}(\mathbf{X})$ for the response $Y$ and construct a non-conformity function $V(\mathbf{X}, Y)$ which measures the similarity between the prediction and the true response. We then apply the score function to the calibration set $\mathcal{D}_\mathcal{C}$ to compute calibration scores $\{V_{1i}\}_{i \in \mathcal{C}}$ and obtain the quantile threshold

$$\widehat{q}_\alpha = Q\left(1 - \alpha; \frac{1}{|\mathcal{C}| + 1} \sum_{i \in \mathcal{C}} \delta_{V_{1i}} + \frac{1}{|\mathcal{C}| + 1} \delta_\infty\right). \quad (2.1)$$

The split conformal prediction interval is defined as $\mathrm{PI}(\mathbf{X}_{2j}) = \{y : V(\mathbf{X}_{2j}, y) \leq \widehat{q}_\alpha\}$. If $\mathcal{D}_1 \cup \{(\mathbf{X}_{2j}, Y_{2j})\}$ are exchangeable, the prediction interval $\mathrm{PI}(\mathbf{X}_{2j})$ is finite-sample valid in the sense that $\mathrm{Pr}(Y_{2j} \in \mathrm{PI}(\mathbf{X}_{2j})) \geq 1 - \alpha$.

However, calibrating marginally with (2.1) does not account for the local information of the test point. Guan (2023) proposed the localized conformal prediction by calibrating with the quantile of a weighted empirical distribution

$$\widehat{q}^*_{\widehat{\alpha}, \mathrm{L}} = Q\left(1 - \widehat{\alpha}; \sum_{i \in \mathcal{C}} H^*(\mathbf{X}_{1i}, \mathbf{X}_{2j})\delta_{V_{1i}} + H^*(\mathbf{X}_{2j}, \mathbf{X}_{2j})\delta_\infty\right),$$

where $H^*(\mathbf{x}, \mathbf{x}') = \frac{H(\mathbf{x}, \mathbf{x}')}{\sum_{k \in \mathcal{C}} H(\mathbf{X}_{1k}, \mathbf{X}_{2j}) + H(\mathbf{X}_{2j}, \mathbf{X}_{2j})}$ for some kernel function $H(\cdot, \cdot)$ characterizing the similarity between its two arguments. Here $\widehat{\alpha}$ is the adjusted level to guarantee finite-sample coverage. Despite its efficiency, the LCP method needs to compute the adjusted level $\widehat{\alpha}$, which is complex and computationally inefficient. As an improvement, Hore & Barber (2024) proposed a randomization technique to circumvent level adjustment. Their method first samples $\tilde{\mathbf{X}}_{2j}$ from the distribution $H(\mathbf{X}_{2j}, \cdot)$, takes the threshold as

$$\widehat{q}_{\alpha, \mathrm{L}} = Q\left(1 - \alpha; \sum_{i \in \mathcal{C}} \tilde{H}^*(\mathbf{X}_{1i}, \tilde{\mathbf{X}}_{2j})\delta_{V_{1i}} + \tilde{H}^*(\mathbf{X}_{2j}, \tilde{\mathbf{X}}_{2j})\delta_\infty\right),$$

and define the prediction interval as

$$\mathrm{PI}_\mathrm{L}(\mathbf{X}_{2j}) = \{y : V(\mathbf{X}_{2j}, y) \leq \widehat{q}_{\alpha, \mathrm{L}}\}, \quad (2.2)$$

where $\tilde{H}^*(\mathbf{x}, \mathbf{x}') = \frac{H(\mathbf{x}, \mathbf{x}')}{\sum_{k \in \mathcal{C}} H(\mathbf{X}_{1k}, \tilde{\mathbf{X}}_{2j}) + H(\mathbf{X}_{2j}, \tilde{\mathbf{X}}_{2j})}$. If $\{(\mathbf{X}_{2j}, Y_{2j})\} \cup \mathcal{D}_1$ are exchangeable, we can compute the density ratio of $\mathbf{X}_{2j}$ and $\mathbf{X}_{1i}$ conditional on $\tilde{\mathbf{X}}_{2j}$ as

$$\frac{\mathrm{d}P_{2, \mathbf{X} | \tilde{\mathbf{X}}_{2j}}}{\mathrm{d}P_{1, \mathbf{X}}}(\mathbf{x}) = \frac{f_{2, \mathbf{X}}(\mathbf{x}) H(\mathbf{x}, \tilde{\mathbf{X}}_{2j})}{f_{1, \mathbf{X}}(\mathbf{x}) \int_{\mathcal{X}} H(\cdot, \tilde{\mathbf{X}}_{2j}) \, \mathrm{d}P_{2, X}(x)} = c \cdot H(\mathbf{x}, \tilde{\mathbf{X}}_{2j}) \propto H(\mathbf{x}, \tilde{\mathbf{X}}_{2j}), \qquad (2.3)$$

which matches the weights in the empirical distribution up to a constant $c$. By the weighted exchangeability in Tibshirani et al. (2019), the randomly localized conformal prediction interval is finite-sample valid in the sense that $\mathrm{Pr}\left(Y_{2j} \in \mathrm{PI}_{\mathrm{L}}(\mathbf{X}_{2j})\right) = \mathbb{E}\left\{\mathrm{Pr}\left(V(\mathbf{X}_{2j}, Y_{2j}) \le \hat{q}_{\alpha, \mathrm{L}} \mid \tilde{\mathbf{X}}_{2j}\right)\right\} \ge 1 - \alpha$.

## 2.2 FROM PREDICTION INTERVALS TO $p$-VALUES

Motivated by the capability of (randomly) localized conformal prediction to capture local information, we invert these prediction intervals to construct the localized conformal $p$-value. Due to its simplicity, we follow Hore & Barber (2024) and take the randomization technique as in (2.2).

Let $H(\mathbf{x}, \mathbf{x}') = \frac{1}{h^d} K\left(\frac{\mathbf{x} - \mathbf{x}'}{h}\right)$ be a bi-variate kernel function with bandwidth $h$, where $K(\cdot)$ is a kernel density function and $d$ is the dimension of the feature vector $\mathbf{x}$ under consideration. Here $K(\cdot)$ can be taken as the Gaussian kernel, the box kernel or any other nonparametric kernel function as long as it is a symmetric density function. Define the localized conformal $p$-value for $(\mathbf{X}_{2j}, Y_{2j}) \in \mathcal{D}_2$ as

$$p_{\mathrm{L}, j} = \frac{\sum_{i \in \mathcal{C}} H(\mathbf{X}_{1i}, \tilde{\mathbf{X}}_{2j}) \mathbb{I}\{V_{2j} \le V_{1i}\} + \xi_j \cdot H(\mathbf{X}_{2j}, \tilde{\mathbf{X}}_{2j})}{\sum_{i \in \mathcal{C}} H(\mathbf{X}_{1i}, \tilde{\mathbf{X}}_{2j}) + H(\mathbf{X}_{2j}, \tilde{\mathbf{X}}_{2j})}, \qquad (2.4)$$

where $\xi_j \sim \mathrm{U}[0, 1]$ is an independent random variable and $\tilde{\mathbf{X}}_{2j}$ is randomly sampled from density $H(\mathbf{X}_{2j}, \cdot)$. The non-conformity score $V$ may depend on both $\mathbf{X}$ and $Y$ or only on $\mathbf{X}$, contingent on the specific problem. The $p_{\mathrm{L}, j}$ can be viewed as a localized counterpart of the conformal $p$-value investigated by Bates et al. (2023) by using weighted empirical distributions. To simplify terminology, we will use CP and LCP to refer to the unweighted conformal $p$-value and our localized conformal $p$-value, respectively in the rest of the article. Since we do not discuss prediction intervals, this should not lead to confusion.

## 2.3 BASIC PROPERTIES

In this section we state some basic properties of the localized conformal $p$-value. The first property is its finite-sample validity.

**Lemma 1** (Finite-sample validity). *For all $0 \le \alpha \le 1$, under the condition $P_1 = P_{2,j}$, the localized conformal $p$-value satisfies $\mathrm{Pr}(p_{\mathrm{L}, j} \le \alpha) \le \alpha$. Furthermore, if the score $V(\mathbf{X}, Y)$ has a continuous distribution, then $\mathrm{Pr}(p_{\mathrm{L}, j} \le \alpha) = \alpha$.*

This lemma is a direct corollary of the weighted exchangeability of $\{(\mathbf{X}_{2j}, Y_{2j})\} \cup \mathcal{C}$ given $\tilde{\mathbf{X}}_{2j}$. By leveraging this property, the LCP can be used for multiple testing to guarantee finite-sample FDR or FWER control.

By the nature of the local-weighting scheme, the LCP can adapt to potential covariate shifts. The following lemma is an analog of the robustness result in Hore & Barber (2024).

**Lemma 2** (Robustness against covariate shift). *Under the condition $P_{1, Y | \mathbf{X}} = P_{2,j, Y | \mathbf{X}}$, denote the covariate density ratio as $g(\mathbf{x}) := \frac{\mathrm{d}P_{2, \mathbf{X}}}{\mathrm{d}P_{1, \mathbf{X}}}(x)$. The LCP satisfies*

$$\mathrm{Pr}(p_{\mathrm{L}, j} \le \alpha) \le \alpha + \|f_{1, \mathbf{X}}\|_\infty \mathbb{E}_{\mathbf{X} \sim P_{H, \mathbf{X}}, \mathbf{U} \sim K(\cdot)} \left\{|g(\mathbf{X} + h\mathbf{U}) - g(\mathbf{X})|\right\},$$

*where the distribution $P_{H, \mathbf{X}}$ has a density function $f_{H, \mathbf{X}}(\mathbf{x}) = \mathbb{E}_{\mathbf{X} \sim P_{1, \mathbf{X}}}\{H(\mathbf{X}, \mathbf{x})\}$.*

This lemma gives a deviation bound of the distribution of LCP from uniform distribution under covariate shift. The excess term will be small with a small $h$ when the density ratio function $g$ satisfies some regularity conditions. For example, if $g$ is Lipchitz continuous, the excess term will

be of order $O(h)$ vanish as $h \to 0$. For another example, if we want to perform conditional testing on some fixed region $\mathcal{B} \subset \mathcal{X}$, we can take $g(x) = \mathbb{I}\{x \in \mathcal{B}\}/\Pr(\mathbf{X} \in \mathcal{B})$ with the shifted covariate distribution being $P_{1,\mathbf{X}}$ conditional on $\mathcal{B}$. The excess term is then dominated by $\Pr(d(\mathbf{X}, \partial\mathcal{B}) \le h)$, which is also an $O(h)$ term. This indicates the LCP is approximately valid conditional on any fixed region $\mathcal{B}$.

As an indirect power characterization, the next lemma studies the point-wise limit of the LCP function which is defined as

$$p_{\mathrm{L}}(\mathbf{x}, y) = \frac{\sum_{i \in \mathcal{C}} H(\mathbf{X}_{1i}, \tilde{\mathbf{X}})\mathbb{I}\{v \le V_{1i}\} + \xi \cdot H(\mathbf{x}, \tilde{\mathbf{X}})}{\sum_{i \in \mathcal{C}} H(\mathbf{X}_{1i}, \tilde{\mathbf{X}}) + H(\mathbf{x}, \tilde{\mathbf{X}})},$$

where $v = V(\mathbf{x}, y)$ and $\tilde{\mathbf{X}}$ is sampled from $H(\mathbf{x}, \cdot)$. With a score function $V$ chosen properly based on the specific problem, a larger score value $v$ will indicate stronger evidence against the pre-specified null hypothesis. Our defined $p$-value therefore reflects evidence against the null contained in a single data point. For the $p$-value to be powerful, the value of $p_{\mathrm{L}}(\mathbf{X}, Y)$ should be small if $(\mathbf{X}, Y)$ is sampled under the alternative. Therefore, for a fixed score value $v$ under the alternative, we can regard a $p$-value function to be asymptotically more powerful than another if its limit function takes smaller value at $v$.

We need some regularity conditions which are commonly used in nonparametric estimations.

**Assumption 1.** *The following conditions hold for $(\mathbf{X}, Y) \sim P_1$:*

- $V(\mathbf{X}, Y)$ *has a continuous distribution with bounded density;*

- *The conditional distribution of the score $V = V(\mathbf{X}, Y)$ satisfies*

$$\|F_{V|\mathbf{X}=\mathbf{x}}(v) - F_{V|\mathbf{X}=\mathbf{x}'}(v)\|_\infty \le L \cdot \|\mathbf{x} - \mathbf{x}'\|_2^\beta$$

  *for some constant $L > 0, 0 < \beta \le 1$. That is, the conditional distribution function $F_{V|\mathbf{X}=\mathbf{x}}$ varies smoothly with $\mathbf{x}$.*

- *The density function $f_{1,\mathbf{X}}(\mathbf{x})$ is continuous, and the conditional density function $f_1(y \mid \mathbf{x})$ is continuous in $\mathbf{x}$.*

**Lemma 3** (Asymptotic behavior). *Assume Assumption 1 holds and the split ratio $|\mathcal{C}|/|\mathcal{D}_1| = \gamma$ for some constant $\gamma > 0$, then the LCP function converges in probability*

$$|p_{\mathrm{L}}(\mathbf{x}, y) - (1 - F_{V|\mathbf{X}=\mathbf{x}}(v))| = O_p\left(\sqrt{h^{2\beta} + \frac{1}{nh^d}}\right)$$

*for any fixed $(\mathbf{x}, y)$, as $h \to 0, nh^d \to \infty$.*

By the weak law of large number, the unweighted CP function satisfies

$$p_{\mathrm{CP}}(\mathbf{x}, y) = \frac{\sum_{i \in \mathcal{C}} \mathbb{I}\{v \le V_{1i}\} + 1}{|\mathcal{C}| + 1} \xrightarrow{p} 1 - F_V(v),$$

where $F_V(\cdot)$ is the marginal distribution function of score $V$. If we take $h$ such that $h \to 0, nh^d \to \infty$, the LCP function converges to $1 - F_{V|\mathbf{X}=\mathbf{x}}(v)$ in probability. Comparing the power then amounts to comparing the value of $1 - F_{V|\mathbf{X}=\mathbf{x}}(v)$ and $1 - F(v)$. For a score value $v$ under the alternative, our conditional testing problem can generally ensure a uniformly small value of $1 - F_{V|\mathbf{X}=\mathbf{x}}(v)$ since the signal lies in the deviation of the conditional distribution. In contrast, the value of $1 - F_V(v)$ could be quite large for $(\mathbf{x}, y)$ pairs with a small conditional scale $V \mid \mathbf{X} = \mathbf{x}$, even if these pairs are sampled under the alternative. Although not uniformly more powerful, the LCP can identify signals in these regions of $\mathcal{X}$ with a small conditional scale $V \mid \mathbf{X} = \mathbf{x}$, which tends to be missed by the CP. This property further motivates us to apply the LCP on conditional testing problems.

## 3 APPLICATIONS ON CONDITIONAL TESTING

In this section, we apply the LCP on several conditional testing problems. We first provide a straight-forward application on sample selection problem in Section 3.1. Then in Section 3.2 we study the

conditional outlier detection problem and adopt the conditional calibration technique to achieve finite-sample FDR control. In Section 3.3 we introduce a novel conditional label screening problem and leverage the LCP to design a screening procedure with FWER control. Our application on the two-sample conditional distribution testing problem is deferred to Appendix C due to space limits.

## 3.1 WARM-UP: BALANCED DATA SELECTION

Consider the sample selection problem which was investigated by Jin & Candès (2023b) and Wu et al. (2023). The test sample $\mathcal{D}_2 = \{\mathbf{X}_{2j}\}_{j=1}^m$ is unlabeled. The goal is to select test samples satisfying a pre-specified rule $Y \in \mathcal{A}$. Therefore, whether to select the $j$th sample amounts to conducting the following hypothesis test:

$$\mathbb{H}_{0j} : Y_{2j} \notin \mathcal{A}, \quad \text{versus} \quad \mathbb{H}_{1j} : Y_{2j} \in \mathcal{A}.$$

In this case, the CP can be accordingly defined as

$$p_j = \frac{\sum_{i \in \mathcal{C}, Y_{1i} \notin \mathcal{A}} \mathbb{I}\{V_{2j} \leq V_{1i}\} + \xi_j}{|\{i \in \mathcal{C} : Y_{1i} \notin \mathcal{A}\}| + 1},$$

and if $p_j \leq \alpha$ we select $\mathbf{X}_{2j}$. Such methods directly controls the per selection error rate (PSER), say

$$\Pr(\mathbf{X}_{2j} \text{ selected} \mid Y_{2j} \notin \mathcal{A}) \leq \alpha.$$

Analogously, we can apply our proposed localized conformal $p$-value instead to achieve certain improvement in terms of conditional performance. The LCP is defined as

$$p_{\mathrm{rL},j} = \frac{\sum_{i \in \mathcal{C}, Y_{1i} \notin \mathcal{A}} H(\mathbf{X}_{1i}, \tilde{\mathbf{X}}_{2j}) \mathbb{I}\{V_{2j} \leq V_{1i}\} + \xi_j \cdot H(\mathbf{X}_{2j}, \tilde{\mathbf{X}}_{2j})}{\sum_{i \in \mathcal{C}, Y_{1i} \notin \mathcal{A}} H(\mathbf{X}_{1i}, \tilde{\mathbf{X}}_{2j}) + H(\mathbf{X}_{2j}, \tilde{\mathbf{X}}_{2j})},$$

where $V$ is a non-conformity score function depending on $\mathcal{A}$. We use $\delta_j = 1, 0$ to indicate selecting $\mathbf{X}_{2j}$ or not, where $\delta_j = \mathbb{I}\{p_{\mathrm{rL},j} \leq \alpha\}$. As a corollary of Lemmas 1-2, the following theorem provides the marginal and conditional properties of this simple selection rule.

**Theorem 1** (Finite-sample PSER control and conditional PSER bound). *Suppose $\{(\mathbf{X}_{1i}, Y_{1i})\}_{i=1}^n$ and $\{(\mathbf{X}_{1i}, Y_{1i})\}_{j=1}^m$ are exchangeable, the selection rule $\delta_j = \mathbb{I}\{p_{\mathrm{rL},j} \leq \alpha\}$ can ensure finite-sample marginal PSER control $\Pr(\delta_j = 1 \mid Y_{2j} \notin \mathcal{A}) \leq \alpha$. Moreover, the conditional PSER inflation bound is given by*

$$\Pr(\delta_j = 1 \mid Y_{2j} \notin \mathcal{A}, \mathbf{X}_{2j} \in \mathcal{B}) \leq \alpha + 2\|f_{1,\mathbf{X},\mathcal{A}}\|_\infty \frac{\Pr_{\mathbf{X} \sim P_{H,\mathbf{X},\mathcal{A}}, \mathbf{U} \sim K(\cdot)}(\|U\|_2 \geq h^{-1} d(\mathbf{X}, \partial\mathcal{B}))}{\Pr(\mathbf{X}_{2j} \in \mathcal{B} \mid Y_{2j} \notin \mathcal{A})},$$

*where $f_{1,\mathbf{X},\mathcal{A}}$ is the conditional density of $\mathbf{X}_{1i} \mid Y_{1i} \notin \mathcal{A}$, $P_{H,\mathbf{X},\mathcal{A}}$ has a density function $f_{H,\mathbf{X},\mathcal{A}}(\mathbf{x}) = \mathbb{E}_{\mathbf{X} \sim f_{1,\mathbf{X},\mathcal{A}}}\{H(\mathbf{X}, \mathbf{x})\}$ and $\partial\mathcal{B}$ is the boundary set of $\mathcal{B}$.*

The second result is obtained by taking the function $g(\mathbf{x}) = \mathbb{I}\{\mathbf{x} \in \mathcal{B}\}/\Pr(\mathbf{X}_{2j} \in \mathcal{B})$ in Lemma 2 and simplifying the deviation term. For general sets $\mathcal{B}$ of regular form (e.g., balls or hypercubes), the excess term will be small with a small $h$. This indicates that using the LCP for data selection can lead to a more balanced selection result since the PSER inflation for different sub-groups is bounded. For instance, by choosing an appropriate $\mathcal{B}$, we can expect that the burden of incorrect selection probability (PSER) will be more evenly distributed among different genders and races via our LCP. A related issue is addressed by Rava et al. (2021), which focuses on controlling group-wise error rates to mitigate imbalances, aligning closely with our objectives.

## 3.2 CONDITIONAL OUTLIER DETECTION

In the conditional outlier detection problem, the available data consists of clean data $\mathcal{D}_1$ and test data $\mathcal{D}_2$ with potential outliers. Both samples are labeled with observed responses. The inliers in $\mathcal{D}_2$ have the same conditional distribution $P_{2,j,Y|\mathbf{X}} = P_{1,Y|\mathbf{X}}$ with $\mathcal{D}_1$ while the outliers may have different conditional distributions from each other. Detecting conditional outliers can be formulated as the following multiple testing problem:

$$\mathbb{H}_{0j} : P_{2,j,Y|\mathbf{X}=\mathbf{x}} = P_{1,Y|\mathbf{X}=\mathbf{x}} \text{ for almost all } \mathbf{x}, \quad \text{versus} \quad \mathbb{H}_{1j} : \text{otherwise}, \tag{3.1}$$

where $(\mathbf{X}_{2j}, Y_{2j}) \in \mathcal{D}_2$ is an inlier if $\mathbb{H}_{0j}$ holds and outlier if $\mathbb{H}_{1j}$ holds. Our goal is to determine the detection set (or the rejection set equivalently) $\mathcal{R} \subseteq \{1, 2, \ldots, m\}$ based on the observed data $\mathcal{D}_1$ and $\mathcal{D}_2$. Denote $\mathcal{I}, \mathcal{O} \subseteq \{1, 2, \ldots, m\}$ as the index sets of inliers and outliers. The rejection set $\mathcal{R}$ should contain as many indices in $\mathcal{O}$ as possible while guaranteeing finite-sample false discovery rate (FDR) control

$$\mathrm{FDR} = \mathbb{E}(\mathrm{FDP}) = \mathbb{E}\left(\frac{|\mathcal{I} \cap \mathcal{R}|}{|\mathcal{R}| \vee 1}\right) \leq \alpha.$$

Bates et al. (2023) utilized the conformal $p$-value to test for marginal outliers by applying the BH procedure on conformal $p$-values computed on the test data. However, this is generally not effective when testing for conditional outliers. As discussed in the previous section, the classical CP targets on the joint distribution of $(\mathbf{X}, Y)$ rather than $Y \mid \mathbf{X}$, and thus cannot identify all information in the conditional distribution of score variables. In light of the capability of the LCP to capture deviations in conditional distributions, we can take it as a refinement to detect conditional outliers.

As proved by Bates et al. (2023), the unweighted conformal $p$-values based on the same calibration set is positive regression dependent on a subset (i.e., PRDS), under which the BH procedure can still guarantee finite-sample FDR control. This property, however, does not hold for the weighted conformal $p$-values. In order to achieve the finite-sample property, we adopt the conditional calibration technique (Fithian & Lei, 2022) to prune the rejection set output by the BH procedure.

To perform multiple testing with the LCP, we first train the non-conformity score function $V(\mathbf{x}, y)$ on $\mathcal{D}_{\mathcal{T}}$ and then compute scores $\{V_{1i}\}_{i \in \mathcal{C}}$ and $\{V_{2j}\}_{j=1}^m$ on $\mathcal{D}_{\mathcal{C}}$ and $\mathcal{D}_2$, respectively. After sampling $\tilde{\mathbf{X}}_{2j}$ for each $\mathbf{X}_{2j}$, the LCP's $\{p_{\mathrm{L},j}\}_{j=1}^m$ are computed as in Eq. (2.4). Define the auxiliary $p$-values as

$$p_{\mathrm{L},j}^{(l)} = \frac{\sum_{i \in \mathcal{C}} H(\mathbf{X}_{1i}, \tilde{\mathbf{X}}_{2j}) \mathbb{I}\{V_{2j} \leq V_{1i}\} + \xi_j \cdot H(\mathbf{X}_{2j}, \tilde{\mathbf{X}}_{2j}) \mathbb{I}\{V_{2j} \leq V_{2l}\}}{\sum_{i \in \mathcal{C}} H(\mathbf{X}_{1i}, \tilde{\mathbf{X}}_{2j}) + H(\mathbf{X}_{2j}, \tilde{\mathbf{X}}_{2j})}. \tag{3.2}$$

for $l \in \{1, 2, \ldots, m\} \setminus \{j\}$. Let $\widehat{\mathcal{R}}_{j \to 0}$ be the rejection set of the BH procedure applied on $\{p_{\mathrm{L},1}^{(j)}, \ldots, p_{\mathrm{L},j-1}^{(j)}, 0, p_{\mathrm{L},j+1}^{(j)}, \ldots, p_{\mathrm{L},m}^{(j)}\}$ and $\mathcal{R}^{\mathrm{init}} = \{j : p_{\mathrm{L},j} \leq \alpha |\widehat{\mathcal{R}}_{j \to 0}|/m\}$ be the initial rejection set. We determine the final rejection set by generating independent $\zeta_1, \ldots, \zeta_m \sim \mathrm{U}[0, 1]$ and pruning $\mathcal{R}^{\mathrm{init}}$ into

$$\mathcal{R} = \left\{j : p_{\mathrm{L},j} \leq \frac{\alpha |\widehat{\mathcal{R}}_{j \to 0}|}{m}, \zeta_j |\widehat{\mathcal{R}}_{j \to 0}| \leq r^*\right\}, \tag{3.3}$$

where $r^* = \max\left\{r : \sum_{j=1}^m \mathbb{I}\left\{p_{\mathrm{L},j} \leq \alpha |\widehat{\mathcal{R}}_{j \to 0}|/m, \zeta_j |\widehat{\mathcal{R}}_{j \to 0}| \leq r\right\} \geq r\right\}$.

The outlier detection procedure is summarized in Algorithm 1 in Appendix D.

The following theorem shows that our detection procedure can guarantee finite-sample FDR control if the distribution of the covariates does not change.

**Theorem 2** (Finite-sample FDR control). *Under the condition that $P_{1,\mathbf{X}} = P_{2,\mathbf{X}}$, the final output $\mathcal{R}$ given by Algorithm 1 ensures* $\mathrm{FDR} \leq \alpha$.

**Remark 1.** *Since the condition $P_{1,\mathbf{X}} = P_{2,\mathbf{X}}$ together with the null hypothesis $P_{1,Y|\mathbf{X}} = P_{2,j,Y|\mathbf{X}}$ lead to the equality of the joint distribution of $(\mathbf{X}, Y)$, our method is not an exact testing procedure for (3.1). Instead, it can be viewed as an extended marginal outlier detection approach with remarkably boosted power on conditional outliers. This aligns with our motivation, as the classical CP often fails to detect conditional outliers effectively.*

### 3.3 CONDITIONAL LABEL SCREENING

In the conditional label screening problem, the response variable is multivariate with $\mathbf{Y} = (Y_1, Y_2, \ldots, Y_S)$ for some random variable $S$. The first sample $\mathcal{D}_1$ is the labeled data while the second sample $\mathcal{D}_2$ is unlabeled. The response vectors are denoted as $\mathbf{Y}_{\ell i} = (Y_{\ell i,1}, \ldots, Y_{\ell i, S_{\ell i}})$ for $\ell = 1, 2$. Our goal is to determine whether each component of the unobserved response $\mathbf{Y}_{2j} = (Y_{2j,1}, \ldots, Y_{2j,S_{2j}})$ satisfies a pre-specified rule for each $\mathbf{X}_{2j} \in \mathcal{D}_2$. For each entry $s$, we define the screening rule as $Y_{2j,s} \in \mathcal{A}_s$ with pre-chosen sets $\mathcal{A}_s$. This can be formulated as a multiple testing problem for each test sample

$$\mathbb{H}_{0j,s} : Y_{2j,s} \notin \mathcal{A}_s, \quad \text{versus} \quad \mathbb{H}_{1j,s} : Y_{2j,s} \in \mathcal{A}_s, \quad 1 \leq s \leq S_{2j}. \tag{3.4}$$

For example, Mohri & Hashimoto (2024); Cherian et al. (2024) considered using conformal prediction techniques to improve the output factuality of large language models (LLM). Specifically, they transform the output of the LLM into a set of claims and aim to construct a filtered claim set that contains no false claims with high probability. Such application can be framed into our conditional label screening problem stated above. The covariate $\mathbf{X}_{2j}$ includes the input prompt $P_{2j}$, the output $R_{2j}$ of the LLM model and a claim vector $\mathbf{C}_{2j}$ of length $S_{2j}$ summarized by another language model. The response vector is a 0-1 vector with $Y_{2j,s} = 1$ or $0$ indicating whether the corresponding claim is correct or not. Here we can take $\mathcal{A}_s = \{1\}$ for every $1 \leq s \leq S_{2j}$ to screen out false claims.

To ensure reliability of the screening procedure, we seek to control the probability of failing to screen out any component of $\mathbf{Y}_{2j}$ that does not meet the rule. Let $\delta_{j,s} = 1$ or $0$ indicate whether $Y_{2j,s}$ is retained after screening. This can be rigorously formulated as

$$\Pr\left(\sum_{s=1}^{S_{2j}} \mathbb{I}\{Y_{2j,s} \notin \mathcal{A}_s, \delta_{j,s} = 1\} > 0\right) \leq \alpha, \tag{3.5}$$

say, controlling the FWER for (3.4). However, the marginal FWER might not be sufficient in real problems. We can use the localization technique to improved conditional validity while still guaranteeing marginal FWER control.

Since the test data $\mathcal{D}_2$ is unlabeled in the current problem, the non-conformity score function $V$ should depend only on the covariate $\mathbf{X}$. We can use the training data $\mathcal{D}_\mathcal{T}$ to estimate the probability $\Pr(Y_{2j,s} \in \mathcal{A}_s)$. This can be achieved by training classification models for each component of $Y$ if the length $S$ is a fixed constant or fitting a joint classification model that outputs a probability vector. In both scenarios, we can define the score vector

$$V(\mathbf{X}_{2j}) = (V_{2j,1}, V_{2j,2}, \ldots, V_{2j,S_{2j}})^\top,$$

where each $V_{2j,s}$ approximates $\Pr(Y_{2j,s} \in \mathcal{A}_s)$. By this definition, we should reject $\mathbb{H}_{0j,s}$ if $V_{2j,s}$ is large, and controlling the FWER amounts to examining the distribution of $\max\{V_{2j,s} : Y_{2j,s} \notin \mathcal{A}_s\}$. This motivates us to define the localized $p$-value

$$p_{\mathrm{L},j,s} = \frac{\sum_{i\in\mathcal{C}} H(\mathbf{X}_{1i}, \tilde{\mathbf{X}}_{2j})\mathbb{I}\{V_{2j,s} \leq \bar{V}_{1i}\} + \xi_j \cdot H(\mathbf{X}_{2j}, \tilde{\mathbf{X}}_{2j})}{\sum_{i\in\mathcal{C}} H(\mathbf{X}_{1i}, \tilde{\mathbf{X}}_{2j}) + H(\mathbf{X}_{2j}, \tilde{\mathbf{X}}_{2j})}, \tag{3.6}$$

where $\bar{V}_{1i} = \max\{V_{1i,s} : Y_{1i,s} \notin \mathcal{A}_s\}$. We can show that the $p_{\mathrm{L},j,s}$'s satisfy the group super-uniform property, i.e.,

$$\Pr\left(\bigcup_{Y_{2j,s}\notin\mathcal{A}_s} \{p_{\mathrm{L},j,s} \leq \alpha\}\right) \leq \alpha,$$

and thus we can screen out components of $\mathbf{Y}_{2j}$ with $p_{\mathrm{L},j,s} \leq \alpha$. We summarize our label screening procedure in Algorithm 2 in Appendix D.

We have the following result.

**Theorem 3** (Finite-sample FWER control). *Suppose $\{(\mathbf{X}_{1i}, \mathbf{Y}_{1i}, S_{1i})\}_{i\in\mathcal{C}} \cup \{(\mathbf{X}_{2j}, \mathbf{Y}_{2j}, S_{2j})\}_{j=1}^m$ are exchangeable, then the label screening procedure given by Algorithm 2 ensures finite-sample FWER control*

$$\mathrm{FWER} = \Pr\left(\sum_{s=1}^{S_{2j}} \mathbb{I}\{Y_{2j,s} \notin \mathcal{A}_s, p_{\mathrm{L},j,s} \leq \alpha\} > 0\right) \leq \alpha.$$

Regarding the conditional property, we can also establish the finite-sample conditional FWER deviation bound for our procedure.

**Theorem 4** (Conditional FWER bound). *Suppose the assumptions in Theorem 3 hold. For any fixed set $\mathcal{B} \subset \mathcal{X}$ with $\Pr(\mathbf{X}_{2j} \in \mathcal{B}) > 0$, the conditional FWER has the following bound*

$$\Pr\left(\sum_{s=1}^{S_{2j}} \mathbb{I}\{Y_{2j,s} \notin \mathcal{A}_s, p_{\mathrm{L},j,s} \leq \alpha\} > 0 \;\Big|\; \mathbf{X}_{2j} \in \mathcal{B}\right)$$

$$\leq \alpha + 2\|f_{1,\mathbf{X}}\|_\infty \frac{\Pr_{\mathbf{X}\sim P_{H,\mathbf{X}}, \mathbf{U}\sim K(\cdot)}(\|U\|_2 \geq h^{-1}d(\mathbf{X}, \partial\mathcal{B}))}{\Pr(\mathbf{X}_{2j} \in \mathcal{B})},$$

*where $\partial\mathcal{B}$ is the boundary of set $\mathcal{B}$.*

In this theorem, the inflation bound is similar to that in the second result of Theorem 1. By a similar rationale, this demonstrate the advantage of our method to mitigate conditional error rate inflation.

## 4 EXPERIMENTS AND EVALUATION

We provide synthetic and real-data experiment results to show the validity and efficiency of our proposed methods for two main applications, respectively in Sections 4.1-4.2. More comprehensive experiments for all our applications on conditional testing are detailed in Appendix E.

### 4.1 SYNTHETIC-DATA RESULTS FOR CONDITIONAL OUTLIER DETECTION

**Data Description.** For the conditional outlier detection problem, we consider a heterogeneous linear regression model with label $Y$ in which the data are generated as follows:

Scenario A1: the covariate vector consists of $\mathbf{X} = (X_1, \ldots, X_{d^*-1})^\top \in \mathbb{R}^{d^*-1}$ with $d^* = 10$ and an additional time feature $t \in \mathbb{R}$. The model is $Y = \mathbf{X}\boldsymbol{\beta} + (3 + 2 \cdot \sin(2\pi \cdot t)) \cdot \varepsilon$, with $X_1, \ldots, X_{d^*-1} \sim U[-1, 1], t \sim U[0, 1]$ and $\varepsilon \sim N(0, 1)$ independently. The coefficient vector is $\boldsymbol{\beta} = (0.5, -0.5, 0.5, -0.5, 0.5, 0, 0, 0, 0)$. The test data contains $10\%$ outliers following the model $Y = \mathbf{X}\boldsymbol{\beta} + (3 + 2 \cdot \sin(2\pi \cdot t)) \cdot \varepsilon + r(t) \cdot \xi$, where $r(t) = 3 \cdot (3 + 1.5 \sin(2\pi \cdot t))$ and $\Pr(\xi = \pm 1) = 1/2$. We also consider a scenario without label $Y$ in Appendix E.1 (Scenario B1).

**Benchmarks.** We abbreviate our method as LCP-od (outlier detection) and compare it with a benchmark method abbreviated as CP. The CP method is implemented by computing the unweighted conformal $p$-values as defined in Bates et al. (2023) and then applying the BH procedure. For the LCP-od and CP methods, we use two score functions: CQR scores and absolute residual scores, and apply various regression or classification algorithms, which are detailed in Appendix E.1.

**Results.** The average FDP above nominal level and power across 500 replications for Scenario A1 are shown in Figure 1. It can be seen that both the LCP-od method and the CP method accurately control the FDR around the nominal level, which validates their finite-sample FDR control property. In terms of power, the LCP-od method demonstrates higher power than the CP benchmark with both two score functions when the sample size $n$ is relatively large. Compared with the absolute residual score, the CQR score leads to relatively higher power due to its adaptivity to the conditional distribution of $Y \mid \mathbf{X}$. However, the power gain is still significant after applying the LCP-od method, illustrating the advantage of local weighting. Moreover, the power of the LCP-od method grows significantly with the sample size $n$, while the power of the CP method only rises slightly. As discussed in the Section 2.3, the unweighted CP cannot identify those outliers with small conditional variance of the score. This explains why the power of the CP method remains "stuck" around the same value. In contrast, the power of the LCP-od method approaches 1 as the sample size increases, demonstrating the full identifiability of the LCP-od method in testing for conditional outliers.

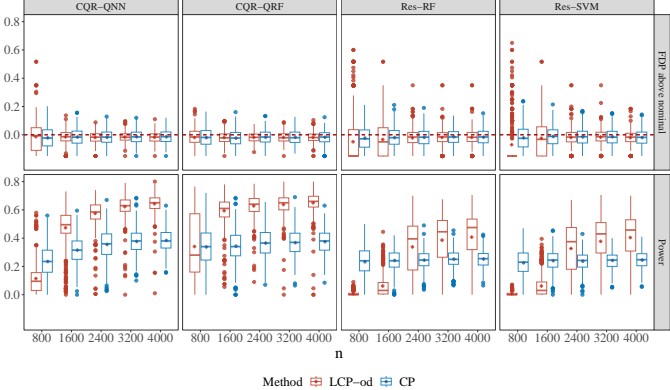

Figure 1: FDP above the nominal level and power under different clean sample sizes $n$ for Scenario A1. The test sample size is fixed at $m = 2000$ and the nominal level is fixed at $\alpha = 0.15$.

## 4.2 CONDITIONAL LABEL SCREENING ON HEALTH DATA

**Dataset.** We utilize the health indicators dataset from Kaggle (Kaggle, 2021) to demonstrate the performance of our conditional label screening method. The dataset comprises $n = 70,692$ samples and 22 variables, including demographic attributes (e.g., sex, age, BMI), lifestyle-related features, as well as several binary health indicators. These indicators capture whether an individual has conditions such as diabetes, coronary heart disease, or has experienced a myocardial infarction, among others. Accordingly, we define the response variable $\mathbf{Y}$ as these disease-related binary labels, with the goal of predicting the risk of these conditions in test data. More specifically, the response vector $\mathbf{Y} = (Y_1, Y_2, Y_3) \in \{0,1\}^3$ is set as follows:

- $Y_1$: Indicates whether an individual has diabetes.
- $Y_2$: Indicates whether an individual has coronary heart disease or myocardial infarction.
- $Y_3$: Indicates whether an individual has experienced a stroke.

Consequently, the label screening targets are $Y_s \in \mathcal{A}_s = \{1\}$ for $s = 1, 2, 3$. We fix the size of the labeled data $\mathcal{D}_1$ at $n = 500$ and the test data $\mathcal{D}_2$ at $m = 2,000$, randomly sampling them from the original dataset without replacement in each replication. We fix the nominal level at $\alpha = 0.05$, and apply two different algorithms to compute the score vectors: linear logistic regression (LL) and random forest (RF).

**Benchmarks and performance measures.** We compare our conditional label screening method via the LCP (abbreviated as LCP-ls) with the thresholding procedure without weighting (abbreviated as THR). The THR method is performed by only replacing the LCP by the classical unweighted CP. We calculate the empirical marginal FWER and conditional FWER (conditional on $\mathbf{X} \in \mathcal{B}$ for a specific set $\mathcal{B}$) for the screening procedure via LCP-ls and the simple thresholding rule, respectively, where $\mathcal{B} = \{\mathbf{X} : \text{sex} = \text{female and BMI} > 30\}$. The precise definitions of two measures are detailed in Section 3.3.

**Results.** We show the average of measures for all $m$ test samples across 500 repetitions. The results are reported in Table 1. We observe that both benchmarks can ensure valid marginal FWER control, while only the label screening procedure via the LCP-ls is capable of controlling both conditional and marginal FWER simultaneously.

Table 1: Empirical conditional FWER (cFWER) and marginal FWER (mFWER) under nominal level $\alpha = 0.05$ for health indicator dataset across 500 replications by using LL and RF algorithms, respectively. The bracket contains the standard error.

| Method | cFWER$_{\text{LL}}$ | cFWER$_{\text{RF}}$ | mFWER$_{\text{LL}}$ | mFWER$_{\text{RF}}$ |
|--------|---------------------|---------------------|---------------------|---------------------|
| LCP-ls | 0.0329 (0.018) | 0.0372 (0.018) | 0.0487 (0.009) | 0.0507 (0.009) |
| THR | 0.0740 (0.028) | 0.0887 (0.029) | 0.0500 (0.011) | 0.0538 (0.012) |

## 5 CONCLUDING REMARKS

We conclude the paper with two remarks. Firstly, the localized conformal prediction is efficient in capturing local or conditional information, but this comes at the cost of a much smaller effective sample size. In our simulation experiments, the power of localized methods grows significantly as the sample size increases in many scenarios. However, when the sample size is small, our proposed methods often exhibit lower power compared to other methods. Therefore, it would be beneficial to increase the effective sample size by utilizing additional data or modifying the definition of the LCP.

Secondly, we define the LCP with a random sampling step for each test point to achieve finite-sample validity. However, the random sampling step introduces external randomness, which degrades the stability of the related methods. Additionally, the LCP is defined by computing the similarity between covariates of the calibration data and the sampled covariate $\tilde{\mathbf{X}}$ instead of the test point $\mathbf{X}$ itself. Although these two issues are equivalent asymptotically, they generally differ in finite-samples. This makes the LCP not effective enough in characterizing local information. A potential solution is to use a different definition for the localized conformal $p$-value to ensure finite-sample validity without randomization, which warrants further consideration.

## REPRODUCIBILITY STATEMENT

Code for implementing our methods and reproducing the experiments and figures in our paper is available at https://github.com/lulin2023/LCP-testing. For details of our implements, please see the pseudo-codes for our all proposed algorithms in Appendix D and the implementation details in Appendix E. Proofs of all theoretical results are included in Appendix F.

## ACKNOWLEDGMENTS AND DISCLOSURE OF FUNDING

We thank anonymous area chair and reviewers for their helpful comments. This research was supported by the National Key R&D Program of China (Grant Nos. 2022YFA1003703, 2022YFA1003800), and the National Natural Science Foundation of China (Grant Nos. 123B2010, 11925106, 12231011, 12326325).

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

# Supplementary Material for "Conditional Testing based on Localized Conformal $p$-values"

This supplementary material contains:

- Preliminary terms for self-containment (Appendix A).
- Additional related works (Appendix B).
- An application on the two-sample conditional distribution test (Appendix C).
- Additional details of our algorithms (Appendix D).
- Additional experiments and implementation details (Appendix E).
- Proofs of all the theoretical results (Appendix F).

## A  PRELIMINARY TERMS FOR SELF-CONTAINMENT

Here, we list the preliminary terms we use in the paper for the sake of clarity and self-containment.

- FDR (Benjamini & Hochberg, 1995), false discovery rate, a widely-adopted error rate notion in the field of multiple testing, is defined as the expected proportion of incorrectly rejected null hypotheses as follows:

$$\mathrm{FDR} = \mathbb{E}\left[\frac{|\mathcal{H}_0 \cap \mathcal{R}|}{|\mathcal{R}| \vee 1}\right],$$

  where $\mathcal{H}_0$ is the unknown set of true null hypotheses, $\mathcal{R}$ represents the set of rejected null hypotheses and then $\mathcal{H}_0 \cap \mathcal{R}$ is the set of false discoveries.

- FWER, family-wise error rate, is defined as the probability of falsely rejecting more or equal to one hypothesis testing:

$$\mathrm{FWER} = \Pr(V > 0),$$

  where $V$ is the number of false discoveries (or rejections).

- PCER (PSER), per comparison (selection) error rate, is defined as follows:

$$\mathrm{PCER} = \mathbb{E}(V)/n,$$

  where $n$ is the total number of hypotheses.

## B  ADDITIONAL RELATED WORKS

**Outlier detection.** Classical outlier detection methods include multivariate outlier detection methods that use model assumptions to identify outliers (Riani et al., 2009; Cerioli, 2010), and machine learning algorithms (Liu et al., 2008; Erfani et al., 2016). Most existing works on conditional outlier detection fall into these two categories (Song et al., 2007; Catterson et al., 2010; Hong & Hauskrecht, 2015). Recent studies have used the conformal inference framework to test for outliers with advanced machine learning models (Bates et al., 2023; Marandon et al., 2024; Liang et al., 2024). Although these methods ensure finite-sample FDR control, they focus primarily on marginal cases and do not address outlier detection in conditional distributions.

**Data selection and subsampling.** Our introduced label screening problem is similar to label-based data selection (Jin & Candès, 2023a;b; Rava et al., 2021; Huo et al., 2024) and subsampling problems (Wu et al., 2023). These works consider a semi-supervised setting where the selection or sampling rule is based on the value of the response variable. The key difference in our problem is that we focus on screening within the multivariate response vector for each data point, rather than selecting different data points from the unlabeled dataset. Additionally, existing works do not account for the conditional properties of their methods, which is the main concern of our paper.

**Two-sample conditional distribution test.** Most existing works focus on testing the equality of two conditional moments (Hall & Hart, 1990; Dette & Neumeyer, 2003), which is less stringent than

testing for equality of distributions. A notable contribution is Hu & Lei (2023), which proposed a U-statistic based on the conformal inference framework by aggregating conformal $p$-values. Chen & Lei (2024) improved this method by de-biasing the functions estimated using machine learning algorithms. Our porposed test statistic, a kernel-weighted U-statistic, is closely related to Hu & Lei (2023) by aggregating localized conformal $p$-values instead.

## C  APPLICATION ON TWO-SAMPLE CONDITIONAL DISTRIBUTION TEST

In the two-sample conditional testing problem, the second sample $\mathcal{D}_2$ is labeled and i.i.d. following a potentially different distribution $(\mathbf{X}_{2j}, Y_{2j}) \sim P_{2,\mathbf{X}} \times P_{2,Y|\mathbf{X}}$ from the first sample $\mathcal{D}_1$. The conditional distribution test can be formulated as

$$\mathbb{H}_0 : P_{1,Y|\mathbf{X}=\mathbf{x}} = P_{2,Y|\mathbf{X}=\mathbf{x}} \text{ for almost all } \mathbf{x}, \quad \text{versus} \quad \mathbb{H}_1 : \text{ otherwise.} \tag{C.1}$$

Hu & Lei (2023) proposed a two-sample conditional distribution test based on the conformal inference framework. To be specific, they first split both samples as $\mathcal{D}_1 = \mathcal{D}_{\mathcal{T}_1} \cup \mathcal{D}_{\mathcal{C}_1}$ and $\mathcal{D}_2 = \mathcal{D}_{\mathcal{T}_2} \cup \mathcal{D}_{\mathcal{C}_2}$ with $|\mathcal{D}_{\mathcal{C}_1}| = |\mathcal{D}_{\mathcal{C}_2}| = n_1, |\mathcal{D}_{\mathcal{T}_1}| = |\mathcal{D}_{\mathcal{T}_2}| = n_2$. For notation convenience, we perform an equal-sized sample splitting with $n_1 = n_2$ and let $\mathcal{C}_1 = \mathcal{C}_2 = \{1, \ldots, n_1\}$ in this section. Thereafter, the score function and density ratio estimator

$$V(\mathbf{x}, y) = \frac{\widehat{f_1(y \mid \mathbf{x})}}{f_2(y \mid \mathbf{x})}, \quad \widehat{g}(\mathbf{x}) = \frac{\widehat{f_{2,\mathbf{X}}(\mathbf{x})}}{f_{1,\mathbf{X}}(\mathbf{x})}$$

are trained on $\mathcal{D}_{\mathcal{T}_1} \cup \mathcal{D}_{\mathcal{T}_2}$ by fitting classification models to distinguish $\mathcal{D}_{\mathcal{T}_1}$ and $\mathcal{D}_{\mathcal{T}_2}$. After computing scores $\{V_{1i}\}_{i \in \mathcal{C}_1}$ and $\{V_{2j}\}_{j \in \mathcal{C}_2}$, the weighted conformal $p$-values are obtained as

$$\widehat{p}_j = \frac{n_1^{-1} \sum_{i \in \mathcal{C}_1} \widehat{g}(\mathbf{X}_{1i}) \widehat{D}_{ij}^*}{n_1^{-1} \sum_{i \in \mathcal{C}_1} \widehat{g}(\mathbf{X}_{1i})}, \quad j \in \mathcal{C}_2.$$

The test statistic $\widehat{T} = \frac{\frac{1}{2} - \frac{1}{n_1} \sum_{j \in \mathcal{C}_2} \widehat{p}_j}{\widehat{\sigma}}$, is constructed by averaging these $p$-values, where $\widehat{D}_{ij}^* = \mathbb{I}\{V_{2j} < V_{1i}\} + \xi_j \mathbb{I}\{V_{2j} = V_{1i}\}$ and $\widehat{\sigma}^2$ is the variance estimator. Under the null hypothesis, they proved that $\sqrt{n_1}\widehat{T}$ is asymptotically normally distributed and the rejection region is given by $\sqrt{n_1}\widehat{T} > \Phi^{-1}(1 - \alpha)$.

We extend the above strategy by aggregating the localized conformal $p$-values computed on the second sample. Since $p$-values are no longer necessary to be finite-sample valid in this case, here we use a simplified variant of the localized conformal $p$-value without randomization

$$p_{\mathrm{SL},j} = \frac{\sum_{i \in \mathcal{C}} H(\mathbf{X}_{1i}, \mathbf{X}_{2j}) \mathbb{I}\{V_{2j} \leq V_{1i}\} + \xi_j \cdot H(\mathbf{X}_{2j}, \mathbf{X}_{2j})}{\sum_{i \in \mathcal{C}} H(\mathbf{X}_{1i}, \mathbf{X}_{2j}) + H(\mathbf{X}_{2j}, \mathbf{X}_{2j})}, \tag{C.2}$$

and our proposed test statistic is defined as

$$\widehat{T}_w = \frac{\frac{1}{n_1} \sum_{i=1}^{n_1} \frac{1}{n_1} \sum_{j=1}^{n_1} H(\mathbf{X}_{1i}, \mathbf{X}_{2j}) \widehat{D}_{ij}}{\widehat{\sigma}_w}, \tag{C.3}$$

where $\widehat{D}_{ij} = 1/2 - \mathbb{I}\{V_{2j} < V_{1i}\} - \xi_j \mathbb{I}\{V_{2j} = V_{1i}\}$ and $\widehat{\sigma}_w^2$ is the variance estimator. This test statistic is constructed by averaging unnormalized LCP's in Eq. (C.2) and performing standardization. From a different perspective, it is also related to the classical U-statistic for model checking problems (Zheng, 1996; Gao & Gijbels, 2008) in which the $\widehat{D}_{ij}$'s are replaced by the residuals. Inherited from the nice property of conformal techniques, our method enjoys model-agnostic features and allows us to employ state-of-the-art algorithms to construct efficient score function $V$ that is able to better measure the discrepancy between two conditional distributions.

We need the following technical assumptions to establish the asymptotic normality of $\widehat{T}_w$. We also make the same regularity assumptions to those in Hu & Lei (2023) without further declaration in theorems, which guarantee identifiability of the problem.

**Assumption 2.** *Let* $V^*(\mathbf{x}, y) = f_2(y|\mathbf{x})/f_1(y|\mathbf{x})$ *be the true conditional density ratio. Denote the oracle statistics as* $V_{1i}^* = V^*(\mathbf{X}_{1i}, Y_{1i}), V_{2j}^* = V^*(\mathbf{X}_{2j}, Y_{2j})$ *and* $D_{ij} = 1/2 - \mathbb{I}\{V_{1i}^* < V_{2j}^*\} - \xi_j \mathbb{I}\{V_{1i}^* = V_{2j}^*\}$. *Suppose that*

- $V_{2j}^*$ has a continuous distribution;

- $\mathbb{E}\{H(\mathbf{X}_{1i}, \mathbf{X}_{2j})(\widehat{D}_{ij} - D_{ij})\} = o_p(1/\sqrt{n_1})$.

Recall that we use classification to construct the score function $V(\mathbf{x}, y)$ to approximate the true conditional density ratio $V^*(\mathbf{x}, y)$ in this problem. This assumption requires the approximation error is sufficiently small after local-weighting. A similar assumption has been made in Hu & Lei (2023) with $H(\mathbf{X}_{1i}, \mathbf{X}_{2j})$ replaced by $\widehat{g}(\mathbf{X}_{1i})$. These two assumptions are introduced both to ensure a vanishing asymptotic bias and achieve asymptotic normality in the presence of covariate shift.

**Theorem 5** (Asymptotic normality). *Under the null hypothesis, suppose Assumption 1 holds for $P_1, P_2$ and $h \to 0, n_1 h^d \to \infty$. Further assume $P_{1,\mathbf{X}} = P_{2,\mathbf{X}}$ or Assumption 2 holds. Then the test statistic (C.3) is asymptotically normally distributed*

$$\widehat{T}_w = \frac{\frac{1}{n_1}\sum_{i=1}^{n_1}\frac{1}{n_1}\sum_{j=1}^{n_1}H(\mathbf{X}_{1i}, \mathbf{X}_{2j})\widehat{D}_{ij}}{\widehat{\sigma}_w} \xrightarrow{d} Z \sim N(0,1),$$

*where the variance estimator is given by*

$$\widehat{\sigma}_w^2 = \frac{1}{n_1^2}\sum_{i=1}^{n_1}\left\{\frac{1}{n_1}\sum_{j=1}^{n_1}H(\mathbf{X}_{1i}, \mathbf{X}_{2j})\widehat{D}_{ij}\right\}^2 + \frac{1}{n_1^2}\sum_{j=1}^{n_1}\left\{\frac{1}{n_1}\sum_{i=1}^{n_1}H(\mathbf{X}_{1i}, \mathbf{X}_{2j})\widehat{D}_{ij}\right\}^2$$

$$- \frac{1}{n_1^4}\sum_{i=1}^{n_1}\sum_{j=1}^{n_1}H(\mathbf{X}_{1i}, \mathbf{X}_{2j})^2\widehat{D}_{ij}^2 - \frac{2}{n_1}\left\{\frac{1}{n_1^2}\sum_{i=1}^{n_1}\sum_{j=1}^{n_1}H(\mathbf{X}_{1i}, \mathbf{X}_{2j})\widehat{D}_{ij}\right\}^2.$$

Note that the convergence of the score function is only needed under covariate shift settings. Given the asymptotic normality property, we can construct our testing procedure as summarized in Algorithm 3.

**Theorem 6** (Behavior under local alternatives). *Suppose Assumption 1 holds for $P_1, P_2$, Assumption 2 holds, and $n_1\widehat{\sigma}_w^2 \xrightarrow{p} \sigma_w^2 > 0$. Then we have*

$$\widehat{T}_w = \frac{\sqrt{n_1}\delta_w}{4\sigma_w}(1 + o_p(1)) + Z + o_p(1),$$

*where $\delta_w = \mathbb{E}_{\mathbf{X}\sim P_{2,\mathbf{X}}, Y, Y' \sim P_{2,Y|\mathbf{X}}}\{f_{1,\mathbf{X}}(\mathbf{X})|V^*(\mathbf{X}, Y) - V^*(\mathbf{X}, Y')|\}$ and $Z \sim N(0,1)$.*

This theorem is similar to Proposition 1 in Hu & Lei (2023). The main difference lies in the form of "signal strength" $\delta_w$, which is $\delta = \mathbb{E}_{(\mathbf{X},Y),(\mathbf{X}',Y')\sim P_2}\{|V^*(\mathbf{X}, Y) - V^*(\mathbf{X}', Y')|\}$ in their article. In comparison, the difference term in $\delta_w$ is based on the same covariate value and therefore better captures the deviation in conditional distribution. This reflects the superiority of our proposed test statistic in the two-sample conditional testing problem.

# D ADDITIONAL DETAILS OF OUR ALGORITHMS

Figure 2 displays our workflow. We also provide the detailed pseudo-codes for our three algorithms in Algorithm 1-Algorithm 3 below for the problems of conditional outlier detection in Section 3.2, conditional label screening in Section 3.3, and two-sample conditional distribution test in Appendix C, respectively.

# E ADDITIONAL EXPERIMENTS RESULTS AND IMPLEMENTATION DETAILS

In this section, we provide implementation details and additional experiments results for our three conditional testing applications, including: (1) more synthetic and real-data results for conditional outlier detection in Appendix E.1 and E.4; (2) synthetic data results for conditional label screening in Appendix E.2; and (3) synthetic and real-data results for two-sample conditional distribution test in Appendix E.3 and E.5.

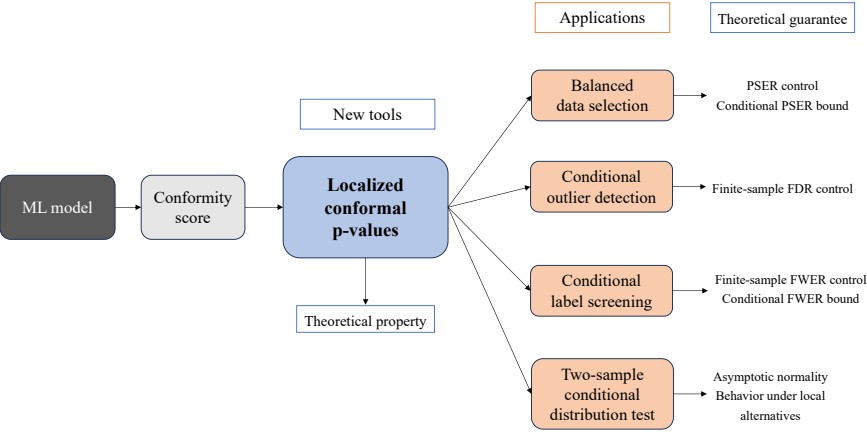

Figure 2: A flow chart of our work.

---

**Algorithm 1** Conditional Outliers Detection via the LCP

**Input:** Clean data $\mathcal{D}_1 = \{(\mathbf{X}_{1i}, Y_{1i})\}_{i=1}^n$ and test data $\mathcal{D}_2 = \{(\mathbf{X}_{2j}, Y_{2j})\}_{j=1}^m$; FDR target level $\alpha \in (0,1)$; Kernel function $H(\cdot, \cdot)$

1: Randomly split $\mathcal{D}_1$ into $\mathcal{D}_\mathcal{T} \cup \mathcal{D}_\mathcal{C}$, train the non-conformity score function $V(\mathbf{x}, y)$ on $\mathcal{D}_\mathcal{T}$ and compute score values $\{V_{1i}\}_{i \in \mathcal{C}}$ and $\{V_{2j}\}_{j=1}^m$ on $\mathcal{D}_\mathcal{C}$ and $\mathcal{D}_2$

    – Outlier detection –

2: **for** $j = 1, \ldots, m$ **do**

3:    Sample $\tilde{\mathbf{X}}_{2j}$ from density $H(\mathbf{X}_{2j}, \cdot)$, sample $\xi_j \sim \mathrm{U}[0,1]$ and construct the LCP $p_{\mathrm{L},j}$ as in (2.4)

4:    Compute auxiliary $p$-values $p_{\mathrm{L},j}^{(\ell)}$ as in (3.2)

5:    (BH procedure) Compute $r_j^* = \max\left\{ r : 1 + \sum_{\ell \neq j} \mathbb{I}\{p_{\mathrm{L},j}^{(\ell)} \leq \alpha r/m\} \geq r \right\}$

6:    Compute $\widehat{\mathcal{R}}_{j \to 0} = \{j\} \cup \{\ell \neq j : p_{\mathrm{L},j}^{(\ell)} \leq \alpha r_j^*/m\}$

7: **end for**

8: Compute the first-step rejection set $\mathcal{R}^{\mathrm{init}} = \{j : p_{\mathrm{L},j} \leq \alpha |\widehat{\mathcal{R}}_{j \to 0}|/m\}$

9: Prune the initial set $\mathcal{R}^{\mathrm{init}}$ and obtain the final rejection set $\mathcal{R}$ as in (3.3)

**Output:** Detected conditional outlier set $\mathcal{R}$

---

**Algorithm 2** Conditional Label Screening via the LCP

**Input:** Labeled data $\mathcal{D}_1 = \{(\mathbf{X}_{1i}, Y_{1i})\}_{i=1}^n$ and test data $\mathcal{D}_2 = \{\mathbf{X}_{2j}\}_{j=1}^m$; FWER target level $\alpha \in (0,1)$; Kernel function $H(\cdot, \cdot)$

1: Randomly split $\mathcal{D}_1$ into $\mathcal{D}_\mathcal{T} \cup \mathcal{D}_\mathcal{C}$ and train the non-conformity score function, compute score vectors $\{V(\mathbf{X}_{1i})\}_{i \in \mathcal{C}}$, $\{V(\mathbf{X}_{2j})\}_{j=1}^m$ on $\mathcal{D}_\mathcal{C}$, $\mathcal{D}_2$ and summary scores $\{\bar{V}_{1i}\}_{i \in \mathcal{C}}$

    – Screening –

2: **for** $j = 1, \ldots, m$ **do**

3:    Sample $\tilde{\mathbf{X}}_{2j}$ from density $H(\mathbf{X}_{2j}, \cdot)$ and $\xi_j \sim \mathrm{U}[0,1]$

4:    **for** $s = 1, \ldots, S_{2j}$ **do**

5:        Construct the LCP $p_{\mathrm{L},j,s}$ as in (3.6)

6:        Determine the screening decision $\delta_{j,s} = \mathbb{I}\{p_{\mathrm{L},j,s} \leq \alpha\}$

7:    **end for**

8: **end for**

**Output:** Screening decision vectors $\boldsymbol{\delta}_j = (\delta_{j,1}, \ldots, \delta_{j,s})^\top$ for each $1 \leq j \leq m$

---

---

**Algorithm 3** Two-sample conditional distribution test via aggregation of simplified LCP's

---

**Input:** Two samples $\mathcal{D}_1$ and $\mathcal{D}_2$; density ratio estimation subroutines $\mathcal{A}_1, \mathcal{A}_2$, kernel density $H(\cdot, \cdot)$, the nominal type I error level $\alpha \in (0, 1)$
1: Randomly split the two samples as $\mathcal{D}_1 = \mathcal{D}_{\mathcal{T}_1} \cup \mathcal{D}_{\mathcal{C}_1}$ and $\mathcal{D}_2 = \mathcal{D}_{\mathcal{T}_2} \cup \mathcal{D}_{\mathcal{C}_2}$
2: $\widehat{g}(\cdot) = \mathcal{A}_1[\{\mathbf{X}_{1i}, i \in \mathcal{T}_1, \mathbf{X}_{2j}, j \in \mathcal{T}_2\}]$
3: $v(\cdot, \cdot) = \mathcal{A}_2[\{(\mathbf{X}_{1i}, Y_{1i}), i \in \mathcal{T}_1, (\mathbf{X}_{2j}, Y_{2j}), j \in \mathcal{T}_2\}]$
4: **for** $j \in \mathcal{C}_2$ **do**
5:     Sample $\xi_j \sim \mathrm{U}[0, 1]$, independently
6:     Calculate $\widehat{D}_{ij} = 1/2 - \mathbb{I}\{V_{1i} < V_{2j}\} - \xi_j \mathbb{I}\{V_{1i} = V_{2j}\}$
7:     Calculate the kernel weights $H(\mathbf{X}_{1i}, \mathbf{X}_{2j}), i \in \mathcal{C}_1$
8: **end for**
9: Calculate the variance estimator $\widehat{\sigma}_\omega^2$ and obtain the weighted statistic $\widehat{T}_\omega$ as in (C.3)
10: Reject $H_0$ if $\Phi(\widehat{T}_\omega) \geq 1 - \alpha$
**Output:** The decision for two-sample conditional distribution test

---

### E.1 ADDITIONAL SYNTHETIC-DATA RESULTS FOR CONDITIONAL OUTLIER DETECTION

**Implementation details.** Since our goal is to test for outliers conditional on $t$ or $\boldsymbol{s}$ in Scenarios A1 and B1, we only use these variables when computing the weights. For both scenarios, the kernel function of the LCP-od method is taken as the Gaussian kernel $H(x, x') = (2\pi h^2)^{-d/2} \exp\{-\|x - x'\|_2^2/(2h^2)\}$ with bandwidth $h = (n/2)^{-1/(d+2)}$ for $d = 1, 2$ in Scenarios A1 and B1. This corresponds to the optimal convergence rate for $\beta = 1$. For Scenario A1, we consider two kinds of score functions: the CQR score and the absolute residual of different regression algorithms. Similar to Romano et al. (2019), for the CQR score, we consider using quantile neural networks (CQR-QNN) and quantile random forests (CQR-QRF). For the absolute residual score, we take two regression algorithms: random forest (Res-RF), and support vector machine (Res-SVM). For Scenario B1, we use one-class classifiers for both the LCP-od and CP methods. We take three kinds of one-class classification algorithms: the isolation forest (IOF), $k$-Nearest-Neighbor ($k$-NN) with $k = 5$ and one-class support vector machine (one-class SVM).

We consider either fixing $\alpha = 0.1$ or $0.15, m = 2000$ and varying $n \in \{800, 1600, 2400, 3200, 4000\}$ or fixing the sample size $n = m = 2000$ and varying $\alpha \in \{0.05, 0.1, 0.15, 0.2\}$.

**Results.** The FDP above the nominal level and power under different nominal levels $\alpha$ for Scenario A1 are shown in Figure 3. The pattern is quite similar to Figure 1 in the main text. The exception here is that the CP method also has its power increasing with the nominal level $\alpha$. This is reasonable since all methods will reject more hypotheses when the nominal error level increases.

We also consider a scenario without label Y.

- **Scenario B1 (without label $Y$):** Another example which does not include the label $Y$. To fit our problem, we consider a spatial setting with $\mathbf{X} = (\boldsymbol{s}, \mathbf{X}^*)$ where $\boldsymbol{s}$ is the spatial variable and $\mathbf{X}^*$ contains the remaining features. We consider $\mathbf{X} \in \mathbb{R}^{d^*}$ with $\boldsymbol{s} = (s_1, s_2) \in \mathbb{R}^2$ and $\mathbf{X}^* \in \mathbb{R}^{d^*-2}$ with $d^* = 50$:

$$\mathbf{X}^* \mid \boldsymbol{s} \sim N(0, r(\boldsymbol{s}) \cdot I_{d^*-2}), \quad r(\boldsymbol{s}) = 0.2 + 0.9\|\boldsymbol{s}\|_2^2$$

  where $s_1, s_2 \sim \mathrm{U}[-1, 1]$ independently. The test data contains $10\%$ outliers with the same distribution for $\boldsymbol{s}$ and a different conditional distribution

$$\mathbf{X}^* \mid \boldsymbol{s} \sim N(0, 4 \cdot r(\boldsymbol{s}) \cdot I_{d^*-2}).$$

**Results.** The results for Scenario B1 are shown in Figure 4-5. In Scenario B1, the power of the LCP-od method grows significantly with the sample size $n$ and the nominal level $\alpha$, while the power of the CP method only rises slightly. Our discussion for Scenario A1 also applies to these results. The only exception that here the CP method has a relatively higher fixed power than those in Scenario A1.

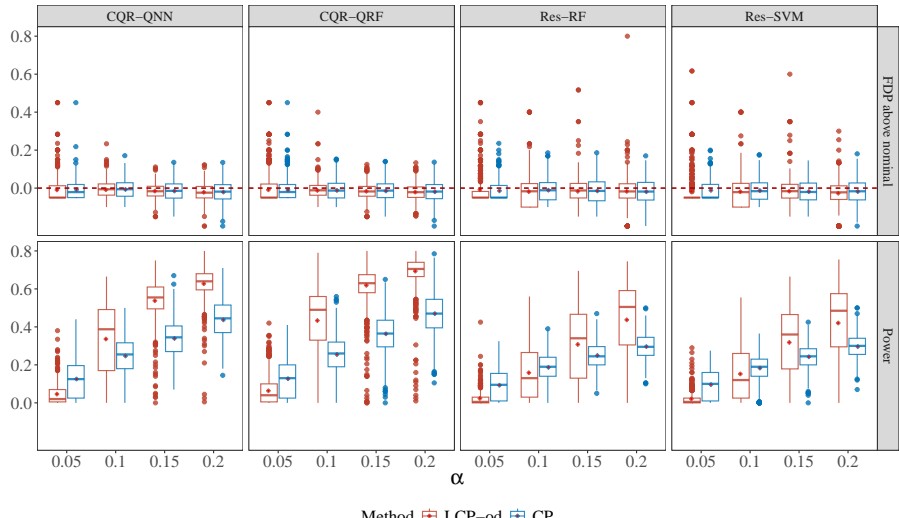

Figure 3: FDP above the nominal level and power under different nominal levels $\alpha$ for Scenario A1. The sample sizes are fixed at $n = m = 2000$.

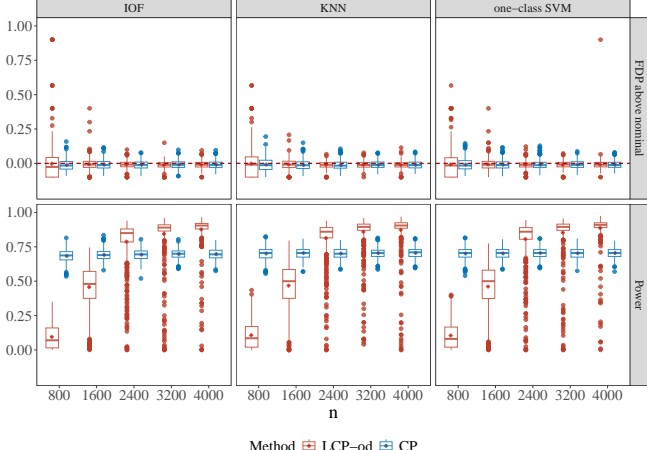

Figure 4: FDP above the nominal level and power under different sample sizes $n$ for Scenario B1. The test sample size is fixed at $m = 2000$ and the nominal level is fixed at $\alpha = 0.1$.

### E.2 SYNTHETIC-DATA RESULTS FOR CONDITIONAL LABEL SCREENING

For the conditional label screening problem, we consider a nonlinear regression scenario:

- **Scenario A2 (nonlinear regression)**: We take a constant $S = 2$ and the response $\mathbf{Y} = (Y_1, Y_2)$ with $Y_1 = -2X_1 + 7X_2^2 + 3\exp(X_3 + 2X_4^2) + \varepsilon$, $Y_2 = -6X_1 + 5X_2^2 + 3\exp(2X_3 + X_4^2) + \varepsilon$, $\mathbf{X} \sim U[-1, 1]^4$ and $\varepsilon \sim \mathcal{N}(0, 1)$. The screening target is $Y_s \in \mathcal{A}_s = [a_s, +\infty)$ where $a_s$ is the 70% quantile of $Y_s$ for $s = 1, 2$, respectively.

**Implementation details.** We fix the sample sizes $n = 500, m = 2000$ and vary $\alpha \in \{0.05, 0.1, 0.15, 0.2\}$. We apply three different algorithms to train a probability prediction function for each component of $\mathbf{Y}$: linear logistic regression (LL), neural network (NN) and random forest (RF).

**Benchmarks.** We compare our conditional label screening method via the LCP (abbreviated as LCP-ls) as summarised in Algorithm 2 with the thresholding procedure without weighting (abbrevi-

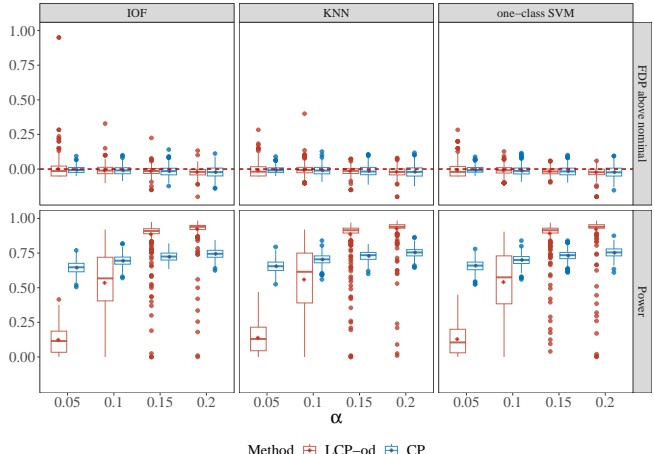

Figure 5: FDP above the nominal level and power under different nominal levels $\alpha$ for Scenario B1. The sample sizes are fixed at $n = m = 2000$.

ated as THR). The THR method is performed by only replacing the LCP by the classical unweighted CP.

**Results.** We calculate the empirical marginal FWER and four kinds of conditional FWER's (conditional on $\mathbf{X} \in \mathcal{B}$ for four different sets $\mathcal{B}$). The evaluation measures are defined as follows:

- (Marginal FWER): $\text{mFWER}_j = \Pr\left(\sum_{s=1}^{S_{2j}} \mathbb{I}\{Y_{2j,s} \notin \mathcal{A}_s, \delta_{j,s} = 1\} > 0\right)$

- (Conditional FWER): $\text{cFWER}_{ij} = \Pr\left(\sum_{s=1}^{S_{2j}} \mathbb{I}\{Y_{2j,s} \notin \mathcal{A}_s, \delta_{j,s} = 1\} > 0 \mid \mathbf{X} \in \mathcal{B}_i\right)$ for
  $i \in \{1, 2, 3, 4\}$ and $j \in \{1, \ldots, m\}$, where $\mathcal{B}_1 = \{\mathbf{X} \in \mathbb{R}^d : X_1 < 0 \text{ and } X_3 < 0\}$, $\mathcal{B}_2 = \{\mathbf{X} \in \mathbb{R}^d : X_1 > 0 \text{ and } X_3 < 0\}$, $\mathcal{B}_3 = \{\mathbf{X} \in \mathbb{R}^d : X_1 < 0 \text{ and } X_3 > 0\}$, and $\mathcal{B}_4 = \{\mathbf{X} \in \mathbb{R}^d : X_1 > 0 \text{ and } X_3 > 0\}$.

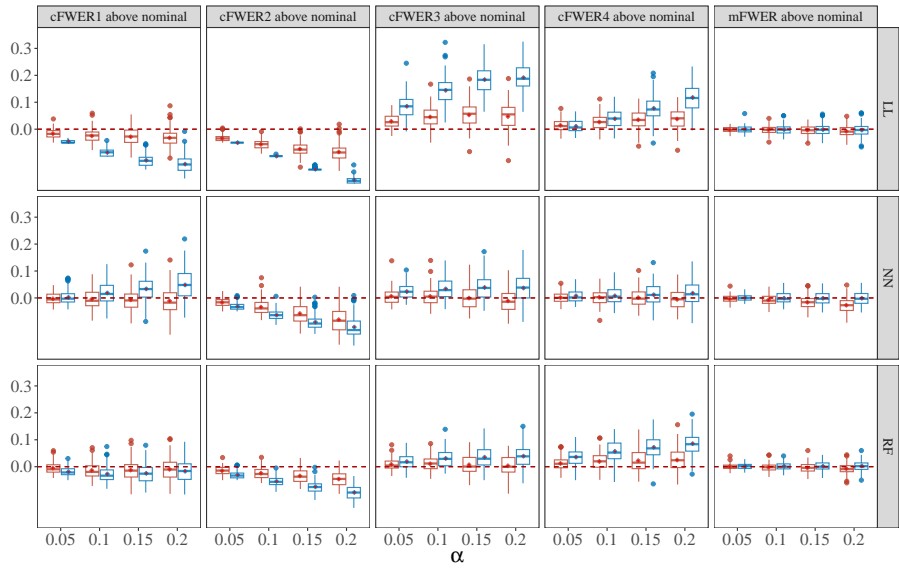

Figure 6: Conditional FWER (cFWER) and marginal FWER (mFWER) above $\alpha$ under different nominal levels $\alpha$ for Scenario A2. The sample size is fixed at $n = 500, m = 2000$.

The results for Scenario A2 are reported in Figure 6. As theoretically ensured, both methods control the marginal FWER at nominal levels. However, the LCP-ls is much more robust against conditionality and exhibit a much smaller conditional FWER inflation than the THR method on subsets $\mathcal{B}_3$ and $\mathcal{B}_4$. Accordingly, the conditional FWER of the THR method on $\mathcal{B}_2$ approaches 0 as the nominal level $\alpha$ increases, while the conditional error rate of the LCP-ls remains much closer to the nominal level. These results suggest that the LCP-ls method offers a more balanced screening result.

### E.3  SYNTHETIC-DATA RESULTS FOR TWO-SAMPLE CONDITIONAL DISTRIBUTION TEST

We consider three different scenarios for the problem of two-sample conditional distribution test, which are analogous to those in Hu & Lei (2023). Since the whole covariate vector $\mathbf{X}$ needs to be conditioned on in this problem, we have $d^* = d$ here.

**Scenario A3**: Let $Y_{1i} = \alpha_1 + \mathbf{X}_{1i}^\top \beta + \varepsilon_{1i}, i = 1, \ldots, n$ and $Y_{2j} = \alpha_2 + \mathbf{X}_{2j}^\top \beta + \varepsilon_{2j}, j = 1, \ldots, m$, where $\mathbf{X}_{1i} \overset{\text{iid}}{\sim} N(\mathbf{0}, I_d), \mathbf{X}_{2j} \overset{\text{iid}}{\sim} N(\boldsymbol{\mu}, I_d)$ with $\boldsymbol{\mu} = (1, 1, -1, -1, 0)^\top$ and $\varepsilon_{1i}, \varepsilon_{2j} \overset{\text{iid}}{\sim} N(0, 1)$ independently. We set $\alpha_1 = \alpha_2 = 0$ under the null and $\alpha_1 = 0, \alpha_2 = 0.5$ under the alternative.

**Scenario B3**: Let $Y_{1i} = \alpha_1 + \beta_1 X_{1i,1} + \beta_2 X_{1i,2} + \beta_3 X_{1i,3}^2 + \beta_4 X_{1i,4}^2 + \beta_5 X_{1i,5}^3 + \varepsilon_{1i}, i = 1, \ldots, n$ and $Y_{2j} = \alpha_{2j} + \beta_1 X_{2j,1} + \beta_2 X_{2j,2} + \beta_3 X_{2j,3}^2 + \beta_4 X_{2j,4}^2 + \beta_5 X_{2j,5}^3 + \varepsilon_{2j}, j = 1, \ldots, m$, where $\mathbf{X}_{1i} \overset{\text{iid}}{\sim} 0.5N(\mathbf{0}, I_d) + 0.5N(\boldsymbol{\mu}, I_d)$ with $\boldsymbol{\mu} = (0.5, 0.5, -0.5, -0.5, 0)^T$, $\mathbf{X}_{2i} \overset{\text{iid}}{\sim} 0.5N(\mathbf{0}, I_d) + 0.5N(\mathbf{0}, 1.5I_d)$ and $\varepsilon_{1i}, \varepsilon_{2j} \overset{\text{iid}}{\sim} t(5)$ independently. We set $\alpha_1 = \alpha_2 = 0$ under the null and $\alpha_1 = 0, \alpha_{2j} = 0.8 \times (1 - 0.1 \cdot \|\mathbf{X}_{2i}\|_2^2)$ under the alternative.

**Scenario C3**: Let $Y_{1i} = \theta(\mathbf{X}_{1i}) + \epsilon_{1i}, i = 1, \ldots, n$ and $Y_{2j} = \theta(\mathbf{X}_{2j}) + \epsilon_{2j}, j = 1, \ldots, m$, where $\theta(\mathbf{X}) = \mathbb{E}(y \mid x)$ is an additive function of B-splines, and $\mathbf{X}_{1i} \overset{\text{iid}}{\sim} 0.5N(\mathbf{0}, I_d) + 0.5N(\boldsymbol{\mu}, I_d)$ with $\boldsymbol{\mu} = (0.5, 0.5, -0.5, -0.5, 0)^T$, $\mathbf{X}_{2i} \overset{\text{iid}}{\sim} 0.5N(\mathbf{0}, I_d) + 0.5N(\mathbf{0}, 1.5I_d)$ and we set $\varepsilon_{\ell i} \overset{\text{iid}}{\sim} N(0, 4/(1 + X_{\ell i}^2(1)))$ under the null and $\varepsilon_{1i} \overset{\text{iid}}{\sim} N(0, 4/(1 + X_{1i}^2(1))), \varepsilon_{2i} \overset{\text{iid}}{\sim} N(0, 1.5/(1 + X_{2i}^2(1)))$ under the alternative.

**Implementation details.** Under each scenario, we fix $d = 5$ and consider different sample sizes $n = m \in \{200, 400, 600, 800, 1000\}$ with equal splitting. The coefficient $\beta$ in model A and B are taken as $\beta = (1, 1, 1, -1, -1)^\top$. For our localized conformal test (LCT) method, we take the Gaussian kernel function and choose the same bandwidth $h = (n/2)^{-1/(d+2)}$. For all three scenarios, we use three different probabilistic classification models to estimate density ratios: linear logistic (LL), random forest (RF) and neural network (NN). The type I error (size) under the null and the power under the alternative are calculated for each scenario across 500 replications with nominal type I error level $\alpha = 0.05$.

**Benchmarks.** We compare our localized conformal test (LCT) as summarised in Algorithm 3 with the following two tests:

- CT: Hu & Lei (2023)'s conformal test as described in Appendix C.
- DCT: Chen & Lei (2024)'s de-biased conformal test, where they formulate the covariate shift problem within the nonparametric framework and utilize the doubly-robust theory to correct the bias of the test statistic.

**Results.** The results for all three scenarios are summarised in Figure 7. Regarding type I error, the CT method is only valid when the training model is correctly specified and exhibits a severely inflated type I error rate under mis-specified models. The DCT method is more robust than the CT method and controls the type I error rate under the nominal level for most scenarios and training models. This corresponds to its doubly-robust property, which relaxes the requirement on accuracy of the estimated density ratios. In contrast, the LCT method controls the type I error rate accurately around the nominal level for any combination of scenario and training model. This shows the robustness and wide applicability of our proposed test statistic.

In terms of power, the CT method is not valid with misspecified training models, so we only discuss the DCT and LCT methods. In Scenarios A3 and B3, the LCT method exhibits significantly higher power than the DCT method especially when the sample size is large. In Scenario C3, the LCT is more powerful than the DCT under the NN algorithm and the two methods perform similarly

when using the LL or RF algorithm. In general, the power improvement is more significant when the classification algorithm can distinguish the two populations better. Except for the numerical performance, the DCT method involves not only data splitting but also a cross-fitting step. The LCT test does not require the latter and is therefore much easier to implement and computationally more efficient.

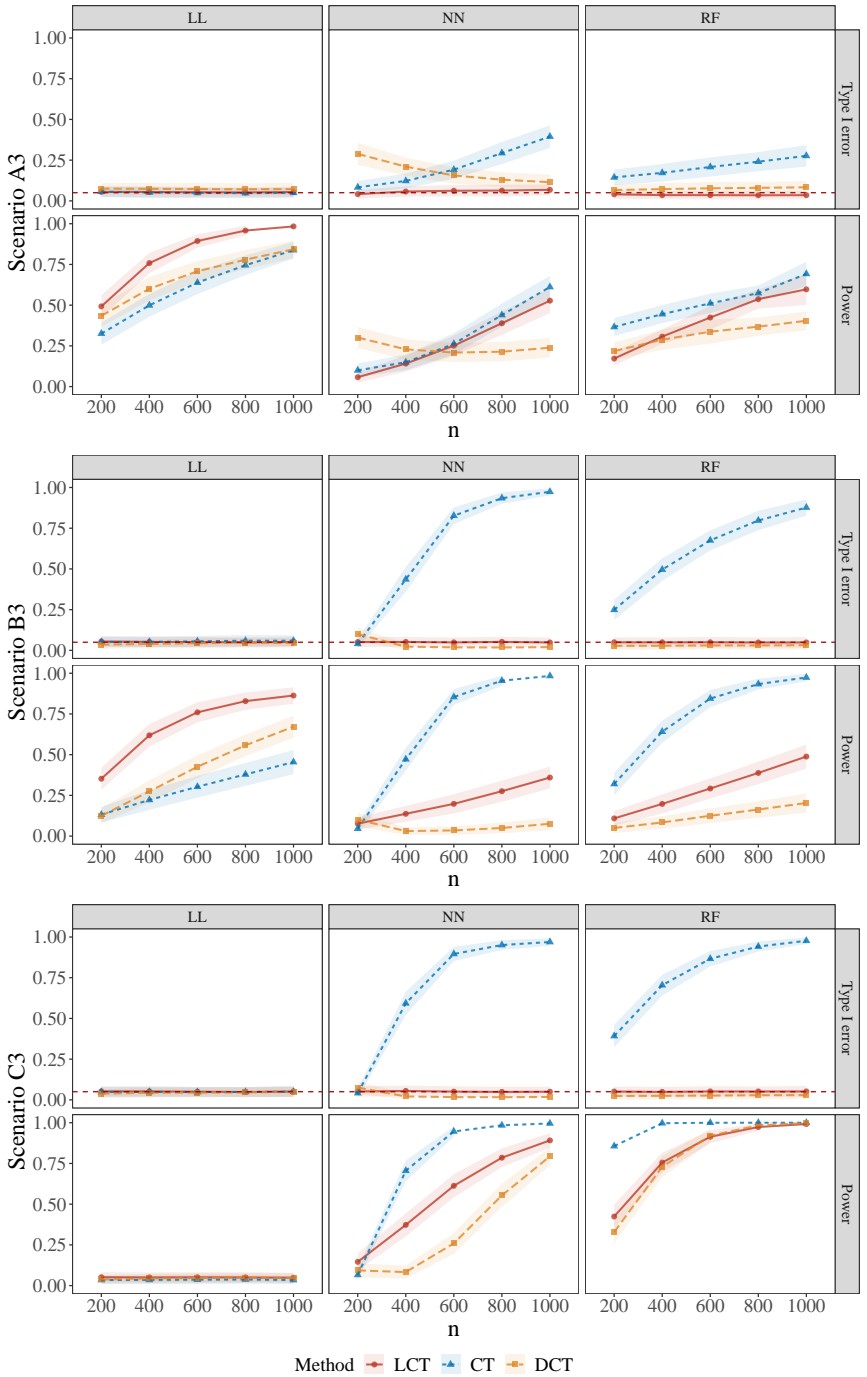

Figure 7: Results for two-sample conditional distribution test: Type I error (size) under $\mathbb{H}_0$ and power under $\mathbb{H}_1$ across different sample sizes $n = m \in \{200, 400, 600, 800, 1000\}$. The red dashed lines denote the nominal level $\alpha = 0.05$. Shading represents error bars of one standard error above and below.

### E.4 CONDITIONAL OUTLIER DETECTION ON SPATIAL DATA

**Dataset.** We consider using the House Sales in the King County, USA dataset (Kaggle, 2016) to demonstrate the performance of our method on the spatial scenario with label $Y$, which has $n = 21613$ individuals with 21 attributes including two spatial features: longitude ($s_1$) and latitude ($s_2$). The response $Y$ is the price of houses.

**Implementation details.** We randomly sample three parts of the data from the whole dataset: $n_{\text{tr}} = 2,000$ training data, $n_{\text{cal}} = 2,000$ calibration data and $n_{\text{te}} = 3,000$ test data. Since the original dataset contains no conditional outliers, we create synthetic outliers and apply different methods for detection. We randomly sample 10% of the testing data to be outliers. For the outliers, we add a random noise $\varepsilon_i$ to the original response $Y_i$. We consider two kinds of synthetic outliers.

- Conditional quantile-based outlier:

$$\tilde{Y}_i = Y_i + \varepsilon_i, \ \varepsilon_i = 0.75 \cdot \text{Quantile}_{0.9}(Y_i \mid s_2) \cdot \xi,$$

  where $\text{Quantile}_{0.9}(Y \mid s_2)$ is the 90% conditional quantile of $Y$ given $s_2$ and $\Pr(\xi = \pm 1) = 1/2$.

- Conditional variance-based outlier:

$$\tilde{Y}_i = Y_i + \varepsilon_i, \ \varepsilon_i = 2 \cdot \sqrt{\text{Var}(Y_i \mid s_2)} \cdot \xi,$$

  where $\text{Var}(Y \mid s_2)$ is the conditional variance of $Y$ given $s_2$ and $\Pr(\xi = 2) = \Pr(\xi = -1) = 1/2$.

All results are based on 500 replications.

**Results.** We compare the LCP-od and the CP methods with the CQR score constructed by two different algorithms: quantile random forest (QRF) and quantile neural network (QNN). The empirical FDR and power for conditional outlier detection are reported in Table 2-3. The result is similar to the simulation part, where all methods control the FDR below the nominal. While the advantage is not as significant as before, the LCP-od method still enjoys the highest power in all cases.

Table 2: Empirical FDR and power for the House Sales dataset with quantile-based outliers. Bold numbers represent the best results. The brackets contain the standard errors.

| $\alpha$ | Method | LCP-od | | CP | |
|---|---|---|---|---|---|
| | Score | CQR-QNN | CQR-QRF | CQR-QNN | CQR-QRF |
| 0.15 | FDR | 0.139 (0.087) | 0.131 (0.030) | 0.128 (0.047) | 0.145 (0.036) |
| | Power | 0.657 (0.096) | **0.871 (0.048)** | 0.593 (0.037) | 0.858 (0.070) |
| 0.2 | FDR | 0.177 (0.033) | 0.175 (0.030) | 0.174 (0.049) | 0.175 (0.037) |
| | Power | 0.882 (0.131) | **0.913 (0.021)** | 0.824 (0.078) | 0.907 (0.022) |

Table 3: Empirical FDR and power for the House Sales dataset with variance-based outliers. Bold numbers represent the best results. The brackets contain the standard errors.

| $\alpha$ | Method | LCP-od | | CP | |
|---|---|---|---|---|---|
| | Score | CQR-QNN | CQR-QRF | CQR-QNN | CQR-QRF |
| 0.15 | FDR | 0.141 (0.051) | 0.127 (0.029) | 0.142 (0.043) | 0.128 (0.035) |
| | Power | 0.520 (0.168) | **0.628 (0.092)** | 0.480 (0.085) | 0.581 (0.119) |
| 0.2 | FDR | 0.181 (0.036) | 0.176 (0.038) | 0.196 (0.034) | 0.186 (0.030) |
| | Power | 0.630 (0.084) | **0.722 (0.056)** | 0.597 (0.099) | 0.692 (0.106) |

### E.5 TWO-SAMPLE CONDITIONAL DISTRIBUTION TEST ON AIRFOIL DATA

**Dataset.** Refer to previous related works (Hu & Lei, 2023; Chen & Lei, 2024), we consider using the airfoil dataset (Brooks et al., 2014) from the UCI Machine Learning Repository to demonstrate the effectiveness of our proposed LCT on real data. This dataset investigates the sound pressure

of various airfoils with $n = 1503$ observations of the response $Y$: the scaled sound pressure and the covariates $\mathbf{X}$ with $d = 5$ dimensions: log frequency, angle of attack, chord length, free-stream velocity, and suction side log displacement thickness.

**Implementation details.** We use part of the rules in Hu & Lei (2023) to split the observations into two samples as follows:

   (i) Random partition. Randomly partition the dataset into two groups with sizes $|\mathcal{D}_1| = 751$ and $|\mathcal{D}_2| = 752$.

   (ii) Exponential tilting. First randomly partition the data into $\mathcal{D}_1, \tilde{\mathcal{D}}_2$. Then construct $\mathcal{D}_2$ by sampling 25% of the points from $\tilde{\mathcal{D}}_2$ with replacement, with probabilities proportional to $\omega(x) = \exp(x^T \alpha)$, where $\alpha = (-1, 0, 0, 0, 1)$.

   (iii) Partition along the response. Split the data according to the value of response, where the first group contains the samples with smaller response values.

In the first two partitions, the two samples satisfy the null hypothesis. The first partition has samples from the same distribution, while the second shows a non-trivial covariate shift. In the last partition, the covariate shift assumption is not satisfied and the alternative hypothesis holds. For all three cases, we use the linear logistic (LL) and neural network (NN) to estimate the density ratios.

**Results.** Similar to Hu & Lei (2023), we use out-of-sample marginal classification error as a proxy of the accuracy of marginal density ratio estimation for all cases (i-iii). The percentage of rejections for cases (i-ii) across 500 repetitions are summarised in Table 4. We can see the rejection proportions of both LCT and DCT are close to the desired level when using LL or NN algorithms, while the CT test fails to control type I error when using NN algorithm due to its low accuracy. For case (iii), we only have a single deterministic generation of the training and testing data. The median $p$-values across 500 random splits for case (iii) are reported in Table 5. While all three tests correctly reject the null in this case, the LCT test gives the smallest $p$-value and is therefore most powerful.

Table 4: Percentage of rejections (PR) and average error of estimating $g$ (Err) in Airfoil dataset for cases (i–ii) using algorithms LL and NN over 500 repetitions, with nominal level $\alpha = 0.05$ for LCT, CT and DCT. The bracket contains the standard error.

| Cases | Method | $PR_{LL}$ | $PR_{NN}$ | $Err_{LL}$ | $Err_{NN}$ |
|---|---|---|---|---|---|
| | LCT | 0.054 (0.0325) | 0.053 (0.0374) | | |
| Case (i) | CT | 0.058 (0.0289) | 0.441 (0.0610) | 0.429 | 0.608 |
| | DCT | 0.053 (0.0299) | 0.066 (0.0268) | | |
| | LCT | 0.067 (0.0323) | 0.066 (0.0390) | | |
| Case (ii) | CT | 0.056 (0.0320) | 0.135 (0.0405) | 0.213 | 0.367 |
| | DCT | 0.056 (0.0327) | 0.035 (0.0250) | | |

Table 5: Median $p$-values (Pval) and average error of estimating $g$ (Err) in Airfoil dataset for case (iii) using algorithms LL and NN over 500 random splits, with nominal level $\alpha = 0.05$ for LCT, CT and DCT.

| Cases | Method | $Pval_{LL}$ | $Pval_{NN}$ | $Err_{LL}$ | $Err_{NN}$ |
|---|---|---|---|---|---|
| | LCT | 0.000 | 0.000 | | |
| Case (iii) | CT | 0.000 | 0.022 | 0.279 | 0.429 |
| | DCT | 0.000 | 0.018 | | |

## F TECHNICAL DETAILS

### F.1 PROOF OF THE DENSITY RATIO IN EQUATION (2.3)

*Proof.* We calculate the density ratio of $\mathbf{X}_{2j}$ and $\mathbf{X}_{1i}$ conditional on $\tilde{\mathbf{X}}_{2j}$ here for the sake of complement. This proof refers to the proof of Proposition 1 in Hore & Barber (2024).

Note that by the sampling step, we have $\tilde{\mathbf{X}}_{2j} \mid (\mathbf{X}_{2j}, Y_{2j}) \sim H(\mathbf{X}_{2j}, \cdot)$. The joint distribution of $(\mathbf{X}_{2j}, Y_{2j}, \tilde{\mathbf{X}}_{2j})$ is as follows:

$$
\begin{cases}
\mathbf{X}_{2j} \sim P_{2,X}, \\
Y_{2j} \mid X_{2j} \sim P_{Y|X}(\cdot \mid \mathbf{X}_{2j}), \\
\tilde{\mathbf{X}}_{2j} \mid (\mathbf{X}_{2j}, Y_{2j}) \sim H(\mathbf{X}_{2j}, \cdot).
\end{cases}
$$

A direct calculation then yields $(\mathbf{X}_{2j}, Y_{2j}) \mid \tilde{\mathbf{X}}_{2j} \sim \left( P_{2,X} \circ H(\cdot, \tilde{\mathbf{X}}_{2j}) \right) \times P_{2,j,Y|X}$ by Bayesian's theorem, where $P_{2,X} \circ H(\cdot, \tilde{\mathbf{X}}_{2j}) = \tilde{P}_{2,X}$, which is the distribution $P_{2,X}$ reweighted by $H(\cdot, \tilde{\mathbf{X}}_{2j})$. That is, $\tilde{P}_{2,X}(A) = (P_{2,X} \circ H(\cdot, \tilde{\mathbf{X}}_{2j}))(A) = \frac{\int_A H(\cdot, \tilde{\mathbf{X}}_{2j}) \, \mathrm{d}P_{2,X}(x)}{\int_{\mathcal{X}} H(\cdot, \tilde{\mathbf{X}}_{2j}) \, \mathrm{d}P_{2,X}(x)}$ for all $A \subseteq \mathcal{X}$.

This is an instance of a covariate shift problem: the distribution of $\mathbf{X}_{2j} \mid \tilde{\mathbf{X}}_{2j} \neq P_{2,X}$, but the conditional distribution distribution of $Y_{2j}$ remains the same as for the training data. i.e., $P_{2,j,Y|X}$.

Hence, by the definition of covariate shift, the density ratio of $\mathbf{X}_{2j}$ and $\mathbf{X}_{1i}$ conditional on $\tilde{\mathbf{X}}_{2j}$ is

$$
\frac{\mathrm{d}P_{2,\mathbf{X}|\tilde{\mathbf{X}}_{2j}}}{\mathrm{d}P_{1,\mathbf{X}}}(\mathbf{x}) = \frac{f_{2,\mathbf{X}}(\mathbf{x}) H(\mathbf{x}, \tilde{\mathbf{X}}_{2j})}{f_{1,\mathbf{X}}(\mathbf{x}) \int_{\mathcal{X}} H(\cdot, \tilde{\mathbf{X}}_{2j}) \, \mathrm{d}P_{2,X}(x)} = \frac{H(\mathbf{x}, \tilde{\mathbf{X}}_{2j})}{\int_{\mathcal{X}} H(\cdot, \tilde{\mathbf{X}}_{2j}) \, \mathrm{d}P_{2,X}(x)} = c \cdot H(\mathbf{x}, \tilde{\mathbf{X}}_{2j}),
$$

where $c = 1 / \int_{\mathcal{X}} H(\cdot, \tilde{\mathbf{X}}_{2j}) \, \mathrm{d}P_{2,X}(x)$ is a constant, which completes the proof. $\qquad \square$

### F.2 PROOF OF LEMMA 2

*Proof.* Denote $P_{1,\tilde{\mathbf{X}}}, P_{2,\tilde{\mathbf{X}}}$ as the marginal distribution of $\tilde{\mathbf{X}}$ if we fist sample $\mathbf{X}$ from $P_{1,\mathbf{X}}$ or $P_{2,\mathbf{X}}$ and then sample $\tilde{\mathbf{X}}$ from $H(\mathbf{X}, \cdot)$. The distributions of the calibration and test points are $P_{1,(\mathbf{X},Y)|\tilde{\mathbf{X}}} \times P_{1,\tilde{\mathbf{X}}}$ and $P_{2,(\mathbf{X},Y)|\tilde{\mathbf{X}}} \times P_{2,\tilde{\mathbf{X}}}$ where $P_{1,(\mathbf{X},Y)|\tilde{\mathbf{X}}}$ and $P_{2,(\mathbf{X},Y)|\tilde{\mathbf{X}}}$ denote the corresponding conditional distributions of calibration and test data points.

Notice that the super-uniform property still holds if the conditional distribution $P_{1,(\mathbf{X},Y)|\tilde{\mathbf{X}}} = P_{2,(\mathbf{X},Y)|\tilde{\mathbf{X}}}$. Therefore, the super-uniform bound holds for an independent tuple $(\mathbf{X}^*, Y^*, \tilde{\mathbf{X}}^*) \sim P_{1,(\mathbf{X},Y)|\mathbf{X}} \times P_{2,\tilde{\mathbf{X}}}$. That is

$$
p_{\mathrm{L}}^* = \frac{\sum_{i \in \mathcal{C}} H(\mathbf{X}_{1i}, \tilde{\mathbf{X}}^*) \mathbb{I}\{\tilde{V}^* \leq V_{1i}\} + \xi^* \cdot H(\mathbf{X}^*, \tilde{\mathbf{X}}^*)}{\sum_{i \in \mathcal{C}} H(\mathbf{X}_{1i}, \tilde{\mathbf{X}}^*) + H(\mathbf{X}^*, \tilde{\mathbf{X}}^*)},
$$

$$
\Pr(p_{\mathrm{L}}^* \leq \alpha) \leq \alpha.
$$

The probability for $(\mathbf{X}_{2n+j}, Y_{2n+j}, \tilde{\mathbf{X}}_{2n+j})$ is then upper bounded by

$$
\begin{aligned}
\Pr(p_{\mathrm{L},j} \leq \alpha) &\leq \alpha + \mathbb{E}_{\tilde{\mathbf{X}} \sim P_{2,\tilde{\mathbf{X}}}} \left\{ \mathrm{d}_{\mathrm{TV}} \left( P_{1,(\mathbf{X},Y)|\tilde{\mathbf{X}}}, P_{2,(\mathbf{X},Y)|\tilde{\mathbf{X}}} \right) \right\} \\
&= \alpha + \mathbb{E}_{\tilde{\mathbf{X}} \sim P_{2,\tilde{\mathbf{X}}}} \left\{ \mathrm{d}_{\mathrm{TV}} \left( P_{1,\mathbf{X}|\tilde{\mathbf{X}}}, P_{2,\mathbf{X}|\tilde{\mathbf{X}}} \right) \right\}.
\end{aligned}
$$

For a fixed $\tilde{\mathbf{X}}$ we have

$$
\begin{aligned}
\mathrm{d}_{\mathrm{TV}} \left( P_{1,\mathbf{X}|\tilde{\mathbf{X}}}, P_{2,\mathbf{X}|\tilde{\mathbf{X}}} \right) &= \frac{1}{2} \mathbb{E}_{\mathbf{X} \sim P_{1,\tilde{\mathbf{X}}|\mathbf{x}}} \left\{ \left| \frac{\mathrm{d}P_{2,\mathbf{X}|\tilde{\mathbf{X}}}(\mathbf{X})}{\mathrm{d}P_{1,\mathbf{X}|\tilde{\mathbf{X}}}(\mathbf{X})} - 1 \right| \right\} \\
&= \frac{1}{2} \mathbb{E}_{\mathbf{X} \sim P_{1,\tilde{\mathbf{X}}|\mathbf{x}}} \left\{ \left| \frac{g(\mathbf{X})}{\mathbb{E}_{\mathbf{X}' \sim P_{1,\mathbf{X}|\tilde{\mathbf{X}}}} \{g(\mathbf{X}')\}} - 1 \right| \right\} \\
&\leq \frac{\mathbb{E}_{\mathbf{X}, \mathbf{X}' \sim P_{1,\mathbf{X}|\tilde{\mathbf{X}}}} \{|g(\mathbf{X}) - g(\mathbf{X}')|\}}{2 \mathbb{E}_{\mathbf{X} \sim P_{1,\mathbf{X}|\tilde{\mathbf{X}}}} \{g(\mathbf{X})\}} \\
&= \mathbb{E}_{\mathbf{X}, \mathbf{X}' \sim P_{1,\mathbf{X}|\tilde{\mathbf{X}}}} \{|g(\mathbf{X}) - g(\mathbf{X}')|\} \frac{\mathbb{E}_{\mathbf{X} \sim P_{1,\mathbf{X}}} \{H(\mathbf{X}, \tilde{\mathbf{X}})\}}{2 \mathbb{E}_{\mathbf{X} \sim P_{1,\mathbf{X}}} \{g(\mathbf{X}) H(\mathbf{X}, \tilde{\mathbf{X}})\}}.
\end{aligned}
$$

Taking expectation with respect to $\tilde{\mathbf{X}}$ we have

$$\mathbb{E}_{\tilde{\mathbf{X}} \sim P_{2,\tilde{\mathbf{x}}}} \left\{ \mathrm{d}_{\mathrm{TV}} \left( P_{1,\mathbf{X}|\tilde{\mathbf{x}}}, P_{2,\mathbf{X}|\tilde{\mathbf{x}}} \right) \right\}$$

$$\leq \int \mathbb{E}_{\mathbf{X} \sim P_{1,\mathbf{x}}} \{ g(\mathbf{X}) H(\mathbf{X}, \mathbf{x}) \} \mathbb{E}_{\mathbf{X}, \mathbf{X}' \sim P_{1,\mathbf{X}|\tilde{\mathbf{X}}=\mathbf{x}}} \{ |g(\mathbf{X}) - g(\mathbf{X}')| \} \frac{\mathbb{E}_{\mathbf{X} \sim P_{1,\mathbf{x}}} \{ H(\mathbf{X}, \mathbf{x}) \}}{2 \mathbb{E}_{\mathbf{X} \sim P_{1,\mathbf{x}}} \{ g(\mathbf{X}) H(\mathbf{X}, \mathbf{x}) \}} \mathrm{d}\mathbf{x}$$

$$= \frac{1}{2} \int \mathbb{E}_{\mathbf{X}, \mathbf{X}' \sim P_{1,\mathbf{X}|\tilde{\mathbf{X}}=\mathbf{x}}} \{ |g(\mathbf{X}) - g(\mathbf{X}')| \} \mathbb{E}_{\mathbf{X} \sim P_{1,\mathbf{x}}} \{ H(\mathbf{X}, \mathbf{x}) \} \mathrm{d}\mathbf{x}$$

$$= \frac{1}{2} \int \mathbb{E}_{\mathbf{X}, \mathbf{X}' \sim P_{1,\mathbf{X}|\tilde{\mathbf{X}}=\mathbf{x}}} \{ |g(\mathbf{X}) - g(\mathbf{X}')| \} \mathbb{E}_{\mathbf{X} \sim P_{1,\mathbf{x}}} \mathrm{d}P_{H,\mathbf{X}}$$

where $P_{H,\mathbf{X}}$ has a density function $f_{H,\mathbf{X}}(\mathbf{x}) = \mathbb{E}_{\mathbf{X} \sim P_{1,\mathbf{x}}} \{ H(\mathbf{X}, \mathbf{x}) \}$.

By integration

$$\mathbb{E}_{\mathbf{X}, \mathbf{X}' \sim P_{1,\mathbf{X}|\tilde{\mathbf{X}}=\mathbf{x}}} \{ |g(\mathbf{X}) - g(\mathbf{X}')| \}$$

$$= \int \frac{1}{h^{2d}} K \left( \frac{\mathbf{x}' - \mathbf{x}}{h} \right) K \left( \frac{\mathbf{x}'' - \mathbf{x}}{h} \right) f_{1,\mathbf{x}}(\mathbf{x}') f_{1,\mathbf{x}}(\mathbf{x}'') |g(\mathbf{x}') - g(\mathbf{x}'')| \mathrm{d}\mathbf{x}' \mathrm{d}\mathbf{x}''$$

$$= \int K(\boldsymbol{u}) K(\boldsymbol{v}) |g(\mathbf{x} + h\boldsymbol{u}) - g(\mathbf{x} + h\boldsymbol{v})| f_{1,\mathbf{x}}(\mathbf{x} + h\boldsymbol{u}) f_{1,\mathbf{x}}(\mathbf{x} + h\boldsymbol{v}) \mathrm{d}\boldsymbol{u} \mathrm{d}\boldsymbol{v}$$

$$\leq 2 \| f_{1,\mathbf{x}} \|_\infty \mathbb{E}_{\boldsymbol{U} \sim K(\cdot)} \{ |g(\mathbf{x} + h\boldsymbol{U}) - g(\mathbf{x})| \}.$$

Combining the results above we conclude

$$\Pr(p_{\mathrm{L},j} \leq \alpha) \leq \alpha + \mathbb{E}_{\tilde{\mathbf{X}} \sim P_{2,\tilde{\mathbf{x}}}} \left\{ \mathrm{d}_{\mathrm{TV}} \left( P_{1,\mathbf{X}|\tilde{\mathbf{x}}}, P_{2,\mathbf{X}|\tilde{\mathbf{x}}} \right) \right\}$$

$$\leq \alpha + \| f_{1,\mathbf{x}} \|_\infty \mathbb{E}_{\mathbf{X} \sim P_{H,\mathbf{X}}, \boldsymbol{U} \sim K(\cdot)} \{ |g(\mathbf{X} + h\boldsymbol{U}) - g(\mathbf{X})| \}.$$

$\square$

### F.3 Discussion on Lemma 3

The deviation bound in Lemma 3 has two terms corresponding to the bias and variance of $p$-value functions. This is different from Lemma 2 in which we only need $h \to 0$ to eliminate bias and ensure validity. Fixing the sample size $n$ and taking $h \to 0$, the LCP will degenerate to the independent uniform random variable $\xi$, which remains valid but does not contain any information of the data. Otherwise if we take $h \to \infty$ it will degenerate to the unweighted CP. Therefore, the bandwidth $h$ controls the trade-off between the conditional validity of the LCP and the effective sample size.

### F.4 Proof of Lemma 3

*Proof.* Remember the definition of the RBF and box kernel

$$H_{\mathrm{RBF}}(\mathbf{x}, \mathbf{x}') = \frac{1}{V_{d,\mathrm{RBF}} h^d} \exp \left\{ -\frac{\| \mathbf{x} - \mathbf{x}' \|_2^2}{2h^2} \right\}, \quad H_{\mathrm{box}}(\mathbf{x}, \mathbf{x}') = \frac{1}{V_{d,\mathrm{box}} h^d} \mathbb{I}\{ \| \mathbf{x} - \mathbf{x}' \|_2 \leq \sqrt{2} h \}$$

where $V_{d,\mathrm{RBF}}, V_{d,\mathrm{box}}$ are normalizing constants.

The LCP function is

$$p_{\mathrm{L}}(\mathbf{x}, y) = \frac{\sum_{i \in \mathcal{C}} H(\mathbf{X}_{1i}, \tilde{\mathbf{X}}) \mathbb{I}\{ v \leq V_{1i} \} + \xi \cdot H(\mathbf{x}, \tilde{\mathbf{X}})}{\sum_{i \in \mathcal{C}} H(\mathbf{X}_{1i}, \tilde{\mathbf{X}}) + H(\mathbf{x}, \tilde{\mathbf{X}})}. \tag{F.1}$$

For the difference we have

$$|p_{\mathrm{L}}(\mathbf{x}, y) - (1 - F_{V|\mathbf{X}=\mathbf{x}}(v))|$$

$$= \left| \frac{\sum_{i \in \mathcal{C}} H(\mathbf{X}_{1i}, \tilde{\mathbf{X}}) [\mathbb{I}\{ v \leq V_{1i} \} - \{ 1 - F_{V|\mathbf{X}=\mathbf{x}}(v) \}] + [\xi - \{ 1 - F_{V|\mathbf{X}=\mathbf{x}}(v) \}] \cdot H(\mathbf{x}, \tilde{\mathbf{X}})}{\sum_{i \in \mathcal{C}} H(\mathbf{X}_{1i}, \tilde{\mathbf{X}}) + H(\mathbf{x}, \tilde{\mathbf{X}})} \right|$$

$$\leq \left| \frac{\sum_{i \in \mathcal{C}} H(\mathbf{X}_{1i}, \tilde{\mathbf{X}}) [\mathbb{I}\{ v \leq V_{1i} \} - \{ 1 - F_{V|\mathbf{X}=\mathbf{x}}(v) \}]}{\sum_{i \in \mathcal{C}} H(\mathbf{X}_{1i}, \tilde{\mathbf{X}})} \right| + \left| \frac{\| H(\mathbf{x}, \mathbf{x}') \|_\infty}{\sum_{i \in \mathcal{C}} H(\mathbf{X}_{1i}, \tilde{\mathbf{X}})} \right|$$

For the denominator $n_H = \sum_{i \in \mathcal{C}} H(\mathbf{X}_{1i}, \tilde{\mathbf{X}})$, first notice for the RBF kernel

$$
\begin{aligned}
n_{\mathrm{RBF}} &= \sum_{i \in \mathcal{C}} \frac{1}{V_{d,\mathrm{RBF}} h^d} \exp\left\{ -\frac{\|\mathbf{X}_{1i} - \tilde{\mathbf{X}}\|_2^2}{2h^2} \right\} \\
&\geq \exp\{-1\} \frac{V_{d,\mathrm{box}}}{V_{d,\mathrm{RBF}}} \sum_{i \in \mathcal{C}} \frac{1}{V_{d,\mathrm{box}} h^d} \mathbb{I}\{\|\mathbf{X}_{1i} - \tilde{\mathbf{X}}\|_2 \leq \sqrt{2}h\} \\
&= \exp\{-1\} \frac{V_{d,\mathrm{box}}}{V_{d,\mathrm{RBF}}} \cdot n_{\mathrm{box}}.
\end{aligned}
$$

Define $n_{\mathrm{box}}^* = \sum_{i \in \mathcal{C}} \mathbb{I}\{\|\mathbf{X}_{1i} - \tilde{\mathbf{X}}\|_2 \leq \sqrt{2}h\}$, for constants $0 < \lambda, \epsilon < 1$ we have

$$
\begin{aligned}
&\Pr\left( \mathbb{E}(n_{\mathrm{box}}^*) - n_{\mathrm{box}}^* \geq \lambda \mathbb{E}(n_{\mathrm{box}}^*) \mid \tilde{\mathbf{X}} \right) \\
={}&\Pr\left( \exp\left\{ \mathbb{E}(n_{\mathrm{box}}^*) - n_{\mathrm{box}}^* \right\} \geq \exp\left\{ \lambda \mathbb{E}(n_{\mathrm{box}}^*) \right\} \mid \tilde{\mathbf{X}} \right) \\
\leq{}&\mathbb{E}\left[ \exp\left\{ \mathbb{E}(n_{\mathrm{box}}^*) - n_{\mathrm{box}}^* \right\} \mid \tilde{\mathbf{X}} \right] \exp\left\{ -\lambda \mathbb{E}(n_{\mathrm{box}}^*) \right\} \\
={}&\mathbb{E}\left[ \exp\left\{ \mathbb{I}\{\mathbf{X}' \in B_{\sqrt{2}h}(\tilde{\mathbf{X}})\} - \Pr(\mathbf{X}' \in B_{\sqrt{2}h}(\tilde{\mathbf{X}})) \right\} \mid \tilde{\mathbf{X}} \right]^{|\mathcal{C}|} \exp\left\{ -\lambda \mathbb{E}(n_{\mathrm{box}}^*) \right\} \\
={}&\left[ \Pr(\mathbf{X} \in B_{\sqrt{2}h}(\tilde{\mathbf{X}})) \exp\left\{ \Pr(\mathbf{X} \notin B_{\sqrt{2}h}(\tilde{\mathbf{X}})) \right\} + \Pr(\mathbf{X} \notin B_{\sqrt{2}h}(\tilde{\mathbf{X}})) \exp\left\{ -\Pr(\mathbf{X} \in B_{\sqrt{2}h}(\tilde{\mathbf{X}})) \right\} \right]^{|\mathcal{C}|} \\
&\cdot \exp\left\{ -\lambda \mathbb{E}(n_{\mathrm{box}}^*) \right\} \\
\leq{}&\left[ \Pr(\mathbf{X} \in B_{\sqrt{2}h}(\tilde{\mathbf{X}})) \exp\{1\} + \{1 - \Pr(\mathbf{X} \in B_{\sqrt{2}h}(\tilde{\mathbf{X}}))\}\{1 - \epsilon \cdot \Pr(\mathbf{X} \in B_{\sqrt{2}h}(\tilde{\mathbf{X}}))\} \right]^{|\mathcal{C}|} \exp\left\{ -\lambda \mathbb{E}(n_{\mathrm{box}}^*) \right\} \\
={}&\left[ 1 + (e - 1 - \epsilon) \Pr(\mathbf{X} \in B_{\sqrt{2}h}(\tilde{\mathbf{X}})) + \epsilon \Pr(\mathbf{X} \in B_{\sqrt{2}h}(\tilde{\mathbf{X}}))^2 \right]^{|\mathcal{C}|} \exp\left\{ -\lambda|\mathcal{C}| \Pr(\mathbf{X}' \in B_{\sqrt{2}h}(\tilde{\mathbf{X}})) \right\} \\
\leq{}&\exp\left\{ -|\mathcal{C}|(1 + \epsilon + \lambda - e) \Pr(\mathbf{X} \in B_{\sqrt{2}h}(\tilde{\mathbf{X}})) + \epsilon|\mathcal{C}| \Pr(\mathbf{X} \in B_{\sqrt{2}h}(\tilde{\mathbf{X}}))^2 \right\}
\end{aligned}
$$

and

$$
2C_d h^d f(\mathbf{x}) \geq C_d h^d \{f(\mathbf{x}) + o(1)\} \geq \Pr(\mathbf{X} \in B_h(\tilde{\mathbf{X}})) \geq C_d h^d \{f(\mathbf{x}) - o(1)\} \geq \frac{C_d h^d f(\mathbf{x})}{2}
$$

for sufficiently large $n$, where $C_d > 0$ only depends on $d$. Taking $\epsilon = \lambda = 0.9$ we have

$$
\Pr\left( n_{\mathrm{box}} \geq \frac{|\mathcal{C}|f(\mathbf{x})}{20 V_{d,\mathrm{box}}} \right) \geq 1 - \exp\left\{ -\frac{2.8 - e}{2} n h^d \gamma C_d f(\mathbf{x}) + 3.6 n h^{2d} \gamma C_d^2 f(\mathbf{x})^2 \right\} \to 1.
$$

Therefore with probability tending to 1 we have $n_H \geq C \cdot |\mathcal{C}| f(\mathbf{x})$ for some fixed constant $C > 0$. Then

$$
\begin{aligned}
&|p_{\mathrm{L}}(\mathbf{x}, y) - (1 - F_{V|\mathbf{X}=\mathbf{x}}(v))| \\
\leq{}&\frac{1}{Cf(\mathbf{x})} \left\{ \left| \frac{\sum_{i \in \mathcal{C}} H(\mathbf{X}_{1i}, \tilde{\mathbf{X}})[\mathbb{I}\{v \leq V_{1i}\} - \{1 - F_{V|\mathbf{X}=\mathbf{x}}(v)\}]}{|\mathcal{C}|} \right| + \left| \frac{1}{n h^d \gamma V_{d,H}} \right| \right\}. \quad \text{(F.2)}
\end{aligned}
$$

For the first term

$$
\begin{aligned}
&\mathbb{E}\left\{ \frac{(\sum_{i \in \mathcal{C}} H(\mathbf{X}_{1i}, \tilde{\mathbf{X}})[\mathbb{I}\{v \leq V_{1i}\} - \{1 - F_{V|\mathbf{X}=\mathbf{x}}(v)\}])^2}{|\mathcal{C}|^2} \right\} \\
={}&\frac{1}{|\mathcal{C}|} \mathbb{E}\left( H(\mathbf{X}_{1i}, \tilde{\mathbf{X}})^2 [\mathbb{I}\{v \leq V_{1i}\} - \{1 - F_{V|\mathbf{X}=\mathbf{x}}(v)\}]^2 \right) \\
&+ \frac{|\mathcal{C}| - 1}{|\mathcal{C}|} \mathbb{E}\left( H(\mathbf{X}_{1i}, \tilde{\mathbf{X}}) H(\mathbf{X}_{1j}, \tilde{\mathbf{X}})[\mathbb{I}\{v \leq V_{1i}\} - \{1 - F_{V|\mathbf{X}=\mathbf{x}}(v)\}] \cdot [\mathbb{I}\{v \leq V_{1j}\} - \{1 - F_{V|\mathbf{X}=\mathbf{x}}(v)\}] \right) \\
\leq{}&\frac{1}{|\mathcal{C}|} \mathbb{E}\left\{ H(\mathbf{X}_{1i}, \tilde{\mathbf{X}})^2 \right\} + \mathbb{E}\left\{ H(\mathbf{X}_{1i}, \tilde{\mathbf{X}}) H(\mathbf{X}_{1j}, \tilde{\mathbf{X}}) \|F_{V|\mathbf{X}=\mathbf{X}_{1i}} - F_{V|\mathbf{X}=\mathbf{x}}\|_\infty \|F_{V|\mathbf{X}=\mathbf{X}_{1j}} - F_{V|\mathbf{X}=\mathbf{x}}\|_\infty \right\}
\end{aligned}
$$

By integration

$$\mathbb{E}\left\{H(\mathbf{X}_{1i}, \tilde{\mathbf{X}})^2\right\}$$

$$=\mathbb{E}\left\{\int \frac{1}{h^{2d}} K^2\left(\frac{\mathbf{x}' - \tilde{\mathbf{X}}}{h}\right) f_{1,\mathbf{X}}(\mathbf{x}')\mathrm{d}\mathbf{x}'\right\}$$

$$=\frac{1}{h^d}\mathbb{E}\left\{\int K^2(\boldsymbol{u}) f_{1,\mathbf{X}}(\tilde{\mathbf{X}} + h\boldsymbol{u})\mathrm{d}\boldsymbol{u}\right\}$$

$$\leq\frac{\|f_{1,\mathbf{X}}\|_\infty}{h^d}\int K^2(\boldsymbol{u})\mathrm{d}\boldsymbol{u}$$

$$\mathbb{E}\left\{H(\mathbf{X}_{1i}, \tilde{\mathbf{X}})H(\mathbf{X}_{1j}, \tilde{\mathbf{X}})\|F_{V|\mathbf{X}=\mathbf{X}_{1i}} - F_{V|\mathbf{X}=\mathbf{x}}\|_\infty\|F_{V|\mathbf{X}=\mathbf{X}_{1j}} - F_{V|\mathbf{X}=\mathbf{x}}\|_\infty\right\}$$

$$=\mathbb{E}\left\{\int \frac{1}{h^{2d}} K\left(\frac{\mathbf{x}' - \tilde{\mathbf{X}}}{h}\right) K\left(\frac{\mathbf{x}'' - \tilde{\mathbf{X}}}{h}\right)\|F_{V|\mathbf{X}=\mathbf{x}'} - F_{V|\mathbf{X}=\mathbf{x}}\|_\infty\|F_{V|\mathbf{X}=\mathbf{x}''} - F_{V|\mathbf{X}=\mathbf{x}}\|_\infty f_{1,\mathbf{X}}(\mathbf{x}')f_{1,\mathbf{X}}(\mathbf{x}'')\mathrm{d}\mathbf{x}'\mathrm{d}\mathbf{x}''\right\}$$

$$\leq\|f_{1,\mathbf{X}}\|_\infty^2\mathbb{E}\left\{\int K(\boldsymbol{u}) K(\boldsymbol{u}')\|F_{V|\mathbf{X}=\tilde{\mathbf{X}}+h\boldsymbol{u}} - F_{V|\mathbf{X}=\mathbf{x}}\|_\infty\|F_{V|\mathbf{X}=\tilde{\mathbf{X}}+h\boldsymbol{u}'} - F_{V|\mathbf{X}=\mathbf{x}}\|_\infty\mathrm{d}\boldsymbol{u}\mathrm{d}\boldsymbol{u}'\right\}$$

$$\leq\|f_{1,\mathbf{X}}\|_\infty^2\mathbb{E}\left\{\int K(\boldsymbol{u}) K(\boldsymbol{u}')\|\tilde{\mathbf{X}} - \mathbf{x} + h\boldsymbol{u}\|_2^\beta\|\tilde{\mathbf{X}} - \mathbf{x} + h\boldsymbol{u}'\|_2^\beta\mathrm{d}\boldsymbol{u}\mathrm{d}\boldsymbol{u}'\right\}$$

$$=\|f_{1,\mathbf{X}}\|_\infty^2\int K(\boldsymbol{u}) K(\boldsymbol{u}')\|\tilde{\mathbf{x}} - \mathbf{x} + h\boldsymbol{u}\|_2^\beta\|\tilde{\mathbf{x}} - \mathbf{x} + h\boldsymbol{u}'\|_2^\beta\frac{1}{h^d}H\left(\frac{\tilde{\mathbf{x}} - \mathbf{x}}{h}\right)\mathrm{d}\boldsymbol{u}\mathrm{d}\boldsymbol{u}'\mathrm{d}\tilde{\mathbf{x}}$$

$$=h^{2\beta}\|f_{1,\mathbf{X}}\|_\infty^2\int K(\boldsymbol{u}) K(\boldsymbol{u}') K(\boldsymbol{u}'')\|\boldsymbol{u} + \boldsymbol{u}''\|_2^\beta\|\boldsymbol{u}' + \boldsymbol{u}''\|_2^\beta\mathrm{d}\boldsymbol{u}\mathrm{d}\boldsymbol{u}'\mathrm{d}\boldsymbol{u}''$$

$$\leq 3h^{2\beta}\|f_{1,\mathbf{X}}\|_\infty^2\left\{\left(\int K(\boldsymbol{u})\|\boldsymbol{u}\|_2^\beta\mathrm{d}\boldsymbol{u}\right)^2 + \int K(\boldsymbol{u})\|\boldsymbol{u}\|_2^{2\beta}\mathrm{d}\boldsymbol{u}\right\}$$

As all integrations in the final expressions exist, we have

$$\mathbb{E}\left\{\frac{(\sum_{i\in\mathcal{C}} H(\mathbf{X}_{1i}, \tilde{\mathbf{X}})[\mathbb{I}\{v \leq V_{1i}\} - \{1 - F_{V|\mathbf{X}=\mathbf{x}}(v)\}])^2}{|\mathcal{C}|^2}\right\}$$

$$\leq\frac{C_{K,1}\|f_{1,\mathbf{X}}\|_\infty}{nh^d} + C_{K,2}\|f_{1,\mathbf{X}}\|_\infty^2 h^{2\beta}.$$

Together with inequality (F.2) we complete the proof. □

### F.5 PROOF OF THEOREM 1

*Proof.* By the construction of the refined LCP's, they are valid conditional on the event $Y_{2j} \notin \mathcal{A}$ in finite sample

$$\Pr(p_{\mathrm{rL},j} \leq \alpha) \leq \alpha,$$

and marginal PSER control result is proved. For the conditional PSER inflation bound, the proof is exactly the same to Theorem 4 with the only difference that the distribution of $\mathbf{X}$ is replaced by the conditional distribution of $\mathbf{X}$ given the event $Y \notin \mathcal{A}$. We therefore delay the computation of the deviation term to the proof of Theorem 4.

### F.6 PROOF OF THEOREM 2

*Proof.* Denote the index set of inliers in $\mathcal{D}_2$ as $\mathbb{H}_0^*$, We firstly need a lemma commonly used in literature relating the conditional calibration technique.

**Lemma 4.** *The pruned rejection set $\mathcal{R}$ satisfies*

$$\mathrm{FDR} = \mathbb{E}\left[\frac{\sum_{j=1}^m \mathbb{I}\{j \in \mathcal{R}, j \in \mathbb{H}_0^*\}}{|\mathcal{R}| \vee 1}\right] \leq \sum_{j=1}^m \mathbb{E}\left[\frac{\mathbb{I}\{p_{\mathrm{L},j} \leq \frac{\alpha|\widehat{\mathcal{R}}_{j\to0}|}{m}, j \in \mathbb{H}_0^*\}}{|\widehat{\mathcal{R}}_{j\to0}|}\right].$$

The proof of Lemma 4 can be found in Jin & Candès (2023a) and is therefore omitted.

Denote $\mathbf{Z}_{1i}$ and $\mathbf{Z}_{2j}$ as tuples $\mathbf{Z}_{1i} = (\mathbf{X}_{1i}, Y_{1i}), \mathbf{Z}_{2j} = (\mathbf{X}_{2j}, Y_{2j})$ and the index set of calibration data $\mathcal{C} = \{i_1, i_2, \ldots, i_{n_c}\}$ where $|\mathcal{C}| = n_c$. Let $\mathcal{Z} = [\mathbf{Z}_{1i_1}, \ldots, \mathbf{Z}_{1i_{n_c}}, \mathbf{Z}_{2j}]$ be the unordered value set of $\mathbf{Z}_{1i_1}, \ldots, \mathbf{Z}_{1i_{n_c}}, \mathbf{Z}_{2j}$. Let $\tilde{\mathcal{X}} = \{\tilde{\mathbf{X}}_{21}, \ldots, \tilde{\mathbf{X}}_{2m}\}$ be the set of all sampled covariates. Note that the unordered value set $[p_{\mathrm{L},l}^{(j)}]_{l \neq j}$ are fully determined by $\tilde{\mathcal{X}}$, $\mathcal{Z}$ and $\{\mathbf{Z}_{2l}\}_{l \neq j}$. Moreover, the $p$-value $p_{\mathrm{L},j}$ is fully determined by $\tilde{\mathbf{X}}_{2j}$, $\mathcal{Z}$ and $\mathbf{Z}_{2j}$. As $\{\mathbf{Z}_{2l}\}_{l \neq j}$ is independent of $\{\mathbf{Z}_{1i_1}, \ldots, \mathbf{Z}_{1i_{n_c}}, \mathbf{Z}_{2j}\}$ conditional on $\tilde{\mathcal{X}}$, we know $p_{\mathrm{L},j}$ is independent of $\{p_{\mathrm{L},l}^{(j)}\}_{l \neq j}$ conditional on $\tilde{\mathcal{X}}$ and $\mathcal{Z}$. By the fact that $|\widehat{\mathcal{R}}_{j \to 0}|$ is fully determined by $[p_{\mathrm{L},l}^{(j)}]_{l \neq j}$, we further have

$$p_{\mathrm{L},j} \perp\!\!\!\perp |\widehat{\mathcal{R}}_{j \to 0}| \,\Big|\, \tilde{\mathcal{X}}, \mathcal{Z}.$$

By the independence we have

$$\mathbb{E}\left[\frac{\mathbb{I}\{p_{\mathrm{L},j} \leq \frac{\alpha |\widehat{\mathcal{R}}_{j \to 0}|}{m}, j \in \mathbb{H}_0\}}{|\widehat{\mathcal{R}}_{j \to 0}|} \,\Big|\, \tilde{\mathcal{X}}, \mathcal{Z}\right] \leq \mathbb{E}\left[\frac{\mathbb{I}\{p_{\mathrm{L},j} \leq \frac{\alpha |\widehat{\mathcal{R}}_{j \to 0}|}{m}\}}{|\widehat{\mathcal{R}}_{j \to 0}|} \,\Big|\, \tilde{\mathcal{X}}, \mathcal{Z}\right]$$
$$= \frac{\mathrm{Pr}\left(p_{\mathrm{L},j} \leq \frac{\alpha |\widehat{\mathcal{R}}_{j \to 0}|}{m} \,\Big|\, \tilde{\mathcal{X}}, \mathcal{Z}\right)}{|\widehat{\mathcal{R}}_{j \to 0}|}$$
$$\leq \frac{\mathrm{Pr}\left(p_{\mathrm{L},j}^* \leq \frac{\alpha |\widehat{\mathcal{R}}_{j \to 0}|}{m} \,\Big|\, \tilde{\mathcal{X}}, \mathcal{Z}\right)}{|\widehat{\mathcal{R}}_{j \to 0}|}.$$

where

$$p_{\mathrm{L},j}^* = \frac{\sum_{i \in \mathcal{C}} H(\mathbf{X}_{1i}, \tilde{\mathbf{X}}_{2j})(\mathbb{I}\{V_{2j} < V_{1i}\} + \xi_j \cdot \mathbb{I}\{V_{2j} = V_{1i}\}) + \xi_j \cdot H(\mathbf{X}_{2j}, \tilde{\mathbf{X}}_{2j})}{\sum_{i \in \mathcal{C}} H(\mathbf{X}_{1i}, \tilde{\mathbf{X}}_{2j}) + H(\mathbf{X}_{2j}, \tilde{\mathbf{X}}_{2j})}.$$

By the weighted exchangeability, $p_{\mathrm{L},j}^* \mid \tilde{\mathcal{X}}, \mathcal{Z} \sim \mathrm{U}[0,1]$. Taking the expectation we have

$$\mathrm{FDR} \leq \sum_{j=1}^m \mathbb{E}\left[\mathbb{E}\left\{\frac{\mathbb{I}\{p_{\mathrm{L},j} \leq \frac{\alpha |\widehat{\mathcal{R}}_{j \to 0}|}{m}\}}{|\widehat{\mathcal{R}}_{j \to 0}|} \,\Big|\, \tilde{\mathcal{X}}, \mathcal{Z}\right\}\right] \leq \sum_{j=1}^m \mathbb{E}\left\{\frac{\mathrm{Pr}\left(p_{\mathrm{L},j}^* \leq \frac{\alpha |\widehat{\mathcal{R}}_{j \to 0}|}{m} \,\Big|\, \tilde{\mathcal{X}}, \mathcal{Z}\right)}{|\widehat{\mathcal{R}}_{j \to 0}|}\right\}$$
$$= \sum_{j=1}^m \frac{\alpha}{m} = \alpha,$$

which completes the proof. □

## F.7 PROOF OF THEOREM 3

Define the uncomputable oracle $p$-value as

$$p_{\mathrm{L},j}^* = \frac{\sum_{i \in \mathcal{C}} H(\mathbf{X}_{1i}, \tilde{\mathbf{X}}_{2j})\mathbb{I}\{\bar{V}_{2j} \leq \bar{V}_{1i}\} + \xi_j \cdot H(\mathbf{X}_{2j}, \tilde{\mathbf{X}}_{2j})}{\sum_{i \in \mathcal{C}} H(\mathbf{X}_{1i}, \tilde{\mathbf{X}}_{2j}) + H(\mathbf{X}_{2j}, \tilde{\mathbf{X}}_{2j})}.$$

By Lemma 1, if $\{(\mathbf{X}_{1i}, \mathbf{Y}_{1i}, S_{1i})\}_{i \in \mathcal{C}} \cup \{(\mathbf{X}_{2j}, \mathbf{Y}_{2j}, S_{2j})\}_{j=1}^m$ are exchangeable, this oracle $p$-value will be super-uniform

$$\mathrm{Pr}(p_{\mathrm{L},j}^* \leq \alpha) \leq \alpha.$$

Note that $\bar{V}_{2j} = \max\{V_{2j,s} : Y_{2j,s} \notin \mathcal{A}_s\}$. This indicates

$$p_{\mathrm{L},j}^* = \min\{p_{\mathrm{L},j,s} : Y_{2j,s} \notin \mathcal{A}_s\}.$$

By direct tranformation we have

$$\text{FWER} = \Pr\left(\sum_{s=1}^{S_{2j}} \mathbb{I}\{Y_{2j,s} \notin \mathcal{A}_s, p_{\mathrm{L},j,s} \le \alpha\} > 0\right)$$

$$= \Pr\left(\bigcup_{Y_{2j,s} \notin \mathcal{A}_s} \{p_{\mathrm{L},j,s} \le \alpha\}\right)$$

$$= \Pr(p_{\mathrm{L},j}^* \le \alpha) \le \alpha,$$

which completes the proof. $\qquad\square$

## F.8   PROOF OF THEOREM 4

*Proof.* By our screening procedure, the conditional FWER can be transformed as

$$\Pr\left(\sum_{s=1}^{S_{2j}} \mathbb{I}\{Y_{2j,s} \notin \mathcal{A}_s, \delta_{j,s} = 1\} > 0 \,\Big|\, \mathbf{X}_{2j} \in \mathcal{B}\right) = \Pr\left(\bigcup_{Y_{2j,s} \notin \mathcal{A}_s} \{p_{\mathrm{L},j,s} \le \alpha\} \,\Big|\, \mathbf{X}_{2j} \in \mathcal{B}\right)$$

$$= \Pr(p_{\mathrm{L},j}^* \le \alpha \mid \mathbf{X}_{2j} \in \mathcal{B}).$$

Since we assume exchangeability, this is equivalent to the covariate shift setting with $g(\mathbf{x}) = \mathbb{I}\{\mathbf{x} \in \mathcal{B}\}/\Pr(\mathbf{X}_{2j} \in \mathcal{B})$. Therefore, we only need to compute the excess term in Lemma 2 with this specific $g$. By transformation

$$\mathbb{E}_{\boldsymbol{U} \sim K(\cdot)}\{|g(\mathbf{x} + h\boldsymbol{U}) - g(\mathbf{x})|\}$$

$$= \frac{1}{\Pr(\mathbf{X}_{2j} \in \mathcal{B})} \int K(\boldsymbol{u})(\mathbb{I}\{x + h\boldsymbol{u} \in \mathcal{B}, \mathbf{x} \notin \mathcal{B}\} + \mathbb{I}\{x + h\boldsymbol{u} \notin \mathcal{B}, \mathbf{x} \in \mathcal{B}\})\mathrm{d}\boldsymbol{u}$$

$$\le \frac{2}{\Pr(\mathbf{X}_{2j} \in \mathcal{B})} \int K(\boldsymbol{u})\mathbb{I}\{\|\boldsymbol{u}\|_2 \ge h^{-1}d(\mathbf{x}, \partial\mathcal{B})\}\mathrm{d}\boldsymbol{u}$$

$$= 2 \cdot \frac{\Pr_{\boldsymbol{U} \sim K(\cdot)}(\|\boldsymbol{U}\|_2 \ge h^{-1}d(\mathbf{x}, \partial\mathcal{B}))}{\Pr(\mathbf{X}_{2j} \in \mathcal{B})}.$$

Following the same proof to Lemma 2 we conclude

$$\Pr\left(\sum_{s=1}^{S_{2j}} \mathbb{I}\{Y_{2j,s} \notin \mathcal{A}_s, \delta_{j,s} = 1\} > 0 \,\Big|\, \mathbf{X}_{2j} \in \mathcal{B}\right) \le \alpha + 2\|f_{1,\mathbf{x}}\|_\infty \frac{\Pr_{\mathbf{X} \sim P_{H,\mathbf{x}}, \boldsymbol{U} \sim K(\cdot)}(\|U\|_2 \ge h^{-1}d(\mathbf{X}, \partial\mathcal{B}))}{\Pr(\mathbf{X}_{2j} \in \mathcal{B})}.$$

$\qquad\square$

## F.9   PROOF OF THEOREM 5

*Proof.* Remember that we assume $n_1 = n_2$ in the main text. We first introduce a lemma about our defined U-statistic.

**Lemma 5.** *For some kernel function $k(\mathbf{z}, \mathbf{z}')$ depending on $n_1$, define the two-sample U-statistic and its projection as*

$$U_{n_1} = \frac{1}{n_1^2} \sum_{i=1}^{n_1} \sum_{j=1}^{n_1} k(\mathbf{Z}_{1i}, \mathbf{Z}_{2j})$$

*and*

$$\widehat{U}_{n_1} = \frac{1}{n_1} \sum_{i=1}^{n_1} \mathbb{E}\{k(\mathbf{Z}_{1i}, \mathbf{Z}_{21}) \mid \mathbf{Z}_{1i}\} + \frac{1}{n_1} \sum_{j=1}^{n_1} \mathbb{E}\{k(\mathbf{Z}_{11}, \mathbf{Z}_{21}) \mid \mathbf{Z}_{2j}\} - \mathbb{E}\{k(\mathbf{Z}_{1i}, \mathbf{Z}_{2j})\}.$$

*Then if $\mathbb{E}\{k(\mathbf{Z}_{1i}, \mathbf{Z}_{2j})^2\} = o(n_1)$, $U_{n_1}$ and $\widehat{U}_{n_1}$ satisfy*

$$\sqrt{n_1}(U_{n_1} - \widehat{U}_{n_1}) = o_p(1).$$

This lemma is an extension of Lemma 3.1 in Powell et al. (1989) which constructs a similar result for one sample U-statistics of degree 2. The proof follows exactly the same and is therefore omitted.

Denote the unnormalized version of $\widehat{T}_w$ and its projection as $\widehat{T}_w^*$ and $\widehat{T}_{w,p}^*$. In our defined statistic $k(\mathbf{Z}_{1i}, \mathbf{Z}_{2j}) = H(\mathbf{X}_{1i}, \mathbf{X}_{2j})\widehat{D}_{ij}$. We first check the condition in Lemma 5. By integration and $\widehat{D}_{ij} < 1$

$$
\begin{aligned}
\mathbb{E}\{k(\mathbf{Z}_{1i}, \mathbf{Z}_{2j})^2\} &\leq \mathbb{E}\{H(\mathbf{X}_{1i}, \mathbf{X}_{2j})^2\} \\
&= \int \frac{1}{h^2 d} K\left(\frac{\mathbf{x}' - \mathbf{x}}{h}\right) f_{1,\mathbf{X}}(\mathbf{x}) f_{2,\mathbf{X}}(\mathbf{x}') \mathrm{d}\mathbf{x}\mathrm{d}\mathbf{x}' \\
&= \frac{1}{h^d} \int K(\boldsymbol{u})^2 f_{1,\mathbf{X}}(\mathbf{x}) f_{2,\mathbf{X}}(\mathbf{x} + h\boldsymbol{u}) \mathrm{d}\mathbf{x}\mathrm{d}\boldsymbol{u} \\
&\leq \frac{1}{h^d} \int K(\boldsymbol{u})^2 \mathrm{d}\boldsymbol{u}.
\end{aligned}
$$

Together with the assumption $n_1 h^d \to \infty$ we have $\mathbb{E}\{k(\mathbf{Z}_{1i}, \mathbf{Z}_{2j})^2\} = o(n_1)$.

Define the centralized projection statistics as $\psi_{1,n_1}(\mathbf{Z}_{1i}) = \mathbb{E}\{k(\mathbf{Z}_{1i}, \mathbf{Z}_{21}) \mid \mathbf{Z}_{1i}\}, \psi_{2,n_1}(\mathbf{Z}_{2j}) = \mathbb{E}\{k(\mathbf{Z}_{11}, \mathbf{Z}_{2j}) \mid \mathbf{Z}_{2j}\}$ and

$$
\psi_{1,n_1} = \frac{1}{n_1} \sum_{i=1}^{n_1} \mathbb{E}\{k(\mathbf{Z}_{1i}, \mathbf{Z}_{21}) \mid \mathbf{Z}_{1i}\} - \mathbb{E}\{k(\mathbf{Z}_{1i}, \mathbf{Z}_{2j})\},
$$

$$
\psi_{2,n_1} = \frac{1}{n_1} \sum_{j=1}^{m_1} \mathbb{E}\{k(\mathbf{Z}_{11}, \mathbf{Z}_{2j}) \mid \mathbf{Z}_{2j}\} - \mathbb{E}\{k(\mathbf{Z}_{1i}, \mathbf{Z}_{2j})\}.
$$

The projection can be decomposed as

$$
\begin{aligned}
\psi_{1,n_1}(\mathbf{Z}_{1i}) =& \mathbb{E}\{k(\mathbf{Z}_{1i}, \mathbf{Z}_{21}) \mid \mathbf{Z}_{1i}\} \\
=& \int \frac{1}{h^d} K\left(\frac{\mathbf{x} - \mathbf{X}_{1i}}{h}\right) \left[\frac{1}{2} - \mathbb{I}\{V_{1i} < V(\mathbf{x}, y)\} - \frac{1}{2}\mathbb{I}\{V_{1i} = V(\mathbf{x}, y)\}\right] f_{2,\mathbf{X}}(\mathbf{x}) f_2(y \mid \mathbf{x}) \mathrm{d}\mathbf{x}\mathrm{d}y \\
=& \int K(\boldsymbol{u}) \left[\frac{1}{2} - \mathbb{I}\{V_{1i} < V(\mathbf{X}_{1i} + h\boldsymbol{u}, y)\} - \frac{1}{2}\mathbb{I}\{V_{1i} = V(\mathbf{X}_{1i} + h\boldsymbol{u}, y)\}\right] \\
& \cdot f_{2,\mathbf{X}}(\mathbf{X}_{1i} + h\boldsymbol{u}) f_2(y \mid \mathbf{X}_{1i} + h\boldsymbol{u}) \mathrm{d}\boldsymbol{u}\mathrm{d}y \\
=& \int K(\boldsymbol{u}) \left[\frac{1}{2} - \mathbb{I}\{V_{1i} < \widehat{V}^*(\mathbf{X}_{1i}, y)\} - \frac{1}{2}\mathbb{I}\{V_{1i} = \widehat{V}^*(\mathbf{X}_{1i}, y)\}\right] f_{2,\mathbf{X}}(\mathbf{X}_{1i}) f_2(y \mid \mathbf{X}_{1i}) \mathrm{d}\boldsymbol{u}\mathrm{d}y \\
& + \psi_{1,n_1,r}(\mathbf{Z}_{1i}). \\
=& \psi_1(\mathbf{Z}_{1i}) + \psi_{1,n_1,r}(\mathbf{Z}_{1i}).
\end{aligned}
$$

By Assumption 1 and $h \to 0$ we know $\psi_{1,n_1,r}(\mathbf{Z}_{1i}) = o_p(1)$. Similar results also hold for $\psi_{2,n_1}(\mathbf{Z}_{2j})$ and $\psi_{2,n_1,r}(\mathbf{Z}_{2j})$.

Note that under the null hypothesis $v(\mathbf{x}, y) \equiv 1$. So we have

$$
\mathbb{E}\{H(\mathbf{X}_{1i}, \mathbf{X}_{2j})D_{ij}\} = \mathbb{E}\left\{H(\mathbf{X}_{1i}, \mathbf{X}_{2j})\left(\frac{1}{2} - \xi_j\right)\right\} = 0
$$

If Assumption 2 holds then

$$
\sqrt{n_1}\mathbb{E}\{H(\mathbf{X}_{1i}, \mathbf{X}_{2j})\widehat{D}_{ij}\} = \sqrt{n_1}\mathbb{E}\{H(\mathbf{X}_{1i}, \mathbf{X}_{2j})(\widehat{D}_{ij} - D_{ij})\} = o_p(1)
$$

If $P_{1,\mathbf{X}} = P_{2,\mathbf{X}}$ then under the null hypothesis we have $P_1 = P_2$ and therefore $\mathbb{E}\{H(\mathbf{X}_{1i}, \mathbf{X}_{2j})\widehat{D}_{ij}\} = 0$ by symmetric.

Now the statistic can be decomposed as

$$
\begin{aligned}
\widehat{T}_w^* =& (\widehat{T}_w^* - \widehat{T}_{w,r}^*) + \frac{1}{n_1}\sum_{i=1}^{n_1}[\psi_1(\mathbf{Z}_{1i}) - \mathbb{E}\{\psi_1(\mathbf{Z}_{1i})\}] + \frac{1}{n_1}\sum_{j=1}^{n_1}[\psi_2(\mathbf{Z}_{2j}) - \mathbb{E}\{\psi_2(\mathbf{Z}_{2j})\}] \\
& + \frac{1}{n_1}\sum_{i=1}^{n_1}[\psi_{1,n_1,r}(\mathbf{Z}_{1i}) - \mathbb{E}\{\psi_{1,n_1,r}(\mathbf{Z}_{1i})\}] + \frac{1}{n_1}\sum_{j=1}^{n_1}[\psi_{2,n_1,r}(\mathbf{Z}_{2j}) - \mathbb{E}\{\psi_{2,n_1,r}(\mathbf{Z}_{2j})\}] \\
& + \mathbb{E}\{H(\mathbf{X}_{1i}, \mathbf{X}_{2j})\widehat{D}_{ij}\} \\
=& \frac{1}{n_1}\sum_{i=1}^{n_1}[\psi_1(\mathbf{Z}_{1i}) - \mathbb{E}\{\psi_1(\mathbf{Z}_{1i})\}] + \frac{1}{n_1}\sum_{j=1}^{n_1}[\psi_2(\mathbf{Z}_{2j}) - \mathbb{E}\{\psi_2(\mathbf{Z}_{2j})\}] + o_p(1/\sqrt{n_1}).
\end{aligned}
$$

As $\psi_1(\mathbf{Z}_{1i})$ and $\psi_2(\mathbf{Z}_{2j})$ are independent random variables not depending on $n_1$, we have

$$
\frac{\widehat{T}_w^*}{\sigma_{n_1}} \xrightarrow{d} N(0,1). \tag{F.3}
$$

where

$$
\sigma_{n_1}^2 = \frac{1}{n_1}\mathrm{Var}\{\psi_1(\mathbf{Z}_{1i})\} + \frac{1}{n_1}\mathrm{Var}\{\psi_2(\mathbf{Z}_{2j})\}.
$$

Hereafter we only need to prove the consistency of the variance estimator. Denote $\sigma_{n_1}^{*2} = \mathrm{Var}(\widehat{T}_w^*)$, the non-asymptotic variance can be decomposed as

$$
\begin{aligned}
\sigma_{n_1}^{*2} =& \frac{1}{n_1^2}\mathbb{E}\{H(\mathbf{X}_{1i}, \mathbf{X}_{2j})^2\widehat{D}_{ij}^2\} + \frac{n_1-1}{n_1^2}\mathbb{E}\left\{H(\mathbf{X}_{1i}, \mathbf{X}_{2j})H(\mathbf{X}_{1k}, \mathbf{X}_{2j})\widehat{D}_{ij}\widehat{D}_{kj}\right\} \\
& + \frac{n_1-1}{n_1^2}\mathbb{E}\left\{H(\mathbf{X}_{1i}, \mathbf{X}_{2j})H(\mathbf{X}_{1i}, \mathbf{X}_{2k})\widehat{D}_{ij}\widehat{D}_{ik}\right\} - \frac{2n_1-1}{n_1^2}\mathbb{E}\left[\left\{H(\mathbf{X}_{1i}, \mathbf{X}_{2j})\widehat{D}_{ij}\right\}\right]^2
\end{aligned}
$$

By similar integration computation, the cross-term expectation converges to $\mathrm{Var}\{\psi_1(\mathbf{Z}_{1i})\}$ and $\mathrm{Var}\{\psi_2(\mathbf{Z}_{2j})\}$ in probability. Removing the vanishing term we have $\sigma_{n_1}^{*2} - \sigma_{n_1}^2 = o_p(1/n_1)$ and $\sigma_{n_1}^*/\sigma_{n_1} \xrightarrow{p} 1$.

The variance estimator $\widehat{\sigma}_w^2$ takes the form

$$
\begin{aligned}
\widehat{\sigma}_w^2 =& \frac{1}{n_1^2}\sum_{i=1}^{n_1}\left\{\frac{1}{n_1}\sum_{j=1}^{n_1}H(\mathbf{X}_{1i}, \mathbf{X}_{2j})\widehat{D}_{ij}\right\}^2 + \frac{1}{n_1^2}\sum_{j=1}^{n_1}\left\{\frac{1}{n_1}\sum_{i=1}^{n_1}H(\mathbf{X}_{1i}, \mathbf{X}_{2j})\widehat{D}_{ij}\right\}^2 \\
& - \frac{1}{n_1^4}\sum_{i=1}^{n_1}\sum_{j=1}^{n_1}H(\mathbf{X}_{1i}, \mathbf{X}_{2j})^2\widehat{D}_{ij}^2 - \frac{2}{n_1}\left\{\frac{1}{n_1^2}\sum_{i=1}^{n_1}\sum_{j=1}^{n_1}H(\mathbf{X}_{1i}, \mathbf{X}_{2j})\widehat{D}_{ij}\right\}^2 \\
=& \frac{1}{n_1^2}\left\{\frac{1}{n_1^2}\sum_{i=1}^{n_1}\sum_{j=1}^{n_1}H(\mathbf{X}_{1i}, \mathbf{X}_{2j})^2\widehat{D}_{ij}^2\right\} + \frac{1}{n_1}\cdot\frac{1}{n_1^3}\sum_{i=1}^{n_1}\sum_{j\neq k}H(\mathbf{X}_{1i}, \mathbf{X}_{2j})H(\mathbf{X}_{1i}, \mathbf{X}_{2k})\widehat{D}_{ij}\widehat{D}_{ik} \\
& + \frac{1}{n_1}\cdot\frac{1}{n_1^3}\sum_{j=1}^{n_1}\sum_{1\neq k}H(\mathbf{X}_{1i}, \mathbf{X}_{2j})H(\mathbf{X}_{1k}, \mathbf{X}_{2j})\widehat{D}_{ij}\widehat{D}_{kj} - \frac{2}{n_1}\left\{\frac{1}{n_1^2}\sum_{i=1}^{n_1}\sum_{j=1}^{n_1}H(\mathbf{X}_{1i}, \mathbf{X}_{2j})\widehat{D}_{ij}\right\}^2
\end{aligned}
$$

It is easy to see that $\widehat{\sigma}_w^2$ consists of term-wise approximations of the expectations in $\sigma_{n_1}^{*2}$. By further computing the second moments of the estimators and Markov's inequality we have $\widehat{\sigma}_w^2 - \sigma_{n_1}^{*2} = o_p(1/n_1)$. Together with (F.3) we conclude

$$
\widehat{T}_w = \frac{\widehat{T}_w^*}{\widehat{\sigma}_w} \xrightarrow{d} N(0,1).
$$

$\square$

### F.10 PROOF OF THEOREM 6

*Proof.* By the same proof with Theorem 5 we know

$$\frac{\widehat{T}_w^* - \mathbb{E}\{H(\mathbf{X}_{1i}, \mathbf{X}_{2j})\widehat{D}_{ij}\}}{\widehat{\sigma}_w} \xrightarrow{d} Z \sim N(0, 1). \tag{F.4}$$

By Assumption 2 we have $\mathbb{E}\{H(\mathbf{X}_{1i}, \mathbf{X}_{2j})(\widehat{D}_{ij} - D_{ij})\} = o_p(1/\sqrt{n_1})$. Combining with the consistency of $\widehat{\sigma}_w$ we know $\sqrt{n_1}\widehat{\sigma}_w/\sigma_w \xrightarrow{p} 1$ and

$$\frac{\widehat{T}_w^* - \mathbb{E}\{H(\mathbf{X}_{1i}, \mathbf{X}_{2j})\widehat{D}_{ij}\}}{\widehat{\sigma}_w} = Z + \frac{\mathbb{E}\{H(\mathbf{X}_{1i}, \mathbf{X}_{2j})D_{ij}\}}{\sigma_w/\sqrt{n_1}}(1 + o_p(1)) + o_p(1).$$

The rest is to compute the bias term $\mathbb{E}\{H(\mathbf{X}_{1i}, \mathbf{X}_{2j})D_{ij}\}$. By integration

$$\begin{aligned}
&\mathbb{E}\{H(\mathbf{X}_{1i}, \mathbf{X}_{2j})D_{ij}\} \\
&= \int \frac{1}{h^d} K\left(\frac{\mathbf{x}_1 - \mathbf{x}_2}{h}\right)\left[\frac{1}{2} - \mathbb{I}\{V^*(\mathbf{x}_1, y_1) < V^*(\mathbf{x}_2, y_2)\}\right] f_{1,\mathbf{X}}(\mathbf{x}_1)f_{2,\mathbf{X}}(\mathbf{x}_2)f_1(y_1 \mid \mathbf{x}_1)f_2(y_2 \mid \mathbf{x}_2)\mathrm{d}\mathbf{x}_1\mathrm{d}\mathbf{x}_2\mathrm{d}y_1\mathrm{d}y_2 \\
&= \int K(\boldsymbol{u})\left[\frac{1}{2} - \mathbb{I}\{V^*(\mathbf{x}_1, y_1) < V^*(\mathbf{x}_1 + h\boldsymbol{u}, y_2)\}\right] f_{1,\mathbf{X}}(\mathbf{x}_1)f_{2,\mathbf{X}}(\mathbf{x}_1 + h\boldsymbol{u})f_1(y_1 \mid \mathbf{x}_1)f_2(y_2 \mid \mathbf{x}_1 + h\boldsymbol{u}) \\
&\qquad \mathrm{d}\mathbf{x}_1\mathrm{d}\boldsymbol{u}\mathrm{d}y_1\mathrm{d}y_2 \\
&= \left(\int \left[\frac{1}{2} - \mathbb{I}\{V^*(\mathbf{x}_1, y_1) < V^*(\mathbf{x}_1, y_2)\}\right] f_{1,\mathbf{X}}(\mathbf{x}_1)f_{2,\mathbf{X}}(\mathbf{x}_1)f_1(y_1 \mid \mathbf{x}_1)f_2(y_2 \mid \mathbf{x}_1)\mathrm{d}\mathbf{x}_1\mathrm{d}y_1\mathrm{d}y_2\right)(1 + o_p(1)) \\
&= \left(\int \left[\frac{1}{2} - \mathbb{I}\{V^*(\mathbf{x}_1, y_1) < V^*(\mathbf{x}_1, y_2)\}\right] V^*(\mathbf{x}_1, y_1)f_{1,\mathbf{X}}(\mathbf{x}_1)f_{2,\mathbf{X}}(\mathbf{x}_1)f_2(y_1 \mid \mathbf{x}_1)f_2(y_2 \mid \mathbf{x}_1)\mathrm{d}\mathbf{x}_1\mathrm{d}y_1\mathrm{d}y_2\right)(1 + o_p(1)) \\
&= \mathbb{E}_{\mathbf{X} \sim P_{2,\mathbf{X}}}\left\{f_{1,\mathbf{X}}(\mathbf{X})\left(\frac{1}{2} - \mathbb{E}_{Y_1, Y_2 \sim P_{2,Y|\mathbf{X}}}[V^*(\mathbf{X}, Y_1)\mathbb{I}\{V^*(\mathbf{X}, Y_1) < V^*(\mathbf{X}, Y_2)\}]\right)\right\}(1 + o_p(1))
\end{aligned}$$

And by transformation

$$\begin{aligned}
&\mathbb{E}_{Y_1, Y_2 \sim P_{2,Y|\mathbf{X}}}[V^*(\mathbf{X}, Y_1)\mathbb{I}\{V^*(\mathbf{X}, Y_1) < V^*(\mathbf{X}, Y_2)\}] \\
&= \frac{1}{2}\mathbb{E}_{Y_1, Y_2 \sim P_{2,Y|\mathbf{X}}}[V^*(\mathbf{X}, Y_1)\mathbb{I}\{V^*(\mathbf{X}, Y_1) < V^*(\mathbf{X}, Y_2)\}] + \frac{1}{2} - \frac{1}{2}\mathbb{E}_{Y_1, Y_2 \sim P_{2,Y|\mathbf{X}}}[V^*(\mathbf{X}, Y_1)\mathbb{I}\{V^*(\mathbf{X}, Y_1) \geq V^*(\mathbf{X}, Y_2)\}] \\
&= \frac{1}{2} - \frac{1}{2}\mathbb{E}_{Y_1, Y_2 \sim P_{2,Y|\mathbf{X}}}[\{V^*(\mathbf{X}, Y_1) - V^*(\mathbf{X}, Y_2)\}\mathbb{I}\{V^*(\mathbf{X}, Y_1) \geq V^*(\mathbf{X}, Y_2)\}] \\
&= \frac{1}{2} - \frac{1}{4}\mathbb{E}_{Y_1, Y_2 \sim P_{2,Y|\mathbf{X}}}\{|V^*(\mathbf{X}, Y_1) - V^*(\mathbf{X}, Y_2)|\}.
\end{aligned}$$

The bias term is then simplified as

$$\mathbb{E}\{H(\mathbf{X}_{1i}, \mathbf{X}_{2j})D_{ij}\} = \frac{1}{4}\mathbb{E}_{\mathbf{X} \sim P_{2,\mathbf{X}}, Y, Y' \sim P_{2,Y|\mathbf{X}}}\{f_{1,\mathbf{X}}(\mathbf{X})|V^*(\mathbf{X}, Y) - V^*(\mathbf{X}, Y')|\}(1 + o_p(1)).$$

Combining all the results above we conclude

$$\widehat{T}_w = \frac{\widehat{T}_w^*}{\widehat{\sigma}_w} = Z + \frac{\mathbb{E}\{H(\mathbf{X}_{1i}, \mathbf{X}_{2j})D_{ij}\}}{\sigma_w/\sqrt{n_1}}\{1 + o_p(1)\} + o_p(1) = \frac{\sqrt{n_1}\delta_w}{4\sigma_w}\{1 + o_p(1)\} + Z + o_p(1).$$

$\square$

