# OpenReview forum: "Conditional Testing based on Localized Conformal $p$-values"
_ICLR.cc/2025/Conference — ICLR 2025 Poster_

### Official Review · Reviewer_hT7S · 2024-10-24

**Soundness:** 3
**Presentation:** 2
**Contribution:** 3
**Rating:** 6
**Confidence:** 4

**Summary:**

This work investigates the possible application of localized conformal p-values, which can be naturally constructed from recent localized conformal prediction methods.

**Strengths:**

This work extends the localized conformal prediction methods to a broader range of inferential targets.

**Weaknesses:**

I believe the writing and presentation should be improved, and more effort needs to be made to demonstrate the usefulness of the results.

**Questions:**

1. Could the authors provide clarifications on the goal of Theorem 3? It feels somewhat distracting from the main focus of this work, as it essentially presents an 'asymptotic property under additional assumptions,' while the main point of the conformal prediction framework is the finite-sample theory without distributional assumptions.

2. Regarding Section 3.2, 'Conditional Outlier Detection,' I'm not sure if the hypothesis (3.1) is what is actually being tested. Specifically, isn't the p-value super-uniform under a weaker condition than $H_{0,j}$ in (3.2), since it is marginally valid?

My understanding is that the localized conformal prediction method from Hore and Barber basically achieves the marginal coverage guarantee, while the conditional miscoverage rates are "likely to be small" if we frame everything in a distribution-free sense.

Thus, I feel that the outlier detection counterpart of the localized conformal prediction framework controls the FDR for testing marginal outliers, but conditional outliers are more likely to be detected—rather than directly detecting conditional outliers with a distribution-free guarantee. Am I missing something?

3. (continued from point 2) In Theorem 5, is the goal of the condition $P_{1,X}=P_{2,X}$ to claim that any detected distribution shift is a label shift rather than a covariate shift, i.e., that any outlier is a conditional outlier? My confusion is that under the condition $P_{1,X}=P_{2,X}$, the null hypothesis in (3.1) is essentially equivalent to stating that the marginal distribution $P_{(X,Y)}$ is the same for the test point and the calibration set, since it is the combination of "$P_X$ is the same" and "$P_{Y|X}$ is the same".

So, to me, Theorem 5 sounds like "Algorithm 1 controls FDR for testing marginal outliers. But if we additionally assume that $P_X$ is the same, then an outlier means $P_{Y|X}$ is different, implying that it is a conditional outlier", which is not very interesting compared to directly detecting the conditional outliers.

Therefore, as mentioned in point 2, for a more useful result, shouldn't the statement be something like: "The procedure detects marginal outliers, but conditional outliers are likely to be better detected if we use the localized conformal p-values"?

4. Could the authors provide explanations for the terms in Theorem 4 and 7, since it's difficult to see how small they are?

5. minor comment: "independency" -> "independence" in page 28

---

> ### Author Response · Authors · 2024-11-21
> **Response to reviewer hT7S (Part 1/2)**
>
> Thank you for your valuable feedback and for taking the time to engage with our work. We appreciate your insightful comments and suggestions. We have made some revisions to improve the presentation according to you and other reviewer's suggestions, which we hope can meet your expectation. For your questions, we have prepared a detailed point-by-point response. We sincerely hope our clarifications resolve your concerns.
>
> > Q1: Could the authors provide clarifications on the goal of Theorem 3? It feels somewhat distracting from the main focus of this work, as it essentially presents an 'asymptotic property under additional assumptions,' while the main point of the conformal prediction framework is the finite-sample theory without distributional assumptions.
>
> To Q1: Of course. You are right on the point that the main point of the conformal prediction framework is the finite-sample theory. Most finite sample theoretical results characterize the validity of the method in terms of type I error (miscoverage, FDR, e.g.), and we also formulate the validity of our p-value in this sense. However, it is often impractical to characterize the efficiency (interval width, power, e.g.) of conformal methods in finite sample. The asymptotic result provided here is to illustrate that our p-value will be more efficient than the classical conformal p-value since it converges to the conditional distribution instead of the marginal one, which means it can better capture changes in the conditional distribution.
>
> > Q2: Regarding Section 3.2, 'Conditional Outlier Detection,' I'm not sure if the hypothesis (3.1) is what is actually being tested. Specifically, isn't the p-value super-uniform under a weaker condition than $ H_{0,j} $ in (3.2), since it is marginally valid?
>
> > Q3: <font style="color:#000000;">(continued from point 2) In Theorem 5, is the goal of the condition P1,X=P2,X to claim that any detected distribution shift is a label shift rather than a covariate shift, i.e., that any outlier is a conditional outlier? My confusion is that under the condition P1,X=P2,X, the null hypothesis in (3.1) is essentially equivalent to stating that the marginal distribution P(X,Y) is the same for the test point and the calibration set, since it is the combination of "PX is the same" and "PY|X is the same".
>
> >So, to me, Theorem 5 sounds like "Algorithm 1 controls FDR for testing marginal outliers. But if we additionally assume that PX is the same, then an outlier means PY|X is different, implying that it is a conditional outlier", which is not very interesting compared to directly detecting the conditional outliers. As mentioned in point 2, for a more useful result, shouldn't the statement be something like: "The procedure detects marginal outliers, but conditional outliers are likely to be better detected if we use the localized conformal p-values"?
>
> To Q2 and Q3:
>
> We are sorry for the confusion. Our procedure can detect conditional outliers for sure. But as you have figured out, the procedure detects not only conditional outliers, but also marginal ones. The assumption $ P_{1,X}=P_{2,X} $ combined with the null hypothesis leads to "$ P_{X,Y} $ is the same", so marginal outliers can be detected by our procedure with finite sample FDR control given that inliers in $ \mathcal{D_2} $ have the same distribution $P_{X,Y}$ with data in $ \mathcal{D}_1$.
>
> Our point for this problem is that the conditional outliers are easily missed by the existing outlier detection procedure using classical conformal p-values. Our procedure is proposed to mitigate this issue and the localized conformal p-value is defined specifically to capture the deviation in conditional distribution.
>
> For your concern to directly detect conditional outliers without the assumption $P_{1,X}=P_{2,X}$, we need to deal with potential covariate shift when constructing the localized p-values. This requires the marginal density ratio of $ X $to be known according to the weighted conformal prediction. Since the ratio is unknown in most cases, an estimation can never be perfect and the finite sample validity will be violated. So we instead make this assumption to give a fintie sample FDR control guarantee.
>
> Per your nice suggesstion, we have added a remark to clarify the usefulness of our results after the FDR control theorem (in line 363-367 in our revised version). Hope this is acceptable to you!

---

> ### Author Response · Authors · 2024-11-21
> **Response to reviewer hT7S (Part 2/2)**
>
> > Q4: Could the authors provide explanations for the terms in Theorem 4 and 7, since it's difficult to see how small they are?
>
> To Q4: Of course. For any fixed subset $ \mathcal{B} $, the $ L_\infty $-norm and the denominator in both theorems are fixed constants. For the excess term to be small, we only need the numerator to be small. In the probability, $ U $ has a fixed dsitribution since the base kernel $ K(\cdot) $ is pre-given. As $ h\to0 $, the density $ f_{H,{\bf X},\mathcal{A}} $ will converge to $ f_{1,{\bf X},\mathcal{A}} $, so the random variables $ d({\bf X},\mathcal{B}) $ and $ \Vert U\Vert_2 $ are both of order $ O_p(1) $. If we modify the event as $ \Pr(h\Vert U\Vert_2\geq d({\bf X},\mathcal{B})) $, it will be clear that the excess term converges to 0 as $ h\to0 $.
>
> > Q5: minor comment: "independency" -> "independence" in page 28.
>
> To Q5: Thank you for pointing that out. We have corrected the wording in the revised paper.

---

> > ### Comment · Reviewer_hT7S · 2024-11-26
> >
> > I appreciate the authors' efforts in providing a thorough response. My questions have been satisfactorily addressed, and I have updated the score accordingly.

---

> > > ### Author Response · Authors · 2024-11-26
> > >
> > > We sincerely appreciate your recognition and your valuable suggestions for improving the quality of our paper!

---

### Official Review · Reviewer_mWDU · 2024-11-02

**Soundness:** 3
**Presentation:** 3
**Contribution:** 3
**Rating:** 6
**Confidence:** 3

**Summary:**

This paper introduces localized conformal p-values for conditional testing problems.
Specifically, the authors inverting the prediction intervals which are constructed by
 existing works to adapt to the conditional distribution of the response. They establish theoretical results on the fundamental properties of their defined p-values, as well as applications of these p-values to conditional testing. The empirical results demonstrate the effectiveness of their proposed methods.

**Strengths:**

The paper presents a clear problem statement, solid theoretical results, and the experimental results effectively reflect the efficiency of their proposed method.

**Weaknesses:**

The structure of paper could be better. The authors could give concise and appropriate names to the important theorems to summarize their meanings, and mark the others as lemmas. There are many theorems in the main text of the paper, so presenting the theorems this way would make them more intuitive and help readers better understand the meaning of the theorems.

**Questions:**

1.What circumstances the "average FDP above nominal level" shown in Figure 1 would become negative?

2.Based on the experimental results in the paper, it is observed that in many cases, LCP-od exhibits greater fluctuations compared to CP. How can this phenomenon be explained?

---

> ### Author Response · Authors · 2024-11-21
> **Response to reviewer mWDU**
>
> > W: The structure of paper could be better. The authors could give concise and appropriate names to the important theorems to summarize their meanings, and mark the others as lemmas.
>
> To weakness:  Thank you for your valuable suggestion regarding the structure of the paper. We have revised the presentation of our theorems accordingly by giving concise, descriptive names to the key theorems and marking others as lemmas and you can find them in our current paper. We agree that this approach makes the results more intuitive and enhances readability. We appreciate your insight in helping us improve the clarity of our work!
>
>
> > Q1: What circumstances the "average FDP above nominal level" shown in Figure 1 would become negative?
>
>
> To Q1:  Apologies for the confusion. By "average FDP above nominal level" we mean the empirical quantity $ \operatorname{FDR} - \alpha $. Thus, when $ \operatorname{FDR} < \alpha $, this value will be negative. We chose to plot "average FDP above $ \alpha $" instead of plotting FDR directly because we also consider results under different nominal levels, $ \alpha\in\{0.05,0.1,0.15,0.2\} $ in Figure 3 and Figure 5. Plotting the gap between FDR and $ \alpha $ provides convenience across varying nominal levels. And for consistency, we also plot this value in Figure 1, although in this case, $ \alpha $is fixed.
>
>
> > Q2: Based on the experimental results in the paper, it is observed that in many cases, LCP-od exhibits greater fluctuations compared to CP. How can this phenomenon be explained?
>
>
> To Q2: Thank you for your questions. In fact, the fluctuations arise from the construction of LCP. Since we define LCP with a random sampling step (as illustrated in line 186), this introduces additional randomness, leading to greater variations or fluctuations. We have mentioned this in the concluding section. The randomization technique used here is important and necessary to ensure the LCP is finite sample valid under the null. To resolve the instability issue, we are working on an extension of our method not requiring randomization. However, incorporating such changes here would require significant alterations to the overall framework of the methodology. Hope this is acceptable to you!
>
>
> We hope that the above interpretation can ease your doubts.

---

### Official Review · Reviewer_sXfi · 2024-11-03

**Soundness:** 3
**Presentation:** 3
**Contribution:** 3
**Rating:** 8
**Confidence:** 3

**Summary:**

Authors introduce a novel (conformal) p-value by converting an existing conformal prediction method that is locally valid, asymptotically conditionally valid, and computationally advantageous. They further discuss the applicability of this approach in various multiple conditional testing problems. From an application perspective, their contributions are four folds: balanced data selection, conditional outlier detection, conditional label screening, and a two-sample conditional distribution test. Each application is rigorously justified through theoretical analysis, and the utility of the proposed methods is empirically demonstrated via extensive numerical experiments.

**Strengths:**

The most notable strength of this work lies in the variety of applications presented. The proposed applications of LCP are all promising, largely due to the LCP’s ability to effectively address conditional testing. In particular, the authors contextualize each application; for instance, in the case of the conditional label screening problem, they suggest that this application could be used to ensure the factuality of large language models, thereby justifying its utility.

Furthermore, the authors thoroughly discuss the theoretical justification of the proposed applications. They rigorously prove how each problem controls specific quantities (e.g., FSER, FDR, and FWER) by synthesizing a considerable amount of literature to support their claims. This strength, the ability to creatively adapt existing methodologies and introduce multiple new applications, demonstrating an original contribution to the field.​

**Weaknesses:**

While they have successfully introduced locally conformal p-values and thoroughly demonstrated their various applications and theoretical properties, their work seems limited in terms of novelty. Specifically, the locally conformal p-values appear to be derived from directly converting the prediction intervals proposed by Hore & Barber (2023), and their application seems to involve integrating these p-values into several already established methods.

Meanwhile, one of the main contributions of their work lies in achieving the asymptotically conditional super-uniform property of the new conformal p-values. Specifically, Theorem 4 and Theorem 7 provide theoretical justification for this, though the authors merely mention that a specific choice of $g(x)$ renders the covariate shift problem equivalent to conditional validity, which is not clear. While this approach poses no issue in terms of soundness, as previously discussed by Hore & Barber (2023), it may lack readability in this regard.

**Questions:**

1. In the application of conditional outlier detection, the goal was to control the FDR using the BH procedure, whereas in the conditional label screening problem, LCP guarantees FWER control. Is there any reason behind controlling different quantities in each case? What motivated the decision to adhere to FWER control instead of FDR control in conditional label screening despite this conservativeness?

  2. In Section 3.1, within the balanced data selection problem, it is assumed that the response variable is not observed. In this case, how is $V_{2j}$ specifically obtained? My understanding is that both $X_{2j}$ and $Y_{2j}$ are required for this, but $Y_{2j}$  is not observed. In this case, is an arbitrary value of $Y_{2j}$ (i.e., any y which is not in $\mathcal{A}$) satistfing $\mathbb{H}_{0j}$ used?

  3. Could you provide more detailed explanation or intuition for how the specific choice of g(x) makes the covariate shift problem equivalent to conditional validity?

  4. In Section 3.3, the context of large language models’ factuality is introduced to frame the conditional label screening problem. However, the numerical study presents results only on more structured synthetic and real datasets. I was wondering if any numerical study reflecting the natural language processing context described in the paper has been attempted. If so, what were the outcomes of those experiments? If not, what additional research would be necessary to conduct such studies?

---

> ### Author Response · Authors · 2024-11-21
> **Response to reviewer sXfi (Part 1/2)**
>
> > W1: While they have successfully introduced locally conformal p-values and thoroughly demonstrated their various applications and theoretical properties, their work seems limited in terms of novelty. Specifically, the locally conformal p-values appear to be derived from directly converting the prediction intervals proposed by Hore & Barber (2023), and their application seems to involve integrating these p-values into several already established methods.
>
> > W2: Meanwhile, one of the main contributions of their work lies in achieving the asymptotically conditional super-uniform property of the new conformal p-values. Specifically, Theorem 4 and Theorem 7 provide theoretical justification for this, though the authors merely mention that a specific choice of $ g({\bf x}) $ renders the covariate shift problem equivalent to conditional validity, which is not clear. While this approach poses no issue in terms of soundness, as previously discussed by Hore & Barber (2023), it may lack readability in this regard.
>
> To W1 & W2:  Thank you for your valuable feedback and for taking the time to engage with our work. We appreciate your insightful comments and suggestions. We first would like to address your concern regarding novelty and highlight some contributions.
>
> + You're correct on the point that our p-value is derived from the prediction interval proposed by Hore & Barber (2023).  However, we believe our work goes beyond this foundation in significant ways. Specifically, in our applications, the conditional outlier detection problem has not yet been explored from the conformal inference perspective, and to the best of our knowledge, the label screening problem is also novel.
> + Furthermore, applying our localized conformal p-values to various application problems demands considerably more effort to establish theoretical guarantees as the localized conformal p-value is more complex than the classical version, highlighting the technical depth of our contributions.
>
> To address your doubts about Theorem 4, Theorem 7 in previous paper (coressponding to Theorem 1, Theorem 4 in our revised version now) and the readability, we have prepared a detailed point-by-point response to your questions. We sincerely hope our clarifications resolve your concerns.

---

> ### Author Response · Authors · 2024-11-21
> **Response to reviewer sXfi (Part 2/2)**
>
> > Q1: Is there any reason behind controlling different quantities in each case? What motivated the decision to adhere to FWER control instead of FDR control in conditional label screening despite this conservativeness?
>
> To Q1:  The rationale for using FWER instead of FDR in the conditional label screening problem lies in our target of perform screening within each multivariate response vector. In this scenario, the multiple testing procedure will be performed on each individual in $ \mathcal{D}_2$, rather than on the entire $ \mathcal{D}_2 $ for once. Controlling the FDR at the individual level only ensures the process make few mistakes among different coordinates of an observation, this may lead to potential errors within each observation. By contrast, controlling the FWER ensures that the screening process does not make mistakes on at least $ 1- \alpha $ proportion of individuals in $ \mathcal{D}_2 $, offering a more robust approach for this type of analysis.
>
> > Q2: In this case, how is $ V_{2j} $ specifically obtained? My understanding is that both $ X_{2j} $ and  $ Y_{2j} $ are required for this, but $ Y_{2j} $ is not observed. In this case, is an arbitrary value of $ Y_{2j} $ (i.e., any $y$ which is not in $ \mathcal{A} $) satistfing $ \mathbb{H}_{0j} $ used?
>
> To Q2: Since our work encompasses a range of different applications, the score function $ V $ may depend on both $ X $ and $ Y $ or solely on $ X $, depending on the specific context. This is clarified after our definition of the localized conformal p-value in Section 2.1.  In the data selection problem, $ Y_{2j} $ is unobserved and the score $ V_{2j} $ involves only a prediction value $ \widehat{\mu}(X_{2j}) $  for any machine learning model $ \widehat{\mu} $ to reflect evidence for $ Y_{2j}\in\mathcal{A} $. So we do not need $ Y_{2j} $ to compute $ V_{2j} $.
>
> > Q3: Could you provide more detailed explanation or intuition for how the specific choice of g(x) makes the covariate shift problem equivalent to conditional validity?
>
> To Q3: Of course. In our definition, the function $ g $ denotes the density ratio function between $ P_{2,X} $ and $ P_{1,X} $. When $ P_{1,X}=P_{2,X} $, we can take another distribution $ Q_{2,X} $ by restricting $ P_{2,X} $ on any given region $ \mathcal{B}\subset\mathbb{R}^d $. According to the theorem, $ \Pr(p_j\leq\alpha\mid X_{2j}\in\mathcal{B})\leq\alpha+\Delta $, where $ \Delta $ stands for the excess term with $ g(x)=\frac{dQ_{2,X}}{dP_{1,X}}(x)=\frac{\mathbb{I}(x\in\mathcal{B})}{\Pr(X\in\mathcal{B})} $ and will vanish as $ h\to0 $. Therefore, the asymptotic robustness against covariate shift implies asymptotic conditional validity of the localized conformal p-value. We have added comments in lines 216–219 of the revised paper to enhance clarity and intuitiveness.
>
> > Q4: However, the numerical study presents results only on more structured synthetic and real datasets. I was wondering if any numerical study reflecting the natural language processing context described in the paper has been attempted. If so, what were the outcomes of those experiments? If not, what additional research would be necessary to conduct such studies?
>
> To Q4:  Thank you for this insightful question, and we apologize for not including experiments directly related to LLM factuality in our study.  We address your points as follows:
>
> 1. **Challenges of Conducting LLM Factuality Experiments**:  Implementing experiments in the context of LLM factuality requires a complex setup. Specifically, it involves generating responses, claims, and corresponding factuality scores for a large, diverse set of queries, which can be computationally intensive. Due to these requirements and our primary focus on developing and validating the label screening methodology itself (rather than specializing in generative modeling), this task is somewhat beyond our current capabilities. We agree that this is indeed a direction worth continuing to learn and explore in our future work; however, it is distinct from the focus of our current work.
> 2. **Versatility of the Proposed Label Screening Method**: Moreover, we 'd like to emphasize that **our proposed method is designed as a versatile tool that can be easily implemented by practitioners, including those working with LLMs**. Although we have demonstrated its application in more structured contexts, such as the medical diagnostic problem in our real-data experiment, this framework is readily adaptable to the LLM factuality setting, as well as other domains requiring label screening. While we focused on structured synthetic and real datasets in our experiments, the method itself is not limited to these scenarios.
>
> We hope this clarifies our approach and its flexibility across different applications, as well as the reasons why we did not include a numerical experiment about the LLM factuality application. We appreciate your thoughtful input and hope this explanation is satisfactory to you!

---

> > ### Comment · Reviewer_sXfi · 2024-11-26
> >
> > I sincerely appreciate your efforts to carefully address my concerns! In particular, the part regarding the asymptotically conditional validity of the conformal p-value (or the controllability of several error rates) has been properly addressed and is now more straightforward. Additionally, from the perspective of contribution, I acknowledge that the new quantity you proposed is novel in certain aspects, and the theoretical justification required for this is not marginal.
> >
> > Accordingly, I am raising my score and recognize that this work is worthy of acceptance.

---

> > > ### Author Response · Authors · 2024-11-27
> > >
> > > Thank you for raising the score! Addressing your thoughtful concerns has helped us improve the presentation of our paper. We will also continue to explore unresolved issues, such as the LLM factuality experiment, in our future work. Thank you again for your invaluable suggestions and your participation in reviewing our work!

---

### Official Review · Reviewer_24ms · 2024-11-04

**Soundness:** 3
**Presentation:** 3
**Contribution:** 3
**Rating:** 8
**Confidence:** 4

**Summary:**

The paper introduces localized conformal $p$-values and applies them to several conditional testing problems, including conditional outlier detection, conditional label screening, and two-sample conditional distribution testing.

**Strengths:**

This paper presents the concept of localized conformal $p$-values and demonstrates their application to some interesting testing problems. The authors conduct appropriate experiments to illustrate the performance of their method.

**Weaknesses:**

Please refer to the questions below.

**Questions:**

1. Could the authors provide a rigorous proof of the density ratio of $X_{2j}$ and $X_{1i}$ conditional on $\tilde{X}_{2j}$.

2. Where is $d$ defined?

3. In different applications of conditional testing, what are the score functions $V$?

4. Please clarify the statement "This indicates that using the LCP for data selection can lead to a more balanced selection result since the PSER inflation for different sub-groups is bounded", and the statement related to fairness.

5. Avoid using abbreviations before introducing them, such as PRDS.

6. In conditional outlier detection, what are the theoretical and empirical performances if $P_{1,X} \neq P_{2,X}$?

7. From my understanding, the theoretical results do not account for the estimation errors. Moreover, kernel functions can lead to the curse of dimensionality, which hinders application in high-dimensional settings.

8. How do the tuning parameters in kernel functions affect the performance, and how should these parameters be chosen in practice?

9. Why does the method in two-sample conditional distribution testing not require randomization? Does the proposed method in this context require the marginal density ratio? Moreover, why does the type I error increase with sample size in Figure 7?

---

> ### Author Response · Authors · 2024-11-21
> **Response to reviewer 24ms (Q1-Q3)**
>
> > Q1:  Could the authors provide a rigorous proof of the density ratio of $ X_{2j} $ and $ X_{1i} $ conditional on $ \tilde{X}_{2j} $ .
>
>
> To Q1: Of course. Due to the limitations of the TeX language on OpenReview, we did not include the proofs directly here. Instead, the proofs are provided in the appendix F.1 of our revised paper (line1293-line1315). The proof refers to the proof of Proposition 1 in [1]. And we noticed an error in line 166 of our previous paper (the constant c is omitted), which we have now corrected. Thank you for bringing this to our attention! !
>
> Reference:
>
> [1] Hore R, Barber R F. Conformal prediction with local weights: randomization enables local guarantees[J]. arXiv preprint, 2023.
>
>
>
> > Q2: Where is $ d $ defined?
>
> To Q2: Apologies for the ambiguity in this notation. We missed defining it, which may have led to confusion.  We have now revised the relevant notations in the paper.
>
> + In line 178, $ d $ in the kernel function represents the dimension of the conditional features. Specifically, in our synthetic-data example Scenario A1, $ d=1 $represents the dimension of time feature $ t $. And in Scenarion B1, $ d=2 $ represents the demension of $ s=(s_1,s_2) $.
> + In line 446 of our previous paper, $ d $ in $ X_{d-1} $ represents the dimension of the covariates, including the time feature. To avoid confusion, we have now used a different notation, $ d^* $, to distinguish between these meanings. That is, we have the whole covariate vector of dimension $ d^* $, and $ d $ of them are used for conditioning.
>
> Thank you for your attention to detail!
>
> > Q3: In different applications of conditional testing, what are the score functions $ V $?
>
> To Q3:  We have introduced the corresponding score functions in our paper. To clarify further, we’ll explain about score function step by step.
>
> + First, the score function $ V $ is a widely-used term in literature on conformal inference ([1]-[2]). Intuitively, $ V(x,y) $ measures how well a true value $ y $ conforms to the prediction of the model at $ x $. A commonly used score function in the problem of conformal prediction is the absolute residual score $ V(x,y)=|y-\hat{f}(x)|$ .
> + Second, similar to the statement in point 1 above, for all applications of conditional testing in our work,  with a score function $ V $ chosen properly, a larger score will indicate stronger evidence against the pre-specified null hypothesis. The difference arises from different null hypothesis.
> +  Specifically, here we make a clarification about the choice of $ V $ for different applications as follows:
>     - For the balanced data selection, $ V $ can be chosen as  a prediction value  $ \widehat{\mu}(X_{2j}) $ for any machine learning model $ \widehat{\mu} $ to approximate $ \operatorname{Pr}(Y_{2j} \in \mathcal{A}) $.
>     - For the conditional outlier detection: (1) If the label $ Y $ is avaible,$ V $can be chosen as the absolute residual scores or CQR scores proposed by [3]; (2) for the case where label$ Y $ is unavaible and , we can apply one-class classifier and choose $ V $ as the score of one-class classifier.
>     - For the conditional label screening, since the test data is unlabelled, the score $ V $ is chosen as a prediction value $ \widehat{\mu}(X_{2j}) $  for any machine learning model $ \widehat{\mu} $ to reflect evidence for $ Y_{2j}\in\mathcal{A} $ .
>     - For the two-sample conditional distribution test, $ V(x,y)=\widehat{\frac{f_1(y|x)}{f_2(y|x)}} $ .
>
> References:
>
> [1] Jin Y, Candès E J. Selection by prediction with conformal p-values[J]. JMLR, 2023.
>
> [2] Hore R, Barber R F. Conformal prediction with local weights: randomization enables local guarantees[J]. arXiv preprint, 2023.
>
> [3] Romano Y, Patterson E, Candes E. Conformalized quantile regression[J]. NeurIPS, 2019.

---

> ### Author Response · Authors · 2024-11-21
> **Response to reviewer 24ms (Q4-Q6)**
>
> > Q4: Please clarify the statement "This indicates that using the LCP for data selection can lead to a more balanced selection result since the PSER inflation for different sub-groups is bounded", and the statement related to fairness.
>
>
> To Q4: Sorry for any confusion. We'd like to provide the following clarifications step by step.
>
> + First, a "more balanced selection result" means that, unlike the naive method that make decisions by directly using unweighted CP, our proposed method, which leverages LCP, achieves a more balanced PSER across different sub-groups.
> + Specifically, by "balanced", we refer to the conditional PSER being more evenly distributed across different subgroups. In Theorem 4, we provide a conditional PSER inflation bound for each subgroup corresponding to $ X \in \mathcal{B} $, while the naive method only guarantees marginal PSER control, as formulated in line 289 (in the revised version). For example, by using LCP, we can expect the conditional PSER across groups (e.g., male and female) to be more uniformly close to $ \alpha $, whereas the unweighted CP may cause the conditional PSER for certain groups to exceed the nominal level $ \alpha $ significantly.
> + Second, regarding the statement about fairness, we'd like to explain why our result about conditional PSER bound is similar to that of Rava et al. (2021) [1]. In their work, they propose an algorithm that achieves statistical parity by controlling the FDR across protected groups. Both our approach and [1] aim to mitigate imbalances in group-wise error rates; however, they refer to this property as "fairness", while we describe it as "balance". And We have revised the corresponding statements in line 314 in the paper to prevent any potential ambiguity for readers.
>
> We hope that the above interpretation can ease your doubts about this statement.
>
> Reference:
>
> [1]  Rava B, Sun W, James G M, et al. A burden shared is a burden halved: A fairness-adjusted approach to classification[J]. arXiv preprint arXiv:2110.05720, 2021.
>
>
> > Q5: Avoid using abbreviations before introducing them, such as PRDS.
> >
>
> To Q5:  Thank you for your helpful comment, and apologies for any ambiguity. "PRDS" stands for positive regression dependent on a subset, a commonly used term in the literature on multiple testing and FDR control, as seen in [1]. We have updated in line 338 in the revised version to introduce the relevant definitions before using abbreviations and have reviewed the paper to ensure all abbreviations are properly introduced.
>
> Reference:
>
> [1] Stephen Bates, Emmanuel Candes, Lihua Lei, Yaniv Romano, and Matteo Sesia. Testing for outliers with conformal p-values. The Annals of Statistics, 51(1):149–178, 2023.
>
>
>
> > Q6: In conditional outlier detection, what are the theoretical and empirical performances if $ P_{1,X}\neq P_{2,X} $?
> >
>
> To Q6:
>
> + First, theoretically, the finite-sample validity of LCP requires the assumption that $ P_{1,X}=P_{2,X} $, as stated in Theorem 2 in line 207. Since the validity of conditional outlier detection algorithm in Theorem 5 relies on the validity of LCP, the finite-sample FDR control for conditional outlier detection also require the same condition $ P_{1,X}=P_{2,X} $.
> + If $ P_{1,X} \neq P_{2,X} $, the LCP remains asymptotically valid, but it is much more difficult to give a deviation bound of the FDR above $ \alpha $. Regarding to the empirical performance when $ P_{1,X}\neq P_{2,X} $ , we have conducted an additional synthetic-data experiment.  Consider a scenario with covariate shift.  For  the training data and calibration data, $ X_1,\dots, X_{k-1}\sim U[-1,1] $, and for the testing data,$ X_1,\dots, X_{k-1}\sim U[-1,1.5] $. The other settings are the same as those in Scenarioa A1. The results across 100 replications are shown in the table below. We can see that the LCP can guarantee FDR control empirically when the shift is mild and still exhibits higher power than the CP.
>
> | Method | $ n $ | $ \operatorname{FDR} $ | $ \operatorname{Power} $ |
> | :---: | :---: | :---: | :---: |
> | LCP | 2400 | 0.154 | 0.556 |
> | | 3200 | 0.150 | 0.629 |
> | | 4000 | 0.155 | 0.644 |
> | CP | 2400 | 0.139 | 0.341 |
> | | 3200 | 0.144 | 0.375 |
> | | 4000 | 0.124 | 0.375 |

---

> ### Author Response · Authors · 2024-11-21
> **Response to reviewer 24ms (Q7-Q9)**
>
> > Q7: From my understanding, the theoretical results do not account for the estimation errors. Moreover, kernel functions can lead to the curse of dimensionality, which hinders application in high-dimensional settings.
>
> To Q7: We would like to provide the following explanations in a step-by-step manner.
>
> + **About the estimation errors**:
>
> (1) First, benefits from the distribution-free and model/algorithm-agnostic property of conformal inference, the validity of our proposed LCP for the first three applications, i.e., balanced data selection, conditional outlier detection and conditional label screening, is not affected by the estimation errors. Therefore, the theoretical results for these three applications (Theorem 4,5,6,7) do not account for the estimation error indeed make sense.
>
> (2) Second, for the last application, i.e., two-sample conditional distribution test, we have considered the estimation errors in our theoretical results, which is reflected in the assumption 2. As we have explained in line 814-line815, the assumption in line 812 means we require the approximation error is sufficiently small after local-weighting.
>
> + **About the curse of dimensionality**:
>
> We acknowledge that the kernel function can lead to the curse of dimensionality. However, in the context of our conditional testing applications, we focus on practical cases that condition only on temporal or spatial variables. In these cases, the dimension of the conditional variables is low enough for our proposed LCP to perform well.
>
>
>
> > Q8: How do the tuning parameters in kernel functions affect the performance, and how should these parameters be chosen in practice?
> >
>
> To Q8:  Note that the demension $ d $ in kernel functions is specified by the data. The only tuning parameter in kernel functions is the bandwidth $h$.
>
> + **How  $ h $ affects the performance of LCP**:
>
> The choice of bandwidth $ h $ affects the extent of localization.  A smaller $ h $ means more localization. Intuitively, for larger values of $ h $, the LCP behaves similarly to the unweighted conformal $ p $ values.
>
> + **Choice of $ h $**:
>
>  In our experiments, we choose $ h $ according to the rule of thumb, which is a commonly used criterion for bandwidth selection in the classical nonparametric literature [1]. Specifically, we set $ h=(n/2)^{-1/(d+2)}$ and we have stated this in line 939 in our paper about the implementation details.
>
> Hope the above explaination is acceptful to you!
>
> Reference:
>
> [1] B. W. Silverman. Density Estimation for Statistics and Data Analysis. Chapman and Hall, 1986.
>
>
>
> > Q9: Why does the method in two-sample conditional distribution testing not require randomization? Does the proposed method in this context require the marginal density ratio? Moreover, why does the type I error increase with sample size in Figure 7?
> >
>
> To Q9:  Thank you for your nice questions! We'd like to make the following clarifications point by point.
>
> 1. **About randomization**:
> + As we explained in line 789, in the context of the two-sample conditional distribution test, the problem becomes much more challenging, and it is no longer necessary to require the $ p $-values to be finite-sample valid. Specifically, in this case, even though we use the LCP with randomization, it is still insufficient for finite-sample control, and the external randomness itself introduces a kind of loss. The advantages of using the simplified LCP outweigh the disadvantages due to its simplicity and stability , whereas the introduction of additional randomness does not. Therefore, we choose to use a simplified LCP without randomization here.
>
> 2. **About marginal density ratio**:
> + No, we don't require the true marginal density ratio. As illustrated in the main text line774 and in Algorithm 3 in line 918-931, we only need to estimate the marginal density ratio $ \widehat{g}(x) $ instead of requiring the true marginal density ratio.
> 3. **About type I error and sample size**:
> + First, we’d like to clarify that for our proposed method (LCT), the type I error can be controlled across all scenarios, including varying sample sizes, as shown in Figure 7. Thus, the type I error of our method does not increase with sample size. In contrast, the CT method requires the training model to be correctly specified and shows a severely inflated type I error rate when the model is misspecified. The DCT performs better than CT due to its doubly robust property.
> + The validity of our method in experiments demonstrates that our assumptions are milder than those of the benchmark methods.
> + However, it is difficult to capture the relationship between type I error and sample size for these two benchmark methods. As shown in the top line charts of Figure 7, the type I error does not always increase with sample size for DCT.
>
>
> Thank you once again for your effort and insightful questions about our work; they have greatly helped us improve our paper! And we hope these clarifications help address your concerns.

---

> > ### Comment · Reviewer_24ms · 2024-11-26
> > **Additional comments**
> >
> > Thank you for your detailed response.
> >
> > I have an additional question regarding Lemma 1. Why is the finite-sample validity achieved by only assuming $P_{1, X} = P_{2,X}$? From the proof of Equation (2.3), the conditional density ratio $dP_{2, Y|X}/dP_{1,Y|X}$ also matters.

---

> > > ### Author Response · Authors · 2024-11-26
> > >
> > > We sincerely thank you for your careful review of our paper and for identifying another small mistake. The assumption here should be $P_{1,X}=P_{2,X}$ together with the null hypothesis (3.1), and we have now corrected it (to $P_1=P_{2,j}$). The conditional outlier detection problem was the initial motivation for developing the localized conformal p-value, and this lemma (previously a theorem) was included immediately after (3.1) in a much earlier version of our paper. During the revisions we made in preparation for submission to ICLR, we overlooked updating the assumption to align with the revised structure. We sincerely apologize for this oversight. We have now carefully reviewed the entire paper to check for and address any other potential typos or errors. Thank you again for your detailed feedback and for bringing this to our attention.

---

> > > > ### Comment · Reviewer_24ms · 2024-11-26
> > > >
> > > > Thank you for your response. My concerns have been well addressed.
> > > >
> > > > I appreciate the thorough investigation of LCP in terms of both finite-sample validity under exchangeability and asymptotic validity under covariate shift, as well as the comprehensive demonstration of its interesting applications. I have decided to raise my score.

---

> > > > > ### Author Response · Authors · 2024-11-27
> > > > >
> > > > > Thank you for raising the score and identifying several mistakes. Addressing your invaluable feedback has greatly improved the precision and presentation of our paper. We sincerely appreciate your time and effort in reviewing our work.

---

### Meta-Review · Area_Chair_Egq6 · 2024-12-22

**Metareview:**

This paper applies conformal inference techniques to perform conditional testing. They define a new localized conformal p value, and use this to perform several practical conditional tests including outlier detecting and label screening. Reviewers found this to be novel and some were more positive after a healthy rebuttal period. I agree with their assessment and appreciate the clean ideas, as well as their application to classical tasks and connections to U-statistics.

**Additional Comments On Reviewer Discussion:**

Authors engaged with reviewers and several reviewers raised their scores in the rebuttal period

---

### Decision · Program_Chairs · 2025-01-22

Accept (Poster)